

# Predicting the spatial distribution of soil organic carbon stock in Swedish forests using remotely sensed and site-specific variables

**Kpade O. L. Hounkpatin[1], Johan Stendahl[1], Mattias Lundblad[1] , Erik Karltun[1],**

[1] Department of Soil and Environment, Swedish University of Agricultural Sciences, P.O. Box 7014, SE-75007, Uppsala, Sweden

*Correspondence to*: Kpade O. L. Hounkpatin (ozias.hounkpatin@slu.se)

**Abstract**

The status of the SOC stock at any position in the landscape is subject to a complex interplay of soil-state factors

operating at different scales and regulating multiple processes resulting either in soils acting as a net sink or net source of carbon. Forest landscapes are characterized by high spatial variability and key drivers of SOC stock might be specific for subareas compared to those influencing the whole landscape. Consequently, separately calibrating models for subareas (local models) that collectively cover a target area can result in different prediction accuracy and SOC stock drivers compared to a single model (global model) that covers the whole area. The goal

of this study was therefore to (1) assess how global and local models differ in predicting the humus layer, mineral soil and total SOC stock in Swedish forests, (2) identify the key factors for SOC stock prediction and their scale of influence.

We use the Swedish National Forest Soil Inventory (NFSI) database and a digital soil mapping approach to

evaluate the prediction performance using Random Forest modelling calibrated locally for the northern, central and southern Sweden (local models) and for the whole Sweden (global model). Models were built by considering (1) only site characteristics which are recorded on the plot during NFSI, (2) remotely sensed variables and (3) both site characteristics and remotely sensed variables.

Local models are generally more effective for predicting SOC stock after testing on independent validation data. Using remotely sensed variables together with NFSI data indicates that such covariates have limited predictive strength but that site specific variables from the NFSI covariates show better explanatory strength for SOC stocks. The most important covariates that influence the humus layer, mineral soil and total SOC stock were related to the site characteristic covariates and include the soil moisture class, vegetation type, soil type and soil texture. Future

studies could focus in mapping these influential site covariates which have potential for future SOC stock prediction models.

## 1. Introduction

About 30 % of the global terrestrial C stock is stored in forests with 60 % located below ground (Pan et al., 2011). These forests act mostly as a large net sink for atmospheric carbon but concerns exist for potential release of C under the impact of global warming over the next century (Price et al., 2013;Kauppi et al., 2014). Moreover, the intensification of forest management for timber, fibre, and fuel to satisfy an ever-increasing demand will likely affect the dynamic of the forest C pool. In recent decades, many studies have focused on assessing the soil organic

carbon (SOC) stock in forest soils (Kumar et al., 2016;Ottoy et al., 2017;Sheikh et al., 2009;Prietzel and



Christophel, 2014) which is crucial to meet the requirement of the climate convention and the Kyoto protocol for reporting all sources and sinks of carbon dioxide and also for the estimation of potential carbon credits (Buchholz et al., 2014;Jandl et al., 2007). In that context, analysis of the carbon (C) cycle in forests is central to understanding management and climate-induced changes in global C pool.


Increased availability of remote sensing data and development of spatial statistical methods has led to an increased use of digital soil mapping (DSM) (Minasny and McBratney, 2016). DSM aims at estimating the spatial distribution of soil classes or soil properties by coupling field and laboratory observations with spatial and non-spatial environmental covariates via quantitative relationships. Many studies used DSM approaches for predicting

SOC stock at different scales and for various land use/land cover, climate and across a wide range of soil types (Söderström et al., 2016;Tranter et al., 2011;Beguin et al., 2017;Mansuy et al., 2014). These studies use different modelling techniques ranging from multiple linear regression to machine learning models such as artificial neural network and support vector machine and boosted regression trees.

The accuracy and precision of predictions resulting from modelling over a large landscape are often reported to be poor because of the spatial heterogeneity encompassing different soil types, topography and soil properties (Grimm et al., 2008;Schulp and Verburg, 2009;Schulp et al., 2013;Tang et al., 2017). Generally, models are applied to the whole study area without prior stratification. However,  models could be calibrated separately for subareas and their predictions can then be combined to cover the whole area (Somarathna et al., 2016). Since spatial variability

is an important characteristic of forest landscapes, key drivers of SOC stock might be specific for subareas compared to those influencing the whole landscape. Management decision in relation to driving factors of SOC stock will likely be more cost-effective as models gain in reliability for specific areas within a given landscape.

Building on the  soil state-factor (climate, organisms, relief, parent material, age) equation developed by Jenny

(1941),  McBratney et al. (2003) introduced the conceptual framework for DSM referred to as SCORPAN which complemented the former with the inclusion of the location coordinates. The relative contribution of any of these factors to model accuracy in DSM vary and some turn out to be more relevant as explanatory variables compared to others. Ottoy et al. (2017) identified relief (highest groundwater level), soil (clay fraction), land use (tree genus) as main predictors for mapping SOC stock in forest soils in Belgium while Mansuy et al. (2014) reported  relief

and climatic variables as the key covariates in mapping C, N and texture in Canadian managed forests. Vasques et al. (2016) recorded parent material among the key covariates in mapping soil properties in tropical dry forest in Brazil. These studies and many others rely mostly on remote sensing variables existing as maps while survey data which present site specific information are left out during modelling. However, soil factors affecting different processes in the landscape operate at different scale and taking into account site specific variables would inform

model local variability which might not be captured by remote sensing variables.

The goal of this study was therefore to (1) assess how global and local models differ in predicting the humus layer, mineral soil and total SOC stock in Sweden forest ecosystems, (2) evaluate to which extent and at which scale remotely sensed variables can explain the variability of SOC stock compared to site specific variables in the



Swedish forest and, (3) identify variables which may have potential for future prediction models in forest
SOC stock assessments.

## 2.  Materials and methods

### 2.1 Data description

Forest data came from the Swedish National Forest Soil Inventory (NFSI) and the National Forest Inventory (NFI).
The NFSI runs concurrently every year with the NFI and consist in repeated survey of forest vegetation and soil
chemical and physical properties. Data from the following inventory periods were considered in the present study:
1993 – 2002, 2003 – 2012 and 2013 – 2015. The NFSI are conducted on ca 23 500 permanent plots (Figure 1)
revisited every 5 years with a radius of 10 m covering all land uses in Sweden except urban areas, cultivated land
and the high mountains. The plots are distributed based on a stratified and random national grid system covering
all the Swedish forest soils. They are organized in quadratic clusters (tracts) consisting in 8 (in the north) to 4 (in
the southwest) circular (314 m$^2$) sample plots. Each plot of the NFSI are inventoried once every 10 years.


Soil samples are collected in a subset of the plots with humus sampling on c. 10 000 plots and mineral soil sampling
on c. 4500 plots (Stendahl et al., 2017). Humus layer volumetric samples are taken using a soil core (core diameter
10 cm) from the O horizon down to 30 cm depth. The mineral soil is sampled at 0-10, 10-20 and 55-65 cm depth
from the mineral soil surface. These samples are dried at 35°C and sieved to <2 mm. Total C is determined for all
samples by dry combustion with elemental analysers (LECO CNS-1000 and LECO TruMac CN). Total O horizon
SOC stock is calculated from sampled amount of soil material and C concentration of the sample. The total mineral
SOC stock down to 50 cm depth for each site is calculated using the SOC stock of measured layers with empirical
model for bulk density (Nilsson and Lundin, 2006), corrections for stoniness (Stendahl et al., 2009) and
interpolation between measured layers. Since potential SOC stock change is very small compared to the entire
SOC stock the averaged SOC stock between the inventories was considered representative of the plots and was
therefore considered for all computations and modelling in order to reduce variability between plots. The organic
and mineral soil SOC stock were summed up to get the total SOC stock.

### 2.2 Explanatory variables for prediction


The set of covariates used in this study consist of topographic variables, climate variables, geochemical and
gamma-ray data, historical land use maps and site characteristics (Table 1).

Topographic variables were computed from high-resolution digital elevation models (DEM) derived from Light
Detection And Ranging (LiDAR) produced by the Swedish National Mapping Agency. It was originally created
with 2-m spatial resolution (Grid 2+, LMV, Gävle, Sweden). However, the initial DEM was resampled in ArcGIS
10 software package using the aggregation procedure with the averaging method to a final resolution of 10 m × 10
m which is reasonable for the data considered in the present study. The topographical covariates were computed
using the SAGA GIS software. However, the depth to water (DTW, 2 x 2 m) considered in this study is an
estimation of the elevation along a defined least-cost-path (Lidberg et al., 2019;Murphy et al., 2008). The depth to



groundwater was obtained from the Swedish Forest Agency (Source: geodata.skogsstyrelsen.se) and computes the difference in elevation in relation to surrounding cells following the vertical flow path.

Climate maps (1 km x 1 km) of the annual mean temperature and annual precipitation for 1970-2000 were obtained from the WorldClim platform (Fick and Hijmans, 2017). The Geological Survey of Sweden (SGU) has produced geochemical data based mainly on the spatial distribution of till which covers about 75% of the Swedish landscape. The following base cations Ca (ppm), Mg (ppm), K (ppm), Na (ppm) and Mn (ppm) were considered for the present study in predicting carbon storage.

Several studies in Sweden pointed to some correlation between gamma-ray data and soil properties (Piikki et al., 2015;Söderström and Eriksson, 2013). Gamma-ray data are recorded by SGU since 1968 with measurements carried out along flight lines at 200 m interval in general. The flight heights were 30 m up to 1994 while subsequent surveys were carried out at 60 m altitude. The concentrations of the following radioisotopes $^{40}$K, $^{232}$Th, and $^{238}$U are measured and corrected for background and cosmic radiation (Erdi-Krausz et al., 2003). The gamma-ray

dataset was filtered for values < 0 which were omitted as they are mostly related to water entities. The resulting gamma-ray data as well as the geochemical data were interpolated into maps either by ordinary kriging or inverse distance weighing when geostatistic assumption such as normal distribution were not met.

The Swedish Forest Agency has developed several forest attributes maps based on the combination of satellite

images and field data from the NFI (Nilsson et al., 2017). Maps (25 x 25 m) of the stand age, tree biomass, tree height and stem volume produced for the year 2010 were used in the present study. Auffret et al. (2017) digitized some historical map series (Ekonomiska kartan) which were initially published in 1935 - 1978. The digitized versions of these maps (1 x 1 m) were only produced for the southern part of Sweden and present past major land use, settlements and infrastructure. These maps were available per counties but were merged into a single raster

file in ArcMap 10.7. For the present study, we consider two variants of these maps: (1) areas which were cropland and are now forest lands, and (2) areas which were grasslands and are now forest lands.

The records of site characteristics (Table 1) are also carried out during the NFSI. Site description include soil types, soil moisture class, soil texture class, vegetation type and parent material class. The soil classification was

based on the World Reference Base (WRB) for soil resources. The position of the water table was the main criterion for defining classes of soil moisture. The texture index was made by manual assessment in the field, e.g. through rolling and washing test.  The vegetation type as reported in Table 1 was defined by combining the descriptions of the field layers. Field layers consisted of four main types which are categorized from fertile  to poor,  namely  herb  types (tall or low),  grounds  without field layer,  grass  types and dwarf-shrub types.


### 2.3 Prediction models: Random Forest and quantile regression forests

The Random Forest (RFR) algorithm was selected for SOC stock prediction. Additionally, the quantile regression forest (QRF) was used to estimate the standard deviation related to the predictions.




RFR is a classification and regression method that builds multiple decision trees. For regression, the tree predictors provide numerical output instead of class labels for classification (Breiman, 2001). The RFR is able to model complex and nonlinear relationships between input predictors and response variables. The RFR is characterized by double randomness in the construction of the decisions trees. An ensemble of growing decision trees is

generated by combining bagging (bootstrap aggregating) along with random feature selection. Bagging consists in producing training datasets (bootstrap sample) by drawing randomly with replacement from the original training dataset generated. A regression tree is fitted to each of the bootstrap samples from a random subset of the input predictors when deciding to split a node. For any new given input $X = x$, RFR provides the prediction of a single tree as a weighted average of the original observations $Y_i, (i = 1, ..., n)$ in each node.


$$\hat{\mu}(x) = \sum_{i=1}^{n} w_i(x, \theta) Y_i \qquad (1)$$

where $w_i$ is the weight vector which results either in a positive constant when the observation $(Y_i, X_i)$ is inherent to the leaf generated from the random vector of variables or is 0 if otherwise. The weight vector $w_i$ is defined as

follows:

$$w_i(x, \theta) = \frac{1\{x_i \in R_{l(x,\theta)}\}}{\{j : x_j \in R_{l(x,\theta)}\}} \qquad (2)$$

$R_{l(x,\theta)}$ is the rectangular subspace defined by the leaf $l(x, \theta)$ of the tree built from the random vector of variables

$\theta$. The conditional mean $E(Y|X = x)$ is computed by averaging the predictions of $k$ single trees which are individually built with independent vectors having similar distributions. The weighted average of trees is computed as follows:

$$w_i(x) = k^{-1} \sum_{t=1}^{k} w_i(x, \theta_t) \qquad (3)$$

The final prediction of the RFR regression is given by:

$$\hat{\mu}(x) = \sum_{i=1}^{n} w_i(x) Y_i \qquad (4)$$

The number of trees to grow in the RFR model (ntree) and the number of randomly selected predictor variables at

each node (mtry) are the two key parameters to be tuned for RFR modelling. To reduce computational load, the ntree was set at 500 while the mtry was tuned using the grid search method in the R "caret" package (Kuhn, 2015) with fiftyfold cross validation. The importance of each input predictor can be assessed by the RFR based on the mean decrease accuracy (MDA). The MDA is computed by (i) randomly permuting the values of each predictor within the OOB set, and (ii) measuring the reduction in model accuracy resulting from that permutation. The

hypothesis is that this permutation would result in little to no effect on model accuracy for less important covariates, while significant drop will follow the permutation of important covariates.

### 2.4 Covariate layers processing for subareas

We considered three subareas in Sweden (Figure 1) which are hereafter reported as northern (North), central

(center) and southern (South) Sweden areas in the remaining of the paper. These areas were defined by merging the northern, central and southern climatic regions which were considered in Ortiz et al. (2013). A buffer of 10 m was considered for the shape files of each subarea to create overlapping zones which ensured smooth transition while merging by averaging the SOC stock values within these shared units. The covariates were delimited for



each subarea. They were resampled to 10 m resolution using the bilinear method for continuous variables and the nearest neighbor method for categorical covariates. A value to point extraction was carried out by overlaying the coordinates of the sampling points of each subareas over the stacked raster files in R (Kuhn, 2015). The pixel values of each subarea were compiled to form the database of the humus layer, mineral and total SOC stock.

### 2.5 Modelling with different category of covariates: global and local models

For modelling, three categories of covariates were considered: (1) only the plot level site specific variables (SSV), (2) all the covariates without the SSV, namely those based whether directly or indirectly on remote sensing variables (RSV) and (3) both the SSV and RSV (allV). Modelling with RFR was carried out with each category of covariate related to its subareas as well as for the compiled dataset for the whole Sweden. Moreover, to reduce computation time while keeping relatively the same level of accuracy, we (1) used feature pre-processing

capabilities implemented in the caret package (Kuhn, 2017) of R to remove highly correlated expressions using a cutoff point of 0.80 and (2) the recursive feature elimination (RFE) to select the optimal set of covariates for each RFR model. The RFE functions by carrying out variable importance classification then proceeds by eliminating iteratively the least important features (Gomes et al., 2019;Hounkpatin et al., 2018). For each RFR model, the RFE was carried out and therefore model-specific optimal set of covariates were identified for both whole Sweden and

subareas.

The RFR models were calibrated and validated for the whole area of Sweden and were therefore called "global models". Each of the datasets of the humus layer, mineral soil, total SOC stock, were divided into two sets. The first set (80%) was used for calibration while the second (20%) was considered for independent validation. We

trained the models based on tenfold cross validation with 5 repetitions using the R "caret" package (Kuhn, 2015).

The models created for each of these subareas are hereafter reported as "local models". The local models were build following the same procedure as for the global models by splitting the local datasets into training (80%) and validation (20%) set. Also, tenfold cross validation with 5 repetitions were considered for calibrating the local

models. Table 2 presents the details about the specific sampling distribution for the calibration and validation datasets over subareas and whole Sweden. It should be noted that the same validation set of each subarea was also considered by the global model for assessing the model performances.

### 2.6 Assessment of model performance and mapping

To compare model performance, we computed several assessment metrics : $R^2$, Lin's concordance (Lawrence and Lin, 1989) correlation ($\boldsymbol{\rho_c}$), root mean square error (RMSE), and mean absolute error (MAE).

$$RMSE = \left[\frac{1}{n} \sum_{i=1}^{n} (P_i - O_i)^2 \right]^{1/2} \tag{6}$$

$$R^2 = \frac{\sum_{i=1}^{n}(P_i - \mu_{obs})^2}{\sum_{i=1}^{n}(O_i - \mu_{obs})^2} \tag{7}$$

$$\rho_c = \frac{2\rho\sigma_{pred}\sigma_{obs}}{\sigma_{pred}^2 + \sigma_{obs}^2 + \left(\mu_{pred} - \mu_{obs}\right)^2} \tag{8}$$



$$MAE = \frac{1}{n} \sum_{i=1}^{n} |P_i - O_i| \qquad (9)$$

where *"P"* is the predicted value, *"O"* is the observed/true value, *"$\mu_{obs}$" and "$\mu_{pred}$"* the means of the observed and predicted values respectively, *"$\sigma^2_{obs}$"* and *"$\sigma^2_{pred}$"* are the associated variances, $\rho$ is the correlation between the observed and the predicted values.

Though these error metrics are widely used for assessing models, they cannot inform about the uncertainty related to the prediction. Therefore, we additionally considered the density distribution of the predicted versus actual SOC stock and the 90% confidence interval (CI). Further, the scattergram of the prediction interval coverage probability (PICP) was also considered (Vaysse and Lagacherie, 2017). The latter is the graphical representation of the proportion of time the actual values of SOC stock fall within a series of p-probability of prediction intervals (PI) limited by (1-p)/2 and (1+p)/2 quantiles.

The SOC stock maps were computed only for the models based on the remote sensing variables (RSV models) because of their availability as maps. A qualitative assessment of the spatial distribution of the humus layer, mineral soil and total SOC stock from the produced maps was carried out and compared to literature.

## 3. Results

### 3.1 Validation performance of global models over whole Sweden

The performance metrics of the cross and independent validation of the RFR models over Sweden are presented in Table 3. The internal accuracy statistics showed that modelling with all variables resulted generally in marginally lower RMSE and higher $R^2$ for all SOC stock. Modelling with allV reduced the cross-validation RMSE by 2%, 1% and 6% compared to SSV models and by 7.9%, 10%, 6% compared to RSV models respectively for the humus layer, mineral soil and total SOC stock. Though modelling with allV resulted in higher cross-validation $R^2$ compared to the remaining models, only 30%, 29% and 28% of the total variance were explained respectively for the total SOC stock, mineral soil and the humus layer SOC stock.

The independent validation showed similar trends as observed for the cross-validation. The Lin's correlation concordance coefficient (CCC) confirmed that the predictive performance of RFR for the different SOC stock were enhanced either by using only SSV or allV. The similarity between the RMSE values of both training and validation data shows that the global models over Sweden did no overfit. However, the explained variances are as lower as for the cross-validation varying from 15% to 27% for the SSV models, 10% to 18% for the RSV models and from 26% to 30% for the allV models. For both cross and independent validation, the RMSE increased with depth with the lowest values recorded for the humus layer.

### 3.2 Validation performance of local models versus global models

As observed for the global models, better accuracy were recorded for the local models based on allV and SSV which present in general lower RMSE as well as higher CCC and $R^2$ when compared to the local RSV models for both cross and independent validation (Table 4).



The cross-validation with the local models resulted in lower RMSE compared to the values recorded for the global models (Table 3) except for the southern Sweden models which recorded higher values no matter the category of variables. Local models with allV reduced the RMSE of cross-validation in relation to the global models (Table 3) by 18% for both North and Central Sweden  for the humus layer SOC stock, by 21% (North) and 20% (center) for the mineral soil SOC stock, and by 9% (North) and 24% (central) for the total SOC stock. The variances explained by the local models after cross validation vary from 15% to 32% for allV models, 15% to 25% for the SSV models and from 6% to 20% for the RSV models. In addition, the RMSE of the local models increased in general from the humus layer to the mineral soil for both cross and independent validation as previously observed for the global models no matter the validation type and category of factors.

The global models were also used to make prediction with the same independent validation set used for the local models. Though the local models outperformed the global models, the results were different based on the subareas and category of variables (Figure 2). However, the local SSV models were more consistent at outperforming the global SSV models compared to RSV and allV models when tested with an independent dataset. For the humus layer (Fig. 2A), the local models associated with all variables performed better than the remaining models except for northern Sweden where global model recorded the lowest RMSE. For the mineral and total soil layers (Fig. 2B-C), only local models showed better performance compared to global models with lowest RMSE. The best local model were mostly associated with all variables or site specific variables especially for central and southern Sweden. Only with the local model of mineral SOC stock for Northern Sweden that remote sensing variables gave a better accuracy as compared to other models.

The local and global models showed similar trend for the density distribution of actual versus predicted SOC stock (Fig. 3). For Fig.3 and Fig.4, only global and local models with the lowest RMSE were reported to avoid redundancies. All RFR models presented an underestimation of lower and higher values of SOC stock while an overestimation was observed for the values centred around the means. However, underestimation of high values was less pronounced with the global models over the entire Sweden and also with the predictions for the humus layer. The local model associated with the remotely sensed variables of the mineral soil SOC stock in northern Sweden also presented a pronounced overestimation of the lower values.

The PICP estimates seem to correspond quite well with the respective confidence level (Fig. 4) except for the humus and mineral SOC stock of southern Sweden. For southern Sweden, it appears that at higher level of confidence the corresponding PICP is higher for the humus layer and lower for the mineral SOC stock. Considering a 90% confidence interval, most of the validation observations (80% - 95%) were located within the prediction interval especially for models based on specific site variables or all variables (Supplementary information SI 1).

### 3.3 Variable importance

The global RFR models using only site factors shows that (Table 5) the latitude (Northing) was the most important variable influencing the distribution the humus layer and the total SOC stock though it ranked second for the mineral soil. A consistent negative but significant correlation was observed between the different SOC stocks and the latitude suggesting lower stock northwards no matter the depth.



The site specific variables took pre-eminence over remote sensing variables both at global and local scales when considering models using all variables (Table 5). The occurrences of soil moisture or soil type among the top two

most influential variables are higher compared to the remaining variables. For the humus layer SOC stock, the key covariate involved in the prediction was the soil moisture reported both for global and local models except in southern Sweden where it came as the third key variable. The most prominent covariate in predicting the mineral soil SOC stock with the global allV model was the soil type as also recorded for southern Sweden while remaining local models indicated Texture for northern and central Sweden. The global model revealed soil moisture and soil

type as the main variables affecting the prediction of the total SOC stock over Sweden. A similar trend is observed in northern Sweden while the remaining models recorded 40k as second key variable in addition to soil moisture and soil type for the central and southern Sweden respectively.

The cumulative contribution of each category of variables to model accuracy based on their contribution to the

MDA using all variables is presented in Fig.5. Topography variables greatly influence model accuracy in the northern part of Sweden contributing to about 30% - 40% of the model MDA especially for the humus layer and mineral soil SOC stock. This is further corroborated by a high correlation of these variables with the SOC stock in northern Sweden (Table 10). For the humus and mineral SOC stock, the importance of topography decreased from the north to the south of Sweden with the gamma-ray, site specific and climate variables gaining more

prominence (contributing together up to 60% of MDA) in central Sweden while site factors were the most influential variable with a share of 40% of MDA in southern Sweden (Fig4.). These categories of variables which ranked first in central and southern Sweden were also classified among the top three variables - site specific variables, climate and gamma-ray data for the global humus layer model.

As observed for the humus layer, topography was less prominent for central and southern Sweden both mineral soil and total SOC stock (Fig. 5). Site specific variables, climate and geochemical data which provided the highest contribution to MDA mineral soil for the global model over Sweden where also the most influential over central and Southern Sweden  contributing together up to 60% and 70% to the MDA. Gamma ray data seemed to play a key role in the distribution of the total SOC stock especially in southern Sweden together with the site specific

variables and climate. It is important to note that for the global model of the total SOC layer, the different category of variables contributed almost equally to the MDA with the gamma-ray and climate taking pre-eminence over the site specific variables. The forest variables had very low contributions as compared to the remaining (Fig.5) category of variables and they were mostly absent from the top 10 (Table 5) while those ranked have very low correlation with the different SOC stock.


### 3.4  Maps of SOC stock

Fig. 6 show the SOC stock maps from the RSV global and local models. Though the global RSV models generally outperformed the local RSV models (Table 4), their predictive maps follow generally the same pattern. Broadly, there is an increasing gradient of SOC stock from North to South for the humus layer, mineral soil and total SOC

stock. The local models tend to present lower values of SOC stock in northern and central Sweden for the humus layer while global model displays higher values over the whole country. For the mineral soil, there seems to be no distinct difference in the spatial prediction of SOC stock which resulted in similar pattern from the North to the



South for both local and global model maps. Since the total SOC stock is the sum between the humus layer and mineral SOC stock, its spatial distribution follows the same trend with lowest SOC recorded in northern and central

Sweden while higher stock are located in the south. No matter the type of SOC stock, the standard deviation tends to follow the same fashion, with higher uncertainties attached to the prediction of SOC stock in southern Sweden compared to the remaining areas independently of the type of models.

Figure 6: Maps of the spatial distribution of the humus layer, mineral soil and total SOC stock based on the RSV

models

## 4    Discussion

### 4.1  Prediction with global and local models

This study examined how global and local models differ in predicting the humus layer, mineral soil and total SOC stock in Sweden forests. The local models recorded lower RMSE at modelling stage with the cross-validation compared to the global models except for the Southern area. When prediction were carried out on the same validation set, local models including those of southern Sweden generally outperformed the global models. This suggests on the one hand that global models with higher sample size might not necessarily result in a more accurate

model compared to models built from a reduced dataset corresponding to a subarea of a bigger region. On the other hand, the particular case for southern Sweden suggests that though a global model might present a comparative advantage at modelling stage, they might not necessarily have a better predictive power when confronted with a new set of samples.

The findings of this study are in line with those of Somarathna et al. (2016) for predicting SOC content who also

found locally calibrated models to perform better than global models. However, the results of the present study differed from the latter in that, the comparative advantage was dependent of the category of variables used.

Findings (Figure 2) showed that local models which outperformed global models were either associated with all variables or site specific variables. For example, local models in central Sweden required all variables to

outperformed global models for the humus layer, mineral soil and total SOC stock. The same pattern was observed for Southern Sweden except for the mineral SOC stock for which the local model was associated with the SSV. The local model for the total SOC stock in Northern Sweden was also associated with SSV. The higher occurrences of SSV and allV with the best local models showed that modelling with RSV alone is not the optimal choice. On the one hand, forest SSV are more relevant for capturing local variability of the sampling plots than the other

variables which are remote sensing products. When both SSV and RSV are used as covariates, the locally specific information at plot scale are complemented by higher scale covariates which cover a larger range of the feature space resulting in model improvement especially for the humus layer.

In addition, using both site characteristics and remotely sensed products for predicting SOC stock generally

increased the variance explained with both cross-validation and independent validation methods for the humus layer, mineral soil and total SOC stock. However, despite the combination of these two category of covariates, the accuracy of the SOC stock prediction remain low for both the global models (maximum $R^2$ is 0.30) and local models (maximum $R^2$ is 0.33). There seems to be no study comparable in scope and methodology targeting the



prediction of SOC stock in forest soils. The closest is the digital mapping of SOC stock for the humus layer and

mineral stock using machine learning models such as RFR and the k-nearest neighbour (kNN) based on dataset
from the US national forest inventory (Cao et al., 2019). The authors also found lower fit between predicted and
observed SOC stock after the independent validation and reported an $R^2$ of 0.20 and 0.11 for the humus layer while
recording an $R^2$ of 0.33 and 0.28 for the mineral soil respectively for the RFR and kNN models. The second relevant
finding for comparison is the study carried out by Nussbaum et al. (2012) using the Swiss National Forest Inventory

data to map the mineral soil (0–30 cm) using a linear regression model. They also recorded an $R^2$ of 0.30 for the
forest mineral soil. Other studies conducted in temperate forests for predicting SOC stock, showed also poor
goodness of fit values with a cross validation $R^2$ of 0.22 (1 m depth) with the boosted regression trees (Ottoy et
al., 2017). For other soil properties, Mansuy et al. (2014) reported for some Canadian managed forest an $R^2$ of
0.04 and 0.05 for SOC content in the humus layer and mineral soil respectively with the kNN while Beguin et al.

(2017) recorded for the Canadian forest an $R^2$ of 0.05 for SOC content for the mineral soil with RFR model.

It is well established that some disparity occurs between observed and predicted values of DSM models. This is
due to different sources of errors leading to low explained variances (Nelson et al., 2011). Sources of errors might
be related to omitting key variables with greater explanatory power or conversely using not essential covariates

with very low explanatory power which only increase the prediction error variance. Not using the key covariates
in relation to SOC stock for forest ecosystem in the present study is less likely since variables considered in this
study well represent the surrogates for soil forming factors considered in the SCORPAN equation defined by
McBratney et al. (2003). In addition, dimension reduction with the removal of highly correlated variables and
exclusion of some others via recursive feature elimination participated in eliminating redundant and non-

informative covariates. On the other hand, correlations (min = 0, max = 0.28) between covariates and the different
SOC stock were found to be poor though significant for most of the predictors (Table 5, SI 2-4). This could be
expected because the data cover a wide range of different site conditions, soil types and parent materials.

Another source of errors could be inherent to the model with prediction accuracy varying with different type of

model. Many studies have already compared different machine learning models and concluded that RFR has
generally a strong predictive ability in different ecosystems (Cao et al., 2019;Forkuor et al., 2017;Wang et al.,
2018). Preliminary steps in the present study also tested extreme gradient boosting and the Cubist models (results
not shown) alongside the RFR with the latter displaying higher predictive capabilities. On the other hand, applying
geostatistical approaches (SI 5) for the humus layer, mineral soil and total SOC stock revealed very low spatial

autocorrelation for the different SOC stock and their regression residuals, suggesting that the structure of these
SOC data is having a shorter range than the sampling interval. For soil properties which vary over short distance
such as SOC stock, data driven models such as RFR might capture better the inherent variability of the data when
the data itself are a good representative of the phenomenon the SOC stock is subject to in the landscape, including
small scale variation. Beguin et al. (2017) recorded poor performance of different models including RFR for

predicting C:N because the sampling scheme failed to capture the distance variation (< 20 km) at which better
accuracy would have occurred. Model accuracy would likely improve if more samples covering the spatial
variability of each inventory plot were taken. The increase in RMSE with depth recorded for some models is
consistent with previous studies where prediction of lower soil layers resulted in lower accuracy (Henderson et al.,



2005;Yam et al., 2019). This may be due to a higher sensitivity of the humus layer which is directly exposed to
the influence of environment variables.

The estimates of SOC stocks are slightly biased towards the extreme values with an underestimation of the lowest
and highest values for both local and global models (Fig. 3). This tends to confirm earlier findings which reported
issues related to the underestimation or overestimation of small / higher values from the RFR model (Čeh et al.,
2018;Hu et al., 2020;Horning, 2010). On the one hand, this seems to be typical for regression models with RFR
because predictions are the average values of all of the trees.  On the other hand, this may also be related to an
under representation of the lower and higher values compared to those centred around the mean in the training
dataset. However, though underestimation of the lowest and highest values could be recorded for all models, the
90% PICP shows in general that the 90% prediction interval covers adequately the observed values of the humus,
mineral and total SOC stock layers (Fig. 4). This is an indication that the prediction intervals are accurate
representative of the prediction uncertainties for each of these SOC stocks for both local and global models.
However, for southern Sweden, the PICP presented higher values for the humus layer and lower values for the
mineral SOC stock with increasing level of confidence, suggesting a higher level of uncertainties in the predictions.
This could be attributed to southern Sweden being characterized by a longer management history and more
intensive forestry compared to northern Sweden (Angelstam, 1997), leading to a diversity of forest management
patterns with potential feedback on SOC stock distribution.

### 4.2 Variable importance and modelling accuracy

SOC stocks in forest soils are the product of the dynamic equilibrium between input flux of plant-derived materials
and output flux of carbon as a result of decomposition. Classical soil forming factors - climate, organisms
(vegetation, fauna, human activities), topography parent material and time are known to govern the amount and
distribution of SOC stock. Though covariates used as proxy for these soil forming factors were considered
separately for the sake of analysis in this study, they are actually involved in dynamic interactions leading to
complex soil processes in the landscape.


With the global RFR models using only site specific variables, the latitude (northing) was the main variable driving
the distribution of the SOC stock with a negative correlation suggesting lower stock northwards (Table 5). The
latitudinal gradient (Millberg et al., 2015) in Sweden results also in climatic gradient (Jungqvist et al., 2014) which
in turn interact with topography (Johansson and Chen, 2003) to determine the heterogeneity in net primary
production in relation to the spatial variability of precipitation and temperature. Even at regional level, the latitude
was still critical and was mostly present among the top ten variables being selected by the local RFR models using
all variables (Table 5). However, climate and topographical variables were overshadowed consistently by SSV
when all the variables were used for modelling both at a national and regional scale. Though precipitation regulates
net primary productivity (NPP) and temperature controls microbial decomposition of organic matter, their local
variability is generally small (Wiesmeier et al., 2019). This makes them less relevant in contrast to SSV taken at
plot level which describe more closely factors controlling the decomposition and stabilization of organic matters.



Among the site characteristics the soil moisture was the key site factor affecting the humus layer SOC stock especially in the northern and central Sweden while vegetation type was ranked first in southern part of Sweden (Table 5). The box plots of these two variables showed that they have clearly different distribution of SOC stock in the humus layer, although some of the inter-quantile ranges overlap (SI 6). As observed for the humus layer, soil moisture was the most important variable associated with total SOC stock along with the soil type. For a sequence from dry to moist soils, there was an increase of SOC stock in the humus layer as well as for the mineral and total stock (box-plot of soil moisture SI 6-8). This might be explained by higher productivity in litter supply as water is more available in the tree root zone of fresh and moist sites. On the other hand, these latter soils are subject to a longer period of saturation (reducing conditions) slowing down decomposition. The impact of soil moisture could also be noted when considering the partial dependence plot of the RFR global model of the humus layer showing the interaction between the soil moisture class and vegetation type (SI 9A). Each vegetation type tends consistently towards higher values of SOC stock for moist sites compared to dry and fresh sites.

Generally, soil type and texture were ranked by the allV global models as the top variables influencing the SOC stock in the mineral soil (Table 5). The link between these two variables could be related to the soil moisture content of their classes. On the one hand, soil types (Histosols, Gleysols) with fine texture (fine silt, clay) having high moisture content are more subject to reducing conditions with higher SOC stock compared to soils (Leptosols, Arenosols) with coarse (stone, boulder, coarse sand) texture.

The addition of the remote sensing products to the site specific variables resulted in limited improvement for both global and local models (Table 3-4). This suggests that their level of distinct complementarity in the feature space is low as the remote sensing products might be carrying redundant information with the site specific variables in relation to the humus layer, mineral soil and total stock. For example, the prominence of site specific variables over topographical variables (Table 5, allV) might be due to the fact they are indirectly incorporated into the definition of the site specific variables. For this study, wetness index, distance to groundwater and depth to water are indexes to characterize soil moisture while gamma ray data describe parent material. Similar observation was shared by Wiesmeier et al. (2011) who also recorded land use and soil type as key variables affecting SOC stock while topographic variables contributed very weakly to model accuracy using Random Forest. However, though lowly ranked among all variables (Table 5, allV), the cumulated variable importance analysis showed that topographical variables stood out in contributing to model accuracy in northern Sweden (Fig. 5) but were less relevant southward. Obviously, higher elevation and derivatives in northern Sweden explains such influence on the SOC stock.

Next to site characteristics and climate, the cumulative gamma-ray data was more consistent in contributing to model accuracy of the total SOC stock compared to geochemical data with the individual ranking further revealing higher occurrence of radioactive K both at global and regional level (Table 5, Fig. 5). This suggests that K-bearing minerals of the parent material has greater explanatory power over total SOC stock than U and Th, the nature of which might require further studies. Malone et al. (2009) also recorded gamma K as the key covariate for mapping SOC stock in an agriculture dominated land use in Australia.



The geochemical data revealed to be key variables in the distribution of SOC stock in southern Sweden especially for the humus layer (Na) and mineral stock (Ca, Na) though of much lower magnitude compared to site specific

variables. However, base cations seemed not to primarily affect SOC distribution but rather environmental variables that regulate their dynamics. The low ranking of forest parameters may be related to (1) their low correlations with the SOC stock data and (2) the fact that the data set cut across different data forest types without any specific stratification which could have created a homogeneous strata for modelling. However, the focus of the present study was not on a specific forest type which could have reduced further the training dataset while

machine learning models require high data samples to learn pattern and accurately predict target values on independent dataset.

### 4.3 Spatial prediction of SOC stock

The maps of the humus layer, mineral soil and total stock presents a pattern of increasing accumulation of SOC

stock from south to north with the highest uncertainties in the southern part of Sweden no matter the predicting models (Fig. 6). In general, it is expected that the global latitudinal trend will result in increasing stock in higher latitude which correspond to colder and humid regions. Possible explanations are associated with slower microbial decomposition rates while other studies suggested non conducive soil conditions such as water logging, low pH values, high aluminium concentration as the main constraints (Dieleman et al., 2013;Hobbie et al., 2000;Wiesmeier

et al., 2019).

The contrary configuration observed in the maps with decreasing South-North distribution in SOC stock for humus layer and mineral (SI 10) are consistent with findings from different studies (Kleja et al., 2008;Fröberg et al., 2011;Hyvonen et al., 2008). These studies advocate that the high SOC stocks in the south could be related to a

higher deposition of nitrogen (N) compared to the center and northern Sweden. It has been suggested that N deposition results in both increasing litter inputs and increasing mean residence time. Also, high concentrations of inorganic N inhibit the activities of lignin-degrading phenol oxidase released by microorganisms (Zak, 2017;Carreiro et al., 2000). However, warmer climate makes trees grow faster along with a higher litter input in the south than in the north. With co-occurring north-south gradient of temperature (lower), pH (higher), soil carbon

(lower) (Iwald, 2016;Framstad, 2013), N deposition might have contributed to strengthen the North-South SOC stock gradient. As southern Sweden (Figure 6) recorded higher range in SOC stocks, the associated average variation around the mean was also larger.

### 4.4 Implication and limitations of the study

DSM relies on existing maps for building regression models and ultimately prediction for mapping. The quality and accuracy of predictions depend as discussed earlier on choosing the most relevant covariates in relation to the target to be predicted. The present study revealed that variables which were available as maps did contribute to the MDA, but site characteristics were more prominent in relation to the SOC stocks in Sweden. Consequently, having high resolution maps of these site characteristics would increase the accuracy of both models and resulting maps.

In situations where additional site data are available, a preliminary study such as carried out in this study would help identify the most relevant site characteristics which are worth considering for mapping in the purpose of increasing the accuracy of the target variable to which they correlate most. For the present study, it appears from



the variable importance that having a map format of the soil moisture class, vegetation type, soil type and soil texture has potential in improving the output maps and reduce the prediction uncertainties.


The present study compared a local and a global modelling approach for DSM. To the question which approach to use while confronted with a big area, our research showed that it is dependent on the type of covariates available. In general, building local models for subareas of the study region will require having covariates which correlate most with the sampling sites thereby offering a better description at a smaller scale. In this study, the site

characteristics were a better representative of the sampling locations and their local models generally performed better than global models. In situation where such site characteristics data are not available, it would be preferable to use a global model for the whole area.

However, machine learning models such as RFR are data driven, and therefore results will vary according to the specificity of a given area. Therefore, there is no silver bullet in the approach to use for any specific area and it

will be necessary to draw conclusions from the modelling results. However, it is very likely that the combination of site characteristic with remote sensing data would result in higher accuracy at both local and global scale than using remote sensing data alone.

The maps produced with the RSV global and local models for the humus layer, mineral soil and total SOC stock

present accurately the distribution of SOC stock observed for Sweden in many studies. Given that the underlying models were not the most accurate in the present study, such maps should be treated with caution for decision especially with the associated high standard deviation. However, they could serve as a high resolution indicator of the spatial trend in SOC stock at different depth for the landscape of the Swedish forest. In addition, the use of a DSM approach in the present study allows: flexibility in future improvement upon acquisition of new covariates

or data point, repeatability in modelling with the application of the same modelling principles using open access software (e.g. R) and capitalizing on multi-source information (topography, site characteristics, forest data, gamma radiometry and geochemical data). Therefore, smaller counties could evaluate this approach on their own dataset for mapping other soil properties (pH, texture, Fe, Al etc.) and SOC stock for local applications.

**Conclusion**

This study has shown that:
- Local models have a comparative advantage over global models when using either site characteristics alone or the combination of the latter with remotely sensed variables for modelling.
- Using remotely sensed variables with soil inventory data indicates that such covariates have limited predictive compared to site specific variables.
- The most important covariates that influence the humus layer, mineral soil and total SOC stock were related to the site characteristic covariates and include the soil moisture class, vegetation type, soil type and soil texture. These variables could potentially improve the spatial accuracy of the final SOC stock
maps when available in a map format as covariates.



**Code**


As a R file (pdf) in the supplement materials.

**Data availability**

The data used in this study is available upon reasonable request sent to Johan Stendahl (Johan.Stendahl@slu.se). The high-resolution digital elevation models (DEM) should be requested by contacting the Swedish national mapping agency (Lantmäteriet, https://www.lantmateriet.se). The climate data used (MAT, MAP) can be downloaded from the source: WorldClim, (Fick and Hijmans, 2017). The geochemical and gamma-ray data can be obtained from the Geological Survey of Sweden (SGU, https://www.sgu.se). Requests for forest maps should

be directed to the Swedish Forest Agency (Skogsstyrelsen, https://www.skogsstyrelsen.se). The dataset related to the historical map series can be freely downloaded from the figshare repository using the following link: https://doi.org/10.17045/sthlmuni.4649854.

**Author contribution:**

Conceptualization of the study for this manuscript was done by KOLH as well as the data curation, formal analysis, and methodology with feedback from all authors. KOLH also wrote the initial draft and all authors were involved in the review and editing of the manuscript.

**Competing interests:**

All authors declare that they have no conflict of interest.

**Acknowledgement**

This work was supported by the Swedish Research Council Formas [Grant agreement: FR-2017/0006].

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





Table 1: List of explanatory variables for predicting SOC stock

| Type | Variables | Abbreviation |
|---|---|---|
| Topography | Elevation (m) | DEM |
| | Slope (%) | Slope |
| | cos(Aspect) | cosAsp |
| | sin(Aspect) | sinAsp |
| | Plan curvature (°m−1) | PLCur |
| | Profile curvature (°m−1) | PRCurv |
| | Terrain ruggedness index | TRI |
| | Saga wetness index | SWI |
| | Distance to streams (mm) | strDist |
| | Depth to water (m) | DTW |
| | Distance to Groundwater (mm) | DTG |
| Climate | Temperature (°C) | Temp |
| | Precipitation (mm) | Prep |
| Geochemical data | Ca, Mg, K, | GeoCa, GeoMg, GeoK, |
| | Na, Mn (ppm) | GeoNa, GeoMn |
| Gamma-ray data | 40K (ppm), 232Th (ppm), 238U (%) | GamK, GamTh, GamU |
| Forest | Stand age (years) | For.Age |
| | Biomass (kg) | For.Biom |
| | Height (m) | For.Height |
| | Stem volume (m$^3$) | For.Vol |
| Historical land use map* | Former Cropland | histCL |
| | Former Grassland | histGL |
| Site characteristics | Soil types | SoilTyp |
| *Levels* | *1 Histosol; 2 Leptosol; 3 Gleysol; 4 Podzol; 5 Umbrisol; 7 Arenosol; 6 Cambisol; 8 Regosol; 9 Unclassified* | |
| | Soil moisture class | SoilMst |
| *Levels* | *1 Dry; 2 Fresh; 3 Fresh/moist; 4 Moist; 5 Wet* | |
| | Soil texture class | Texture |
| *Levels* | *0 Boulders in the profile; 1 Stone/Boulder/Bedrock; 2 Gravel/Gravely till; 3 Coarse sand/Sandy till; 4 Sand/Sandy silty till;5 Fine sand/Silty sandy till; 6 Coarse silt/Coarse silty till; 7 Fine silt/Fine silty till; 8 Clay/Clayish till/Gyttja; 9 Peat* | |
| | Parent material | ParMat |
| *Levels* | *1 Well sorted sediments; 2 Poorly sorted sediments; 3 Till; 4 Bedrock; 5 Peat* | |
| | Vegetation type | VegTyp |
| *Levels* | *1 tall herbs without shrubs; 2 tall herbs with shrubs/bilberry; 3 tall herb with shrubs /vitis idea ; 4 low herbs without shrubs; 5 low herbs with shrubs /bilberry; 6 low herbs with shrubs /vitis idea ; 7 without field layer; 8 broad leaved grass; 9 narrow leaved grass; 10 tall sedge; 11 low sedge; 12 horse tail type; 13 bilberry type; 14 vitis idaea/whortleberry, marsh rosemary;15 crowberry/heather type; 16 poor shrubs type* | |
| *Coordinates* | *northern* | *NorthC* |
| | *Eastern* | *EastC* |

\*: used only for the southern part of Sweden





Table 2: Descriptive statistics for the training and validation datasets

| | | Training | | | | | | | | Validation | | | | | | | |
|---|---|---|---|---|---|---|---|---|---|---|---|---|---|---|---|---|---|
| | | n | min | max | median | mean | sd | cv | skewness | n | min | max | median | mean | sd | cv | skewness |
| Humus layer (t C/ha) | North | 1008 | 0 | 246.00 | 18.10 | 23.57 | 21.78 | 0.92 | 4.25 | 252 | 0 | 128.00 | 18.05 | 23.05 | 18.28 | 0.79 | 2.57 |
| | Center | 1708 | 0 | 299.52 | 18.42 | 23.87 | 21.49 | 0.90 | 3.61 | 424 | 0 | 143.77 | 18.42 | 23.54 | 19.31 | 0.82 | 2.33 |
| | South | 1763 | 0 | 418.80 | 23.05 | 30.07 | 34.79 | 1.16 | 3.36 | 440 | 0 | 418.80 | 23.06 | 30.59 | 39.75 | 1.30 | 4.42 |
| | All | 4479 | 0 | 418.80 | 19.52 | 26.24 | 27.72 | 1.06 | 3.88 | 1116 | 0 | 418.80 | 19.63 | 26.21 | 29.18 | 1.11 | 5.07 |
| Mineral (t C/ha) | North | 478 | 2.36 | 305.59 | 41.46 | 46.85 | 25.96 | 0.55 | 3.44 | 116 | 16.21 | 136.47 | 41.34 | 47.73 | 23.07 | 0.48 | 116 |
| | Center | 785 | 0 | 224.24 | 48.31 | 53.28 | 26.46 | 0.50 | 1.60 | 196 | 0 | 143.14 | 48.27 | 52.49 | 23.51 | 0.45 | 196 |
| | South | 875 | 0 | 386.70 | 62.43 | 68.32 | 40.49 | 0.59 | 1.93 | 216 | 0 | 206.00 | 63.09 | 68.34 | 37.91 | 0.55 | 216 |
| | All | 2138 | 0 | 386.70 | 51.44 | 58.22 | 33.73 | 0.58 | 2.15 | 528 | 0 | 206.00 | 51.81 | 57.93 | 31.39 | 0.54 | 528 |
| Total (t C/ha) | North | 478 | 16.11 | 360.18 | 62.93 | 72.46 | 39.62 | 0.55 | 2.94 | 116 | 20.02 | 331.56 | 63.00 | 75.12 | 45.76 | 0.61 | 2.90 |
| | Center | 784 | 12.88 | 229.87 | 71.80 | 77.33 | 31.34 | 0.41 | 1.32 | 196 | 15.92 | 254.69 | 71.69 | 78.67 | 37.20 | 0.47 | 1.78 |
| | South | 870 | 0 | 487.37 | 89.19 | 99.34 | 50.63 | 0.51 | 2.37 | 216 | 16.78 | 357.23 | 88.99 | 97.61 | 44.47 | 0.46 | 2.01 |
| | All | 2132 | 0 | 487.37 | 76.26 | 85.22 | 43.57 | 0.51 | 2.47 | 528 | 15.92 | 357.23 | 76.46 | 85.64 | 43.32 | 0.51 | 2.11 |


Table 3: Cross-validation and independent validation of the global random forest models

| | | Cross-Validation | | Independant Validation | | | | |
|---|---|---|---|---|---|---|---|---|
| | | RMSE (t C/ha) | $R^2$ | RMSE (t C/ha) | MAE (t C/ha) | Bias (t C/ha) | CCC | $R^2$ |
| Site specific variables (SSV) | Humus layer | 23.9 (±2.74) | 0.26 (±0.06) | 20.8 | 13.7 | 0.45 | 0.43 | 0.26 |
| | Mineral soil | 28.6 (±3.34) | 0.27 (±0.06) | 27.9 | 19.8 | 0.45 | 0.43 | 0.27 |
| | Total | 38.9 (±4.28) | 0.21 (±0.06) | 38.9 | 27.8 | 1.81 | 0.34 | 0.15 |
| Remote sensing variables (RSV) | Humus layer | 25.4 (±3.20) | 0.15 (±0.04) | 22.1 | 15.2 | 1.27 | 0.28 | 0.17 |
| | Mineral soil | 31.5 (±3.33) | 0.13 (±0.05) | 30.7 | 21.8 | 0.95 | 0.23 | 0.10 |
| | Total | 38.9 (±4.40) | 0.20 (±0.05) | 38.4 | 27.7 | 0.87 | 0.32 | 0.18 |
| All variables (allV) | Humus layer | 23.4 (±2.90) | 0.28 (±0.06) | 20.3 | 13.7 | 1.35 | 0.47 | 0.30 |
| | Mineral soil | 28.3 (±3.46) | 0.29 (±0.07) | 28.2 | 20.2 | 1.17 | 0.41 | 0.26 |
| | Total | 36.5 (±4.35) | 0.30 (±0.06) | 35.5 | 25.6 | 1.42 | 0.41 | 0.27 |

RMSE: root mean square error, MAE: mean absolute error, CCC: Lin's correlation concordance coefficient









Table 4: Cross-validation and independent validation of the local

| | | | Cross-Validation | | Independant Validation | | | | |
| --- | --- | --- | --- | --- | --- | --- | --- | --- | --- |
| | | | RMSE (t C/ha) | R² | RMSE (t C/ha) | MAE (t C/ha) | Bias (t C/ha) | CCC | R² |
| Site specific variables | Humus layer | North | 19.4 (±4.33) | 0.19 (±0.11) | 18.7 | 11.8 | -0.85 | 0.35 | 0.22 |
| | | center | 19.3 (±3.84) | 0.19 (±0.07) | 18.1 | 12.1 | 0.47 | 0.30 | 0.14 |
| | | South | 30.1 (±4.62) | 0.25 (±0.09) | 25.6 | 17.8 | 1.45 | 0.46 | 0.26 |
| | Mineral soil | North | 23.0 (±7.25) | 0.12 (±0.07) | 23.5 | 15.6 | -0.25 | 0.28 | 0.17 |
| | | center | 23.1 (±3.37) | 0.13 (±0.09) | 27.0 | 19.0 | -2.11 | 0.28 | 0.20 |
| | | South | 35.5 (±5.48) | 0.24 (±0.07) | 31.8 | 23.0 | 2.55 | 0.42 | 0.24 |
| | Total | North | 34.7 (±8.88) | 0.22 (±0.14) | 25.1 | 19.3 | 2.01 | 0.50 | 0.33 |
| | | center | 29.1 (±2.85) | 0.14 (±0.07) | 33.7 | 24.1 | -1.87 | 0.22 | 0.10 |
| | | South | 47.7 (±7.80) | 0.15 (±0.08) | 47.4 | 34.3 | 3.79 | 0.23 | 0.07 |
| Remote sensing variables | Humus layer | North | 19.4 (±4.84) | 0.18 (±0.09) | 19.7 | 13.1 | -0.83 | 0.28 | 0.14 |
| | | center | 20.4 (±4.33) | 0.08 (±0.04) | 18.0 | 12.5 | 1.29 | 0.19 | 0.12 |
| | | South | 31.3 (±5.49) | 0.18 (±0.08) | 26.5 | 18.9 | 2.22 | 0.32 | 0.19 |
| | Mineral soil | North | 24.2 (±7.55) | 0.08 (±0.06) | 21.2 | 15.9 | 0.39 | 0.22 | 0.13 |
| | | center | 24.1 (±3.22) | 0.05 (±0.04) | 28.7 | 20.0 | -1.56 | 0.12 | 0.09 |
| | | South | 38.6 (±5.26) | 0.10 (±0.07) | 36.1 | 26.6 | 4.33 | 0.14 | 0.05 |
| | Total | North | 35.2 (±8.51) | 0.20 (±0.10) | 29.6 | 22.8 | 4.83 | 0.37 | 0.16 |
| | | center | 28.9 (±3.07) | 0.16 (±0.08) | 35.1 | 25.9 | -0.08 | 0.11 | 0.03 |
| | | South | 47.2 (±7.21) | 0.12 (±0.07) | 44.6 | 32.9 | 1.48 | 0.19 | 0.10 |
| All variables | Humus layer | North | 19.0 (±4.67) | 0.22 (±0.08) | 18.8 | 12.4 | -0.44 | 0.33 | 0.21 |
| | | center | 19.0 (±4.05) | 0.20 (±0.07) | 16.9 | 11.5 | 1.15 | 0.34 | 0.23 |
| | | South | 28.5 (±5.15) | 0.32 (±0.08) | 23.6 | 16.7 | 2.31 | 0.52 | 0.36 |
| | Mineral soil | North | 22.3 (±6.51) | 0.19 (±0.07) | 24.5 | 18.1 | 4.00 | 0.25 | 0.12 |
| | | center | 22.6 (±2.82) | 0.17 (±0.09) | 26.6 | 18.5 | -1.76 | 0.28 | 0.25 |
| | | South | 35.3 (±5.42) | 0.25 (±0.09) | 32.8 | 24.1 | 3.59 | 0.37 | 0.20 |
| | Total | North | 33.2 (±8.29) | 0.28 (±0.13) | 25.2 | 19.7 | 2.96 | 0.48 | 0.33 |
| | | center | 27.7 (±2.90) | 0.23 (±0.06) | 31.9 | 23.4 | 0.23 | 0.28 | 0.20 |
| | | South | 44.9 (±7.29) | 0.22 (±0.09) | 42.9 | 31.7 | 2.64 | 0.29 | 0.17 |

RMSE: root mean square error, MAE: mean absolute error, CCC: Lin's correlation concordance coefficient







Table 5: Top 10 variables for the global and local random models for the humus layer, mineral soil and total SOC stock with the Pearson´s coefficient of correlation (values in bracket, *: p $\leq$ 0.05, p $\leq$ 0.05;**: p $\leq$ 0.01;***: p $\leq$ 0.001) for the litter, soil and total Cstock

| | | | n | Most important variables[1] |
|---|---|---|---|---|
| Site specific variables (N = 7) | Global | Humus layer | 7 | Northing (-0.12***), Soil moisture, Easting (-0.09***), Vegetation type, Soil type, Parent materiel, Texture |
| | | Mineral soil | 7 | Easting (-0.13***), Northing (-0.27***), Soil type, Vegetation type, Texture, Parent materiel, Soil moisture |
| | | Total | 7 | Northing (-0.28***), Soil moisture, Soil type, Easting (-0.15***), Texture, Vegetation type, Parent materiel |
| | North | Humus layer | 7 | Soil moisture, Soil type, Vegetation type, Northing (-0.09**), Easting (0.06), Parent materiel, Texture |
| | | Mineral soil | 7 | Easting (-0.13**), Vegetation type, Texture, Parent materiel , Northing (-0.09*), Soil moisture, Soil type |
| | | Total | 7 | Soil moisture, Soil type, Vegetation type, Texture, Parent materiel, Easting (0.00), Northing (-0.11*) |
| | Center | Humus layer | 4 | Soil moisture, Northing (-0.08***), Easting (-0.02), Vegetation type |
| | | Mineral soil | 4 | Parent material, Texture, Northing (-0.05), Soil type, Soil moisture, Easting (0.00) |
| | | Total | 7 | Soil moisture, Northing (-0.13***), Easting (0.03), Parent materiel, Texture, Vegetation type, Soil type |
| | South | Humus layer | 4 | Vegetation type, Soil moisture, Soil type, Easting (-0.15***) |
| | | Mineral soil | 7 | Soil type, Easting (0.02), Northing (-0.08*),Vegetation type, Parent materiel, Texture, Soil moisture |
| | | Total | 4 | Soil type, Easting (-0.10**), Soil moisture, Northing (-0.13***) |
| Remote sensing variables (N = 26) | Global | Humus layer | 20 | Mn (-0.08***), Precipitation (0.16***), 40K (-0.20***), 232Th (-0.15***), Na (0.03), Terrain ruggedness (-0.11***),     K (-0.07***), Distance to groundwater (-0.14***), 238U (-0.10***), sinAsp (-0.06***) |
| | | Mineral soil | 20 | Temperature (0.28***), Precipitation (0.18***), Mn (0.05***), Elevation (-0.16***), Terrain ruggedness (-0.12***), Na (-0.05), Ca (0.22***), 40K (-0.13***), Wetness Index (0.11***), K (0.01) |
| | | Total | 21 | Temperature (0.28***), K ()Distance to groundwater (-0.17***), Precipitation (0.21***), 40K (-0.24***), 232Th (0.16***), Na (-0.01), K (-0.04), Mn (-0.02), 238U (-0.11***), Elevation (-0.18***) |
| | North | Humus layer | 8 | 40K (-0.23***), Distance to groundwater (-0.18***), Elevation (-0.15***), Ca (-0.06), Temperature (0.16***), Mn (-0.10**), Na (-0.06), K (0.00) |
| | | Mineral soil | 8 | 40K (-0.25***), Wetness index (-0.01), Ca (-0.05), Na (-0.09), Temperature (0.01), Precipitation (0.16***), K (0.04), Aspect (0.03), Stand Age (0.02), Elevation (0.13*) |
| | | Total | 8 | Depth to water (-0.22***), 40K (-0.30***), K (0.04) ,Temperature (0.10*), Precipitation (0.16***), Elevation (-0.06), Mn (-0.07), Distance to streams (-0.15***) |
| | Center | Humus layer | 16 | 238U (-0.05), 232Th (-0.10***), Aspect (0.04), 40K (-0.16***), Terrain ruggedness (-0.11***), Elevation (-0.06), Precipitation (0.08**), Distance to groundwater (-0.16***), sinAsp (-0.06*), Profile curvature (-0.04) |
| | | Mineral soil | 19 | 40K (-0.11*), 232Th (-0.07*), Mn (0.04), Elevation (-0.03), Wetness index (0.09*), Stand age (0.06), 238U (-0.02), sinus Aspect (-0.01), Height (0.07), Precipitation (0.08***) |
| | | Total | 16 | 40K (-0.21***), Depth to water (-0.11**), 232Th (-0.12***), Mn (-0.02), Elevation (-0.08*), sinAsp (-0.06), Na (0.02), Precipitation (0.13***), Terrain ruggedness (-0.14***), Aspect (0.03) |
| (N = 28) | South | Humus layer | 16 | 40K (-0.24***), Precipitation (0.18***), Mn (-0.11***), Na (0.13***), Distance to groundwater (-0.12***), K (0.11***), Stem Volume (0.05*), Slope (-0.09***), Stand age (0.05*), 238U (-0.16***) |
| | | Mineral soil | 20 | Temperature (0.18***), Precipitation (0.02), Stand age (-0.07*), Distance to groundwater  (-0.16***), Na (0.05), Elevation (-0.11**), Ca (0.24***), 232Th (-0.09**), Slope (-0.09**), 40K (-0.09*) |
| | | Total | 21 | Precipitation (0.14***), Distance to groundwater (-0.17***), Elevation (0.00), Slope (-0.11**), Temperature (0.12***), 238U (-0.20***), Height (-0.11**), Ca (0.12***), K (0.04), 40K (-0.22***) |
| All variables (N = 33) | Global | Humus layer | 28 | Soil moisture, Vegetation type, Northing, Easting, Precipitation, Profile curvature (-0.04**), Temperature (0.12***), 232Th, 40K, Mn |
| | | Mineral soil | 28 | Soil type, Parent material, Texture, Temperature, Vegetation type, Easting, Northing, Elevation, Mn, Na |
| | | Total | 16 | Soil moisture, Soil type, Precipitation, 40K, Elevation, Na, Northing, Distance to groundwater, Mn, K |
| | North | Humus layer | 28 | Soil moisture, Distance to groundwater, Mn, Elevation, Temperature, Ca, K, Northing, 40K, Na |
| | | Mineral soil | 16 | Texture, Wetness index, precipitation, K, Distance to groundwater, 238U (-0.13),Vegetation type, 40K, 232Th (-0.13**), Elevation |
| | | Total | 8 | Soil type, Soil moisture, Depth to water, Vegetation type, Texture, 40K, K, Precipitation |
| | Center | Humus layer | 26 | Soil moisture, 232Th, 40K, Northing, 238U, Easting, Elevation, Profile curvature, Precipitation, Ca  (-0.03) |
| | | Mineral soil | 26 | Texture, Parent materiel, Precipitation, Northing, Mn, Elevation, Soil type, 40K, Ca (0.00), Easting |
| | | Total | 26 | Soil moisture, 40K, Northing, Parent materiel, Texture, Elevation, 232Th, Depth to water, Precipitation, Na |
| (N = 35) | South | Humus layer | 16 | Vegetation type, Soil type, Soil moisture, Easting, Na, 40K, Precipitation, Temperature (-0.03), Stem Volume, K |
| | | Mineral soil | 29 | Soil type, Parent materiel, Texture, Vegetation type, Easting, Northing, Precipitation, Ca, Na, Temperature |
| | | Total | 30 | Soil type, 40K, Northing, Soil moisture, Precipitation, Easting, 238U, Na (0.14***), Texture, Wetness index (0.03) |

[1]Site specific variables have no correlation values since they are categorical variables. Pearson´s coefficient of correlation are provided at first occurrence for a specific category of variables and SOC type.  N : Total number of covariates, n = number of covariates after feature selection.






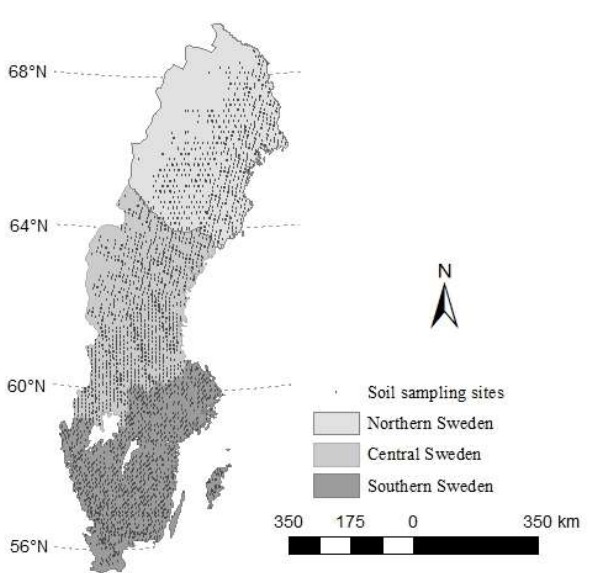


Figure 1: Sites from the Swedish Forest Soil Inventory for northern, central and southern Sweden







Figure 2: Local and global models ranked by decreasing RMSE per subareas and category of variables along with corresponding R². A: litter layer, B: mineral soil layer, C: total soil layer, SSV: site specific variables, RSV: remote sensing variables, allV: all variables

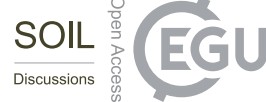

Figure 3: Density plots of the actual versus predicted humus layer, mineral soil and total SOC stock from the local and global Random Forest models (with lowest root mean square errors), line: average values, SSV: site specific variables, RSV: remote sensing variables, allV: all variables; lines: mean of SOC stock



Figure 4: Prediction interval coverage probability of the local and global random models for the humus layer

(A), mineral soil and total SOC stock. SSV: site specific





Figure 5: Variable importance of the main category of variables for local and global random models for the humus layer, mineral soil and total SOC stock



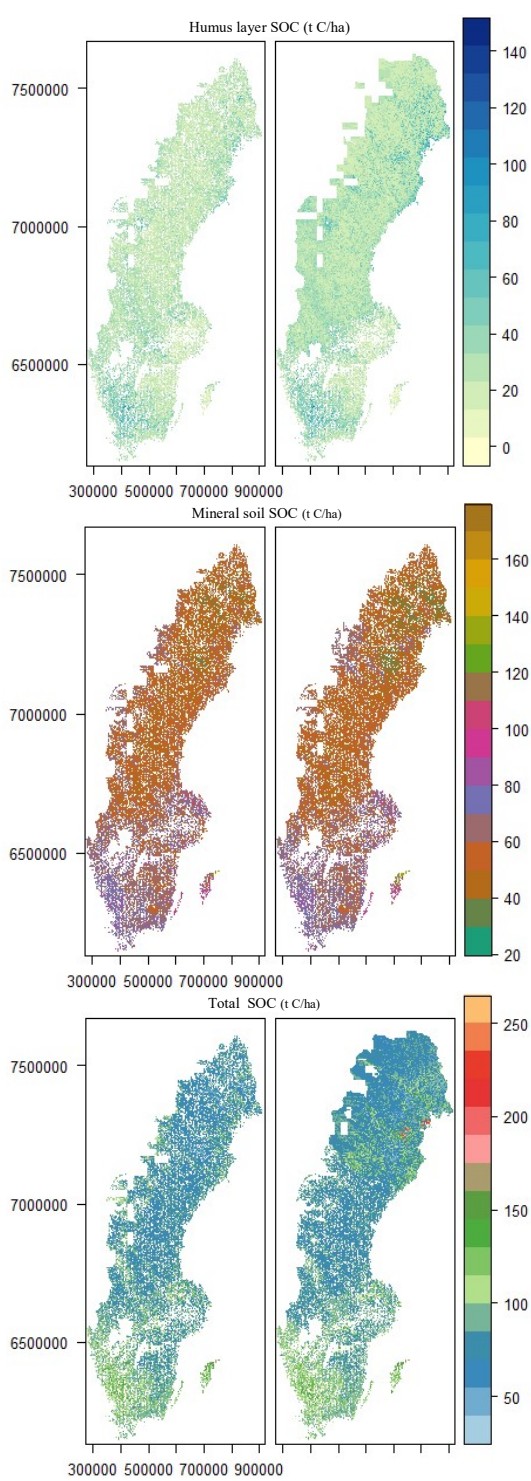

Figure 6: Maps of the spatial distribution of the humus layer, mineral soil and total SOC stock based on the remote sensing variables