# Peer review of "Predicting the spatial distribution of soil organic carbon stock"

_SOIL, 2020_

## Referee Comment (RC1) · Anonymous Referee #1 · 13 Jan 2021

The manuscript attempted to compare the model performance of global and local models in predicting humus layer, mineral soil and total SOC stock and to identify the controlling factors for SOC stock prediction. Besides, this study also investigated the effect different combinations of data from site characteristics and remotely sensed variables on model performance. The results from independent dataset indicated that the local models generally had better model performance than the global models. The only use of remotely sensed variables had limited predictive ability while site characteristics had better explanatory strength in estimating SOC stocks. The authors suggest that further

work can focus on mapping these influential site covariates. The manuscript is overall well-written with clear objectives and reasonable methodology. However, two major limitation of this manuscript are: (1) in comparing global and local models, I can tell from Table 4 and Table 5 that some local models had higher R2 than that of global models while the remaining local ones had lower R2. Therefore, the conclusion that local models have a comparative advantage over global models is not convincing for me. Again, it is not fair to compare the performance indicators as mentioned in this study for local and global models. Instead, for the global model, authors should also calculate indicators for three regions separately to make them comparable for these in local models; (2) As mentioned by authors, the site characteristics are only available at the visited plots, therefore the digital maps can not be produced by the models built with site characteristics, which certainly limits the usage of site characteristics. I am afraid that the importance of these site characteristics in this study have been overlooked as the observed site characteristics are directly used in independent validation which certainly ignores the inaccuracy of these site characteristics if they are mapped by certain algorithms. I mean, the site characteristics used in a fair independent validation should come from the predicted maps of these relevant site characteristics, not from observed data. Therefore, I suggest a major revision before it can be published. Specific comments are listed below: Line 49: which are nonspatial environmental covariates? If they are not spatial, how they can be used in DSM? Line 58: Authors miss at least two papers on comparing local and global models in DSM. Piikki, K., & Söderström, M. (2019). Digital soil mapping of arable land in Sweden–Validation of performance at multiple scales. Geoderma, 352, 342-350. Song, X. D., Wu, H. Y., Ju, B., Liu, F., Yang, F., Li, D. C., ... & Zhang, G. L. (2020). Pedoclimatic zone-based three-dimensional soil organic carbon mapping in China. Geoderma, 363, 114145. Line 66: SCORPAN also includes soil information compared to Jenny's soil forming theory. Lines 89-90: It seems to me that this manuscript does not take the SOC changes during three periods into account. I would add relevant statements with supporting references. Lines 103-104: Since mineral soil is sampled at 0-10, 10-20 and 55-65 cm depth, which kind

of interpolation is used to harmonize them at 0-50 cm? Please provide more details. Line 158: RF is commonly used instead of RFR. Line 216: Which method is used in recursive feature elimination in caret package? Random forest? Line 247: What does p mentioned here?

———————————————————

---

## Referee Comment (RC2) · Madlene Nussbaum (Referee) · 14 Jan 2021

The manuscript shows the spatial prediction of carbon stocks in forests of Sweden for tow compartments (organic layer and mineral soil 0-50 cm depth) and the sum of the two. Random forest with parameter tuning and feature selection is used and prediction intervals are computed by quantile regression forest. Different model configuration is shown along two axes: 1) The data is split into subregions and separate model calibrations are presented. The results are then compared to a model covering the complete area. 2) Different subsets of covariates as model input are formed according

to their origin. Covariates only available at the observed sites and geodata available for whole Sweden were tested separately. In addition models combining all covariates were fitted and compared to the other two versions.

The methods are fully described and the results thoroughly discussed. The prediction intervals are validated with data not used to create the models which is so far rarely seen. Moreover, the distribution of the actual data is compared to the distribution of all predicted pixels. The manuscript is well written and the figures mostly are well prepared. Well done, Ozias!

However, I share the concerns of referee #1:

1. On page 6 lines 227–232 the data splitting approach is explained. I do not fully understand how it was actually done as I got puzzled by the last sentence. Were the validation data the same data points for the global and the local models (last sentence) or were they independently chosen (rest of this paragraph)?

   Nevertheless, the validation statistics should be computed on the exact same validation dataset. Especially the measure $R^2$ is very sensitive to different values of observed data. This means for the global model three local sets of validation statistics should be computed on the same data points. Of course overall statistics can also be presented, but should not be compared to the local statistics, because the standard deviation of the total vector of observed values might be larger.

   To achieve such a validation the 20-80% splitting should be done as stratified random sampling from the data points with the subareas as strata (which maybe already has been done).

2. The covariates only available at the observed locations cannot be used to create prediction at locations without any observation. Hence, they are not useful to

create area-wide maps unless one creates maps of these site specific characteristics beforehand. The same geodata would be used to create these predictions. Such a two-step prediction approach (some sort of a random forest co-kriging) is only meaningful if 1) there are more observations of this intermediate response (site characteristic) than for the final target response (SOC stock) which I understand is not the case for the current data set; 2) the intermediate response results for some reason in much better model performance and predictive accuracy.

The current study neglects to consider errors of the needed spatial prediction of the site characteristic covariates. The error of the spatial prediction might possibly be as high as to render these covariates useless. At least, the interpretation should put this clearly into perspective.

Moreover, please consider the small comments directly added to the manuscript using PDF annotations.

14.01.2021 / M. Nussbaum, BFH-HAFL

Please also note the supplement to this comment:
https://soil.copernicus.org/preprints/soil-2020-75/soil-2020-75-RC2-supplement.pdf

**Supplement:**

[revised manuscript text omitted]

---

## Referee Comment (RC3) · Anonymous Referee #3 · 19 Jan 2021

Manuscript title: "Predicting the spatial distribution of soil organic carbon stock in Swedish forests using remotely sensed and site-specific variables."

Reviewer comments

General comment

The manuscript is well written and generally clear, with an appropriate structure and the information provided in tables and figures is useful and necessary, although I have some specific comments for the presentation of some figures. The length of the paper is appropriate, as the presentation of the results is synthetic, and the discussion gives a concise explanation of the observed results supported by other findings in the SOC literature.

This research paper investigated the key variables for predicting soil organic carbon (SOC) stocks in the litter layer, mineral soil, and total SOC of forest soils in Sweden, and maps its spatial distribution using random forest models. The study compares the accuracy of global models (calibrated for the whole study area, Sweden) and local models for north, central and southern Sweden. The calibration data originated from the Swedish National Forest and as predictor variables they compared three different sets: 1) only site variables observed at the sampling plots, 2) remote sensing variables, and 3) all variables.

My main comment may be more a suggestion for the follow-up study. Mapping some of the site variables that were more decisive for SOC prediction (soil moisture class, vegetation type, soil type and soil texture) and including them as covariates for mapping may improve the model accuracy for mapping. However, as these variables will be themselves estimated with statistical models, there may be an increase in the uncertainty due to error propagation. Hence, the uncertainty of the map for a model including all variables may be a conservative estimate. Also, in that future scenario, consider that if you calibrate the model with the data observed at the plots but map it with the gridded estimates of the site variables, the accuracy may also overestimated. If you calibrate the model with the gridded predictions for soil moisture class, soil texture, etc., perhaps they may not be as relevant covariates, and maybe the accuracy of the model will not be the highest.

I recommend acceptance after minor revisions, for the following sections.

Specific comments

L51-53: Maybe you can include geostatistics as part of the modelling methods.

L55: consider changing "modelling over a large landscape" to "large extent".

L66: include soil with "the inclusion of the location coordinates".

L86-94: This paragraph is somewhat confusing. Does the NFSI run every year or every 5 years? Maybe give a reference for the NFSI or the NFI datasets.

L99-L100: Please, indicate if the content of inorganic carbon in mineral soil is negligible in the study area.

L104: what type of interpolation? Splines? Linear?

L150: the soil moisture class is later shown as a very important variable. Perhaps you could describe it a bit more. It refers to the frequency of the year it is dry/moist due to the proximity of the water table, or also influenced by soil texture (e.g., soil drainage class)?

L153: The field layer refers to the understory?

L159: did you also used the QRF models to predict the 5th and 95th percentiles and create the 90% prediction interval?

L201-203: I would have used a larger buffer, but I imagine that you assessed that this length was appropriate. Later in the results, I miss some information on how the local maps overlapped, whether or not there was some edge effect. In figure 6 it seems that there were not large differences in the boundaries.

L222-225: This sentence is not very clear. You split the dataset into a calibration (80%) and independent validation (20%). And then you apply the tenfold cross-validation, repeated 5 times, on the calibration subset, right?

L276-285: Here, the results for the independent validation are a bit short compared with the report for the cross-validation, or if you refer to the independent validation as "local models after cross validation" (L282), maybe that it not very clear. I was looking at Table 4 and the numbers don't seem to correspond completely. Please revise this (e.g., $R^2$ for allV at independent validation 12-36 % and 17-32 % at cross-validation).

L292-293: This sentence is not completely clear. Please specify that the local models had better performance than the global models in term of RMSE within each set of variables.

L360-362: Could you provide the map of standard deviation?

L491-495: The relevance of soil texture as predictor of SOC stock is also explained by the physico-chemical SOC stabilization mechanisms. Clay minerals and clay and silt sized particles generally have a positive correlation with mineral SOC stocks, as the association of organic matter with mineral surfaces, and occlusion inside aggregates hinders microbial decomposition and enhances SOC accumulation. There are many references in the literature on this topic. For example:

Lützow, M.V., Kögel-Knabner, I., Ekschmitt, K., Matzner, E., Guggenberger, G., Marschner, B. and Flessa, H., 2006. Stabilization of organic matter in temperate soils: mechanisms and their relevance under different soil conditions–a review. *European journal of soil science*, *57*(4), pp.426-445.

L550-554 & 559: Consider my comment on how there would be error propagation and the final uncertainty of the SOC predictions may be increased when maps of these site attributes are included as predictors for mapping. However, I would also include these maps as predictor variables when they are available.

Technical comments

L20: "Random Forest models".
L43: Place the abbreviation of carbon (C) before in the text, in line 35 when you use it for the first time.
L228: "buil**t**".
L294: "best local model**s**".

L326: 40**K** (capitalize the K).
L372: "When prediction**s** were carried out".
L385: "outperform"
L386-387: Please indicate that you refer to the best local models.
L397: "remain**ed** low"
L475: "organic matter"

Figure 6: Please, indicate in the caption which in the figure (left/right) are the global and the local models. Also, maybe you could include a figure with the standard deviation or the 5[th] and 95[th] percentiles (predicted with the quantile random forests regression) so we can also visualize the uncertainty. The colour scale for the total SOC stock (in less extent) but mainly for the mineral SOC stock is not very clear, as there are different tonalities of green and brown for different value ranges. Could you use a different sequential palette, like for the predictions of the humus layer? (maybe multi-hue sequential palette). For example, the package *colorspace* has many options.

Supplementary material S1: Maybe you can expand the supplementary material one more page and make the plots larger on their y axis.  They are not very clear like this.

---

## Referee Comment (RC4) · Anonymous Referee #3 · 19 Jan 2021

I forgot to add something. when you map over the whole study area, what GIS layer did you use to mask non-forested areas? you could include that in the methods.

---

## Author Comment (AC4) · 24 Feb 2021

Dear Reviewer,

Please find enclosed the revision of our manuscript "Predicting the spatial distribution of soil organic carbon stock in Swedish forests using remotely sensed and site-specific covariates" by Hounkpatin et al. We thank you for the very competent review of our paper and the productive comments towards the improvement of our manuscript. We agree with almost all of them and revised the manuscript accordingly.

[Figure]

We hope that our paper is now acceptable for publication in SOIL.

Yours sincerely,

Ozias Hounkpatin

Answers to reviewer

The manuscript attempted to compare the model performance of global and local models in predicting humus layer, mineral soil and total SOC stock and to identify the controlling factors for SOC stock prediction. Besides, this study also investigated the effect different combinations of data from site characteristics and remotely sensed variables on model performance. The results from independent dataset indicated that the local models generally had better model performance than the global models. The only use of remotely sensed variables had limited predictive ability while site characteristics had better explanatory strength in estimating SOC stocks. The authors suggest that further work can focus on mapping these influential site covariates. The manuscript is overall well-written with clear objectives and reasonable methodology. However, two major limitation of this manuscript are: (1) in comparing global and local models, I can tell from Table 4 and Table 5 that some local models had higher R2 than that of global models while the remaining local ones had lower R2. Therefore, the conclusion that local models have a comparative advantage over global models is not convincing for me. Again, it is not fair to compare the performance indicators as mentioned in this study for local and global models. Instead, for the global model, authors should also calculate indicators for three regions separately to make them comparable for these in local models; - Author's Response: The error metrics were actually computed based on the same validation set for both global and local models, this to avoid comparison bias. Table 3 actually shows the error metrics for the global models when the independent validation set for the (different strata) North, Central and South Sweden were merged. Table 4 shows the specific error metrics for the local models against their respective validation set. Figure 2 presents the error metric results of each global model

for the three subareas along with corresponding local models (unfortunately repeating error metrics of local models from Table 4 for the independent validation. To ensure clarity, we provided more details in the section (see lines 224 – 231): related to the data splitting procedure.

- Indeed some local models had higher R2 than that of global models while the remaining local ones had lower R2. However, we considered their performance in relation to the specific category of predictors the models were built from. In that regards, our conclusion was that in general local models with site specific covariates only or those built using all the covariates did in most cases perform marginally better than the global models.

Manuscript lines (see lines 224 – 231): The RF models built on data covering the whole area of Sweden are hereafter called "global models". The RF models created for each of the subareas are hereafter reported as "local models". Considering the subareas as strata, the local models were built by randomly splitting the local datasets into calibration (80%) and validation (20%) subset. The training set of the global models was constituted by merging the 80% split of the strata dataset. To avoid comparison bias, both the global and local models were evaluated on the same validation set. Consequently, each global model was evaluated again three validation subset separately corresponding each to the 20% split of the Northern, Central and Southern local dataset. The merged 20% split of these local datasets was then used as validation set at national scale. We trained both global and local models based on tenfold cross validation with 5 repetitions using the R "caret" package (Kuhn, 2015). (2) As mentioned by authors, the site characteristics are only available at the visited plots, therefore the digital maps can not be produced by the models built with site characteristics, which certainly limits the usage of site characteristics. I am afraid that the importance of these site characteristics in this study have been overlooked as the observed site characteristics are directly used in independent validation which certainly ignores the inaccuracy of these site characteristics if they are mapped by certain algorithms. I mean, the site char-
acteristics used in a fair independent validation should come from the predicted maps of these relevant site characteristics, not from observed data. Therefore, I suggest a major revision before it can be published. - Author's Response: We share the concern of the reviewer that all covariates should actually be maps for DSM of the whole area while the site specific data were only observations. Initially, the focus was mainly evaluating the geodata we now have versus site specific variables and not necessarily making a map. We made the map based on the geodata to give a visual trend of the distribution of the SOC stock. We also agree that the current study neglects to consider mapping uncertainties from potential maps of site specific variables which might actually translate into lower prediction accuracies of SOC stock compared to models based on their observation data. Our interpretation failed to put this clearly into perspective and we have now created a section discussing this issue. To elaborate more on this issue in that section, we also made some preliminary maps (which need further improvements because of data imbalance issues) of the site specific variables and used these maps only as covariate to assess the quality of their predictions.

Manuscript lines (see lines 585 – 615): see section 4.4 Implication and limitations of the study DSM relies on existing maps for building regression models and ultimately prediction for mapping. The quality and accuracy of predictions depend as discussed earlier on choosing the most relevant covariates in relation to the target to be predicted. The present study revealed that covariates which were available as maps did contribute to the MDA, but site characteristics were more prominent in relation to the SOC stock in Sweden. This might suggest that mapping these variables that were more decisive for SOC stock prediction and including them as covariates for mapping may improve accuracy. Since the primary focus of the present study was mainly to evaluate the GoC data versus the SSC data at different scale of modelling and not primarily for making a map, the observed data of the latter were used. However, since only the observed data of the site characteristics were considered, this study fails to consider that the mapping of these site characteristics would involve modelling errors and the propagation of these errors into the final maps of these site variables might actually

reduce their predictive power. For completeness, we carried out a preliminary mapping of the site characteristics (SI-11) using additional soil inventory data, random forest and the GoC as predictors. These mapped site characteristics (mSSC) were then used as covariate for predicting the SOC stocks.

Table 6 presents the error metrics after independent validation for both local and global models along with the percentage margin of the RMSE in relation of the models based on GoC. First, the local mSSC based models still recorded lower RMSE as compared to global mSSC models. Compared to the GoC models, the overall positive percentage margin of the RMSE for the independent validations indicated that the mSSC models recorded the lowest RMSE. However, when assessing the RMSE margin between the global models of GoC and SSC, negative percentages were mainly recorded for the Northern Sweden independently of the depth. This indicated that the mSSC based global models were less accurate at predicting SOC stock locally in the Northern Sweden while the mSSC based local model did present a better prediction for Northern Sweden for the humus layer and mineral SOC stock. The mean SOC predictions based on the mSSC showed a stronger increasing gradient from Northern to Southern Sweden (SI-12) as compared to the pattern observed with the GoC maps. However, the uncertainty distribution was of similar magnitude (SI-12) as those observed for the maps based on GoC probably due to error propagations as these covariates were used to make the site specific characteristics maps. This suggests that the SSC should still be supplemented for improvement at this stage with other covariates different from the GoC such as multi-temporal spectral (e.g. normalized difference vegetation index) data that are able to capture vegetation dynamic in forests. Notwithstanding the fact that was obviously error propagation, the study of which was beyond the scope of this study, the preceding results tend to confirm the potential of the high resolution maps of the site characteristics to contribute to the improvement of SOC stock prediction as compared to using only the GoC data. Given that the preliminary mappings of the SSC recorded low kappa (0.17 – 0.48) values (SI-11) at this stage further improvements are still necessary to improve their predictive ability and associated coefficient of variations.

Specific comments are listed below: Line 49: which are nonspatial environmental co-variates? If they are not spatial, how they can be used in DSM? - Author's Response: Sentence corrected with removal of the word non-spatial Manuscript lines (see lines 47 – 50): DSM aims at estimating the spatial distribution of soil classes or soil properties by coupling field and laboratory observations with spatial environmental covariates via quantitative relationships. Line 58: Authors miss at least two papers on comparing local and global models in DSM. Piikki, K., & Söderström, M. (2019). Digital soil mapping of arable land in Sweden–Validation of performance at multiple scales. Geoderma, 352, 342-350. Song, X. D., Wu, H. Y., Ju, B., Liu, F., Yang, F., Li, D. C., ... & Zhang, G. L. (2020). Pedoclimatic zone-based three-dimensional soil organic carbon mapping in China. Geoderma, 363, 114145. - Author's Response: Thanks for mentioning these papers. The suggested papers have been included. Manuscript lines (see lines 58 – 60): However, models could be calibrated separately for subareas and their predictions can then be combined to cover the whole area (Somarathna et al., 2016; Piikki and Söderström, 2019; Song et al., 2020). Line 66: SCORPAN also includes soil information compared to Jenny's soil forming theory. - Author's Response: We agree with the comment and included soil information in addition to location position for additional elements from SCORPAN compared to Jenny's soil forming theory. Manuscript lines (see lines 65 – 67): Building on the soil state-factor (climate, organisms, relief, parent material, age) equation developed by Jenny (1941), McBratney et al. (2003) introduced the conceptual framework for DSM referred to as SCORPAN which complemented the former with the inclusion of soil information and location coordinates. Lines 89-90: It seems to me that this manuscript does not take the SOC changes during three periods into account. I would add relevant statements with supporting references. - Author's Response: The reviewer is right that the manuscript does not focus on SOC changes over the three inventory periods. We rather focus on SOC stock by considering the averaged SOC at plot level as a good representative of the plots which we indicated in lines 107 – 110. We therefore explained clearly our focus and added further references for the dataset. Manuscript lines (see lines 89 – 93): The NFSI runs concurrently every

year with the NFI and consist in repeated survey of forest vegetation and soil chemical and physical properties (Stendahl et al., 2017;Ortiz et al., 2013). Data from the following inventory periods were considered in the present study: 1993 – 2002, 2003 – 2012 and 2013 – 2015. However, the present paper did not focus on SOC changes over these three inventory periods but on SOC stock using plot scale as a unit. Manuscript lines (see lines 110 – 112): Since potential SOC stock change is very small compared to the entire SOC stock the averaged SOC stock between the inventories was considered representative of the plots and was therefore considered for all computations and modelling in order to reduce variability between plots

Lines 103-104: Since mineral soil is sampled at 0-10, 10-20 and 55-65 cm depth, which kind of interpolation is used to harmonize them at 0-50 cm? Please provide more details. - Author's Response: The carbon stocks for each layer with data is calculated. Thereafter the carbon pool is assumed to change linearly between measured layers. Even if the deepest sampled horizon (55-60 cm) is deeper than the top 50 cm it is utilized in the interpolation but the carbon stock below 50 cm is not counted in. Manuscript lines (see lines 107 - 110): The total mineral SOC stock down to 50 cm depth for each site is calculated using the SOC stock of measured layers with empirical model for bulk density (Nilsson and Lundin, 2006), corrections for stoniness (Stendahl et al., 2009) and linear interpolation between measured layers Line 158: RF is commonly used instead of RFR. - Author's Response: RFR has been replaced by RF throughout the document. Manuscript lines (see lines 65): The Random Forest (RF) algorithm was selected for SOC stock prediction. Line 216: Which method is used in recursive feature elimination in caret package? Random forest? Author's Response: Random Forest was used as method in recursive feature elimination in caret package. We now added this method in the referred line. Manuscript lines (see lines 220 – 221): . . .and (2) the recursive feature elimination (RFE) using RF as method to select the optimal set of covariates for each RF model. Line 247: What does p mentioned here? - Author's Response: The p stands for probability value. We corrected and put the p in bracket after probability. - Manuscript lines (see lines 250):

The latter is the graphical representation of the proportion of time the actual values of SOC stock fall within a series of probability (p) of prediction intervals (PI) limited by (1-p)/2 and (1+p)/2 quantiles.

Please also note the supplement to this comment:
https://soil.copernicus.org/preprints/soil-2020-75/soil-2020-75-AC4-supplement.pdf

**Supplement:**

Table 6: Cross-validation, independent validation of the global and local random forest models based on the mapped site specific covariates compared to models based on observed site specific covariates and grouped of covariates

| | | | RMSE (t C/ha) | $R^2$ | ΔRMSE GoC-mSSC (%) |
|---|---|---|---|---|---|
| | | | Independant validation | | |
| Global models | Humus layer | All Sweden | 20.87 | 0.27 | 5.57 |
| | Mineral soil | All Sweden | 26.57 | 0.28 | 13.44 |
| | Total | All Sweden | 34.35 | 0.28 | 10.55 |
| Local models | Humus layer | North | 19.22 | 0.22 | 2.46 |
| | | center | 17.56 | 0.16 | 2.47 |
| | | South | 24.81 | 0.31 | 6.37 |
| | Mineral soil | North | 20.21 | 0.22 | 4.68 |
| | | center | 22.89 | 0.09 | 20.23 |
| | | South | 32.11 | 0.29 | 11.04 |
| | Total | North | 29.82 | 0.28 | -0.74 |
| | | center | 27.36 | 0.09 | 22.06 |
| | | South | 41.79 | 0.23 | 6.30 |
| | | | Error metrics by Global models for local validation set | | |
| | Humus layer | North | 20.28 | 0.14 | -1.91 |
| | | center | 18.02 | 0.13 | -1.24 |
| | | South | 25.27 | 0.27 | 4.64 |
| | Mineral soil | North | 22.72 | 0.10 | -4.70 |
| | | center | 23.14 | 0.05 | 20.75 |
| | | South | 34.96 | 0.15 | 2.35 |
| | Total | North | 40.70 | 0.01 | -43.56 |
| | | center | 30.00 | 0.07 | 13.49 |
| | | South | 43.18 | 0.18 | 0.74 |

ΔRMSE GoC-mSSC (%): percentage estimate of the difference between the root mean square error of models based on group of covariates and mapped site specific covariates, negative values: models with mapped site specific covariates present higher root mean square error as compared to models based on either observed site specific variable or group of covariates, positive values : models with mapped site specific covariates present lower root mean square error as compared to models based on either observed site

[Figure]

Figure 6: Mean SOC stock prediction and Prediction uncertainties of the spatial distribution of the humus layer, mineral soil and total SOC stock based on the group of covariates

**Predicting the spatial distribution of soil organic carbon stock in Swedish boreal forest using remotely sensed and site-specific covariates**

**Kpade O. L. Hounkpatin[a1], Johan Stendahl[a], Mattias Lundblad[a] , Erik Karltun[a],**

[a] Department of Soil and Environment, Swedish University of Agricultural Sciences, P.O. Box 7014, SE-75007, Uppsala, Sweden

**Supplementary Information (SI)**
* * *
[1] Correspondence : ozias.hounkpatin@slu.se

Figure SI-1A: Coverage of 90 %-prediction intervals computed for the local and global random forest models for the humus layer, mineral soil and total SOC stock based on the site specific covariate models

[Figure]

Figure SI-1B: Coverage of 90 %-prediction intervals computed for the local and global random forest models for the humus layer, mineral soil and total SOC stock based on the group of covariate models

[Figure]

Figure SI-1C: Coverage of 90 %-prediction intervals computed for the local and global random forest models for the humus layer, mineral soil and total SOC stock based on all covariate models

[Figure]

Figure SI-2: Correlation matrix of all variables with the humus layer SOC stock dataset over Sweden. Refer to table 1 for the definition of the abbreviation

[Figure]

Figure SI-3: Correlation matrix of all variables with the mineral soil SOC stock dataset over Sweden. Refer to table 1 for the definition of the abbreviation

[Figure]

Figure SI-4: Correlation matrix of all variables with the total soil SOC stock dataset over Sweden. Refer to table 1 for the definition of the abbreviation

[Figure]

Figure SI-5: Variogram for the SOC stock and regression residuals of the local and global Random Forest models for the humus layer, mineral soil and total SOC stock. Log: log transformation, sqrt: square root transformation, Sph: spherical model, Ste: Matern, M. Stein's parameterization.

[Figure]

Figure SI-6: Box-plot for humus layer SOC stock by soil moisture, soil type, vegetation type, parent material and soil texture. Lines within the boxes give the median, red point within the boxes is the mean, boxes the 25th and 75th percentile (Refer to Table 2 for definition of the classes of each variable)

[Figure]

Figure SI-7: Box-plot for the mineral soil SOC stock by soil moisture, soil type, vegetation type, soil texture and parent material. Lines within the boxes give the median, red point within the boxes is the mean, boxes the 25th and 75th percentile (Refer to Table 2 for definition of the classes of each variable)

[Figure]

Figure SI-8: Box-plot for total SOC stock by soil moisture, soil type, vegetation type, soil texture and parent material. Lines within the boxes give the median, red point within the boxes is the mean, boxes the 25th and 75th percentile (Refer to Table 2 for definition of the classes of each variable)

[Figure]

Figure SI-9: Partial dependence plots. Refer to Table 1 for the definition of the abbreviation and numbers for the classes of the variables.

[Figure]

A: Partial plot between the humus layer SOC stock,
soil moisture and vegetation type
(RF Global model with all variables)

B: Partial plot between the humus layer SOC stock,
soil moisture and Northing
(RF Global model with all variables)

C: Partial plot between the mineral SOC stock,
Northing and soil type
(RF Global model with site specific variables)

D: Partial plot between the mineral SOC stock,
Easting and soil type
(RF Local model with site specific variables)

E: Partial plot between the total SOC stock,
soil moisture and soil type
(RF Global model all variables)

F: Partial plot between the total SOC stock,
precipitation and soil moisture
(RF Global model all variables)

Figure SI-10: Partial dependence plots between SOC stock, northing, temperature and precipitation.

A

[Figure]

A: Partial plot between the humus layer
SOC stock, Northing and Temperature
(RF Global model with all variables)

B

B: Partial plot between the humus layer
SOC stock, Northing and precipitation
(RF Global model with all variables)

C

[Figure]

C: Partial plot between the mineral SOC stock,
Northing and Temperature
(RF Global model with all variables)

D

[Figure]

D: Partial plot between the mineral SOC stock,
Northing and precipitation
(RF Global model with all variables)

E

[Figure]

E: Partial plot between the total SOC stock,
Northing and Temperature
(RF Global model with all variables)

F

[Figure]

F: Partial plot between the total SOC stock,
Northing and precipitation
(RF Global model with all variables)

Figure SI-11: Maps of the site specific covariates

[Figure]

[Figure]

**Figure SI-12: SOC prediction and uncertainties based on mapped site-specific covariates**

[Figure]

**Asset SI-13: R code for SOC stock prediction**

```
**-----------------------------------Script R SOC stock modelling ----------------------------------**
###################################################################
**Author : Kpade Ozias L. Hounkpatin**
**Paper : Predicting the spatial distribution of soil organic carbon stock in Swedish forests using remotely**
**sensed and site-specific variables**
**Code: The following code applies for litter layer, mineral soil and total soil carbon stock**

**Load necessary Packages**
library(quantregForest)
library(rgdal)
library(raster)
library(spatial.tools)
library(caret)
library(randomForest)
library(imputeTS)
library(stringr)
library(sp)
library(xgboost)
library(plyr)
library(dplyr)
library(parallel)
library(tidyverse)
library(Metrics)
library(ithir)
library(gridExtra)
library(doParallel)
library(pastecs)
library(utils)
library(broom) # for tidy()
library(knitr) # for kable()
library(stargazer)
library(pdp)
**Set options**
options(digits = 2,scipen=999)
**Set my working directory**
setwd("yourpath")

**--------------------------------------------------------------------------------------------------------------**
**-----Load data, Deal with NA values & remove problematic column-----------------------------------------------**
**--------------------------------------------------------------------------------------------------------------**
**---Load the table**
litterdata<- read.table("./path/litterdata.txt", header = T, sep = "\t")
**-- Change data class for site specific variables**
**cor.train <- lapply(cor.train[:], as.factor), #insert where factor variables start and end in [:]**
**cor.test<- lapply(cor.test[:], as.factor), #insert where factor variables start and end in [:]**
**Transform into spatial soildata frame**
coordinates(litterdata) <- ~EastC+NorthC
**Project the soildata into EPSG : 3006**
crs<-CRS("+init=epsg:3006")
proj4string(litterdata) <- crs
**-Load the predictors**
rlist<-list.files("path", pattern="tif$", full.names=TRUE)#Load in the covariates
covStack <- stack(rlist)#Stack all the covariates
**Extract multi value from covariates---TRAIN**
litterdata_cov<- raster::extract(covStack, litterdata,sp = 1, method = "simple")
```

```
**-----------------------------------------------------------------------------------------------------------------------**
**-------------------------train set, Validation set---------------------------------------------------------------------**
**-----------------------------------------------------------------------------------------------------------------------**
**Specify the percentage that will be used for training. the remaining will be used for testing**
set.seed(1)
split <- 0.8
**Training samples  for soil moisture**
trainIndex <- createDataPartition(c(litterdata_cov["C_lit"], recursive=T), p=split, list = F)
pcDat<- liptData[trainIndex,] # Training set
ptDat<-liptData[-trainIndex,] # Validation set

**-----------------------------------------------------------------------------------------------------------------------**
**------------------CHECKING HIGHLY CORRELATED COVARIATES----------------------------------------------------------------**
**-----------------------------------------------------------------------------------------------------------------------**
**Only for numeric variables**
**corstarsl: https://github.com/kyuni22/ksmv/blob/master/functions/corstarsl.R**
corstarsl <- function(x){
  require(Hmisc)
  x <- as.matrix(x)
  R <- rcorr(x)$r
  p <- rcorr(x)$P
  ## define notions for significance levels; spacing is important.
  mystars <- ifelse(p < .001, "***", ifelse(p < .01, "** ", ifelse(p < .05, "* ", " ")))
  ## trunctuate the matrix that holds the correlations to two decimal
  R <- format(round(cbind(rep(-1.11, ncol(x)), R), 2))[,-1]
  ## build a new matrix that includes the correlations with their apropriate stars
  Rnew <- matrix(paste(R, mystars, sep=""), ncol=ncol(x))
  diag(Rnew) <- paste(diag(R), " ", sep="")
  rownames(Rnew) <- colnames(x)
  colnames(Rnew) <- paste(colnames(x), "", sep="")
  ## remove upper triangle
  Rnew <- as.matrix(Rnew)
  Rnew[upper.tri(Rnew, diag = TRUE)] <- ""
  Rnew <- as.data.frame(Rnew)
  ## remove last column and return the matrix (which is now a data frame)
  Rnew <- cbind(Rnew[1:length(Rnew)-1])
  return(Rnew)
}
corr.data<-corstarsl(pcDat[sapply(pcDat,is.numeric)])
**Save correlation data**
write.table(corr.data, file = str_c("./path/litterSOC/RFEdata_LitterSOC.txt"), sep = "\t",row.names=F)
descrCorr <- cor(pcDat[sapply(pcDat,is.numeric)])
highcorr <- caret::findCorrelation(descrCorr, cutoff = 0.80,names = TRUE)
cor.train<-pcDat %>% dplyr::select(-highcorr)
cor.test<-ptDat%>% dplyr::select(-highcorr)

**-----------------------------------------------------------------------------------------------------------------------**
**------------------Feature Engineering_RFE------------------------------------------------------------------------------**
**-----------------------------------------------------------------------------------------------------------------------**
xtrain <- cor.train[, 2:ncol(cor.train)]
xtest <-cor.test[, 2:ncol(cor.test)]
ytrain <- cor.train[, 1]
ytest <- cor.test[, 1]
rfe.ctrl = rfeControl(functions = rfFuncs,
              method = "cv",
              number = 50,
              allowParallel = TRUE,
              verbose = TRUE)
set.seed(1)
RF_rfe<- rfe(x=xtrain, y=ytrain,rfeControl = rfe.ctrl)
```

```
modelFile <-paste("./path",sep = "", "rfemodel_LitterSOC", ".rds")
saveRDS(object = RF_rfe, file = modelFile)
**Get names of the eliminated variables**
RFNotsel<-xtrain %>% dplyr::select(-one_of(predictors(RF_rfe)[1:length(predictors(RF_rfe))]))
**Get final rfe train dataset**
se_RF_train<-xtrain %>% dplyr::select(one_of(predictors(RF_rfe)[1:length(predictors(RF_rfe))]))
RF_train<-cbind(cor.train$C_lit,se_RF_train)
names(RF_train)[1] <- "C_lit" #Rename First column
write.table(RF_train, file = str_c("./path/train_rfe_LitterSOC.txt"), sep = "\t",row.names=F)
**Get final rfe test dataset**
se_RF_test<-xtest %>% dplyr::select(one_of(predictors(RF_rfe)[1:length(predictors(RF_rfe))]))
RF_test<-cbind(cor.test$C_lit,se_RF_test)
names(RF_test)[1] <- "C_lit" #Rename First column
write.table(RF_test, file = str_c("./path/test_rfe_LitterSOC.txt"), sep = "\t",row.names=F)

**-------------------------------------------------------------------------------------------------------------**
**-------------------TUNE RF model-----------------------------------------------------------------------------**
**-------------------------------------------------------------------------------------------------------------**
RFgrid <- expand.grid(.mtry = 2:ncol(RF_train))
control <- trainControl(method="repeatedcv", number=10,
                repeats = 3,
                verboseIter = TRUE,
                returnData = "FALSE",
                returnResamp = "all",
                allowParallel = TRUE)
set.seed(1)
Tune_RF<- train(x=xtrain,y=ytrain,method = "rf",
         trControl=control,tuneGrid = RFgrid)
**Save the model**
modelFile <-paste("./path",sep = "", "tunemodel_LitterSOC", ".rds")
saveRDS(object = Tune_RF, file = modelFile)
RF_tune_Table<-as.data.frame(Tune_RF$bestTune)

**-------------------------------------------------------------------------------------------------------------**
**------------------CROSS VALIDATION with RF-------------------------------------------------------------------**
**-------------------------------------------------------------------------------------------------------------**
control <- trainControl(method="repeatedcv", number=10,
                repeats=5,
                verboseIter = TRUE,
                returnData = "FALSE",
                returnResamp = "all",
                allowParallel = TRUE)
RFgrid<-expand.grid(mtry = RF_tune_Table$mtry)
set.seed(7)
fit_RF <- train(x=RF_train[,2:ncol(RF_train)],
         y=RF_train$C_lit,method = "rf",
         trControl=control,
         tuneGrid = RFgrid,
         importance = TRUE)
modelFile <-paste("./path", sep="","CVmodelRF_LitterSOC",".rds")
saveRDS(object = fit_RF, file = modelFile)
**Get and Save resampling statistics**
CV_RF_Stat.desc<-as.data.frame(stat.desc(fit_RF$resample,basic=FALSE))
write.table(CV_RF_Stat.desc, file = str_c("./path/Stat.desc_LitterSOC.txt"), sep = "\t",row.names=F)
**Predict for external validation**
C_lit.pred.RF<- as.data.frame(predict(fit_RF, xtest))
**Get external validation statistics and save to disk**
obs.C_lit.pred.RF<-as.data.frame(cbind(ytest, C_lit.pred.RF))
names(obs.C_lit.pred.RF)[1]<-"obs"
names(obs.C_lit.pred.RF)[2]<-"pred"
```

```
RF.goof<-as.data.frame(goof(obs.C_lit.pred.RF$obs,obs.C_lit.pred.RF$pred))
**Write table for Results EV for RF**
RF.goof$mae<-mae(obs.C_lit.pred.RF$obs,obs.C_lit.pred.RF$pred)
write.table(RF.goof, file = str_c("./path/goofRF_LitterSOC.txt"), sep = "\t",row.names=F)

**--------------------------------------------------------------------------------------------------------------------------**
**------------------Variable Importance-------------------------------------------------------------------------------------**
**--------------------------------------------------------------------------------------------------------------------------**
VI.RF<-varImp(fit_RF)
RF.VImp<-as.data.frame(VI.RF[1])
names(RF.VImp)[1]<-"VImp.RF"
RF.VImp<-as_tibble(setNames(cbind(rownames(RF.VImp), RF.VImp, row.names = NULL), c("Factors",
"Importance")))
RF.VImp<-as.data.frame(RF.VImp[order(-RF.VImp$Importance),])
write.table(RF.VImp, file = str_c("./path/VarImpCVRF_LitterSOC.txt"), sep = "\t")

**--------------------------------------------------------------------------------------------------------------------------**
**------------------Density Plots-------------------------------------------------------------------------------------------**
**--------------------------------------------------------------------------------------------------------------------------**
RF_SOC.test<-cbind(RF_test$C_lit,C_lit.pred.RF)
RF_SOC.test<-setNames(RF_SOC.test, c("C_lit","RF_pred"))
write.table(RF_SOC.test, file = str_c("path/Obs_PredRF.data_LitterSOC.txt"), sep = "\t",row.names=F)
**RF_density plots with actual and predicted SOC**
RF_density<-RF_SOC.test[,c(1,length(RF_SOC.test))]
RF_density$Type1<-"Actual"
RF_density$Type2<-"PredictedC"
RF_d.soil<-RF_density[,c(1,3)]
RF_d.pred<-RF_density[,c(2,4)]
RF_d.soil<-setNames(RF_d.soil, c("SOC",          "Type"))
RF_d.pred<-setNames(RF_d.pred, c("SOC",        "Type"))
RF_density.data<-(as.data.frame(rbind(RF_d.soil,RF_d.pred)))
RF_predType<-as.data.frame(ddply(RF_density.data, "Type", summarise, rating.pred=mean(SOC)))
write.table(RF_predType, file = str_c("./path/densityRF_predType_LitterSOC.txt"), sep = "\t",row.names=F)
write.table(RF_density.data, file = str_c("./path/densityRF_data_LitterSOC.txt"), sep = "\t",row.names=F)
**Density plots with semi-transparent fill**
N10RF_densityP_lit<-ggplot(RF_density.data, aes(x=SOC, fill=Type))+geom_density(alpha=0.4)+
  geom_vline(data=RF_predType, aes(xintercept=rating.pred,  colour=Type),linetype="dashed")+
        labs(y = "Density", x = "SOC stock",title = "litter SOC stock",
        linetype="dashed", size=1,show.legend=FALSE)+
        scale_color_manual(values=c("#999999", "#E69F00"))+
        scale_y_continuous(breaks=c(0,0.035, 0.065))+
  theme(axis.title.x = element_blank())+
  theme(legend.position = "none")+
  theme(axis.text=element_text(size=9, face="bold"),axis.title=element_text(size=10, face="bold"),#,face="bold"
      plot.title = element_text(size = 14, face = "bold",hjust = 0.5))+
  theme(panel.background = element_rect(fill = "white", colour = "#f7f9fb",size = 2, linetype = "solid"))

**--------------------------------------------------------------------------------------------------------------------------**
**------------------Compute Partial dependance------------------------------------------------------------------------------**
**--------------------------------------------------------------------------------------------------------------------------**
**Compute partial dependence data**
var1<-as.character(RF.VImp["1",1])
var2<-as.character(RF.VImp["2",1])
var3<-as.character(RF.VImp["3",1])
RF_train<-as.data.frame(RF_train)
pd <- pdp::partial(fit_RF, pred.var = c(var1, var2),train=RF_train, rug=TRUE)
**Default PDP**
pdp1 <- plotPartial(pd)
**Add contour lines and use a different color palette**
rwb <- colorRampPalette(c("red", "white", "blue"))
```

```r
pdp2 <- plotPartial(pd, contour = TRUE, col.regions = rwb,rug=TRUE,train=RF_train)
**3-D surface**
pdp3 <- plotPartial(pd, levelplot = FALSE, zlab = "C_lit", screen = list(z = -20, x = -80))
**plot(pdp3)**
tiff("./path/3Dpdp_Var1_Var2__LitterSOC.tiff", height = 3, width = 4, units = 'in', res=300)
dev.off()
tiff("./path/grid_pdp_Var1_Var2_LitterSOC.tiff", height = 3, width = 6, units = 'in', res=300)
grid.arrange(pdp1, pdp2, pdp3,ncol = 3)
dev.off()
**-------------------------------------------For other variables**
p1d <- pdp::partial(fit_RF, pred.var = c(var1, var3),train=RF_train, rug=TRUE)
**Default PDP**
pdp11<- plotPartial(p1d)
**Add contour lines and use a different color palette**
rwb <- colorRampPalette(c("red", "white", "blue"))
pdp12 <- plotPartial(p1d, contour = TRUE, col.regions = rwb,rug=TRUE,train=RF_train)
**3-D surface**
pdp13 <- plotPartial(p1d, levelplot = FALSE, zlab = "C_lit", colorkey = FALSE,
            screen = list(z = -40, x = -80))
tiff("./path/3Dpdp_Var1_Var3CVRF_LitterSOC.tiff", height = 3, width = 4, units = 'in', res=300)
plot(pdp13)
dev.off()
tiff("./path/grid_pdp_Var1_Var3CVRF_LitterSOC.tiff", height = 3, width = 6, units = 'in', res=300)
grid.arrange(pdp11, pdp12, pdp13,ncol = 3)
dev.off()

**--------------------------------------------------------------------------------------------------------------------**
**-----------------Prediction interval coverage probability --------------------------------------------------------**
**--------------------------------------------------------------------------------------------------------------------**
C_lit.quantRF <- quantregForest(x = RF_train[,2:(ncol(RF_train))],
                    y = RF_train$C_lit, mtry=RF_tune_Table$mtry,
                    ntree=500)
**predict 0.01, 0.02,..., 0.99 quantiles for validation data**
C_lit.pred.distribution <- predict(C_lit.quantRF,
                    newdata = RF_test,
                    what = seq(0.01, 0.99, by = 0.01))
**Coverage probabilities plot**
**create sequence of nominal probabilities**
ss <- seq(0,1,0.01)
**compute coverage for sequence**
t.prop.inside <- sapply(ss, function(ii){
  boot.quantile <-  t( apply(C_lit.pred.distribution, 1, quantile,
                    probs = c(0,ii) ) )[,2]
  return(sum(boot.quantile <= RF_test$C_lit)/nrow(RF_test))
})
plot.data<-as.data.frame(cbind(ss,t.prop.inside[length(ss):1]))
names(plot.data)[2]<-"Int"
plot<-ggplot(plot.data, aes(x = ss, y = Int))+
  geom_line() +
  geom_abline(color="red",linetype="dashed", size=0.3)+
  labs( x="Confidence level",y = "Coverage probabilities")+
  theme(
    axis.title.x = element_blank(), axis.title.y= element_blank(),
    panel.background = element_rect(fill = "white", colour = "#f7f9fb",size = 2, linetype = "solid"),
    axis.text=element_text(size=6, face="bold"),axis.title=element_text(size=12),#,face="bold"
    plot.background = element_rect(fill = "white")
  )
```

---

## Author Comment (AC5) · 24 Feb 2021

See attachement with Figure 6 and Table 6 inserted

Please also note the supplement to this comment:
https://soil.copernicus.org/preprints/soil-2020-75/soil-2020-75-AC5-supplement.pdf

---

## Author Comment (AC6) · 1 Mar 2021

Dear Reviewer,

Please find enclosed the revision of our manuscript "Predicting the spatial distribution of soil organic carbon stock in Swedish forests using remotely sensed and site-specific covariates" by Hounkpatin et al. We thank you for the very competent review of our paper and the productive comments towards the improvement of our manuscript. We agree with almost all of them and revised the manuscript accordingly.

We hope that our paper is now acceptable for publication in SOIL.

Yours sincerely,

**Ozias Hounkpatin**

Answers to reviewer

The manuscript shows the spatial prediction of carbon stocks in forests of Sweden for tow compartments (organic layer and mineral soil 0-50 cm depth) and the sum of the two. Random forest with parameter tuning and feature selection is used and prediction intervals are computed by quantile regression forest. Different model conïňAguration is shown along two axes: 1) The data is split into subregions and separate model calibrations are presented. The results are then compared to a model covering the complete area. 2) Different subsets of covariates as model input are formed according to their origin. Covariates only available at the observed sites and geodata available for whole Sweden were tested separately. In addition models combining all covariates were ïňĄtted and compared to the other two versions.

The methods are fully described and the results thoroughly discussed. The prediction intervals are validated with data not used to create the models which is so far rarely seen. Moreover, the distribution of the actual data is compared to the distribution of all predicted pixels. The manuscript is well written and the ïňĄgures mostly are well prepared. Well done, Ozias!

However, I share the concerns of referee #1:

1. On page 6 lines 227–232 the data splitting approach is explained. I do not fully understand how it was actually done as I got puzzled by the last sentence. Were the validation data the same data points for the global and the local models (last sentence) or were they independently chosen (rest of this paragraph)? Nevertheless, the validation statistics should be computed on the exact same validation dataset. Especially the measure R2 is very sensitive to different values of observed data. This means for
the global model three local sets of validation statistics should be computed on the same data points. Of course overall statistics can also be presented, but should not be compared to the local statistics, because the standard deviation of the total vector of observed values might be larger. To achieve such a validation the 20-80% splitting should be done as stratiinAed random sampling from the data points with the subareas as strata (which maybe already has been done). Author's Response: Thanks for your enquiry. The error metrics were actually computed based on the same validation set for both global and local models, this to avoid comparison bias. Table 3 actually shows the error metrics for the global models when the independent validation set for the (different strata) North, Central and South Sweden were aggregated. Table 4 shows the specific error metrics for the local models against their respective validation set. Figure 2 presents the error metrics results of each global model for the three subareas along with corresponding local models (unfortunately repeating error metrics of local models from Table 4 for the independent validation. To ensure clarity, we provided more details in the section (see lines 224 - 231) related to the data splitting procedure. Manuscript lines (see lines 224 - 231): The RF models built on data covering the whole area of Sweden are hereafter called "global models". The RF models created for each of the subareas are hereafter reported as "local models". Considering the subareas as strata, the local models were built by randomly splitting the local datasets into calibration (80%) and validation (20%) subset. The training set of the global models was constituted by merging the 80% split of the strata dataset. To avoid comparison bias, both the global and local models were evaluated on the same validation set. Consequently, each global model was evaluated again three validation subset separately corresponding each to the 20% split of the Northern, Central and Southern local dataset. The merged 20% split of these local datasets was then used as validation set at national scale. We trained both global and local models on the calibration subset using tenfold cross validation with 5 repetitions using the R "caret" package (Kuhn, 2015).

2. The covariates only available at the observed locations cannot be used to create

**SOILD**
prediction at locations without any observation. Hence, they are not useful to create area-wide maps unless one creates maps of these site speciīňĄc characteristics beforehand. The same geodata would be used to create these predictions. Such a twostep prediction approach (some sort of a random forest co-kriging) is only meaningful if 1) there are more observations of this intermediate response (site characteristic) than for the ĩňĄnal target response (SOC stock) which I understand is not the case for the current data set; 2) the intermediate response results for some reason in much better model performance and predictive accuracy. The current study neglects to consider errors of the needed spatial prediction of the site characteristic covariates. The error of the spatial prediction might possibly be as high as to render these covariates useless. At least, the interpretation should put this clearly into perspective.

Author's Response: We share the concern of the reviewer that all covariates should actually be maps for DSM of the whole area while the site specific data were only observations. Initially, the focus was mainly evaluating the geodata we now have versus site specific variables and not necessarily making a map. We made the map based on the geodata to give a visual trend of the distribution of the SOC stock. We also agree that the current study neglects to consider mapping uncertainties from potential maps of site specific variables which might actually translate into lower prediction accuracies of SOC stock compared to models based on their observation data. Our interpretation failed to put this clearly into perspective and we have now created a section discussing this issue. To elaborate more on this issue in that section, we made some preliminary maps (which need further improvements because of low kappa values) of the site specific variables using Random Forest and additional dataset from the inventory data. These maps were then used as covariates to predict SOC stock.

Manuscript lines (see lines 585 - 615): see section 4.4 Implication and limitations of the study DSM relies on existing maps for building regression models and ultimately prediction for mapping. The quality and accuracy of predictions depend as discussed earlier on choosing the most relevant covariates in relation to the target to be predicted.

SOILD
The present study revealed that covariates which were available as maps did contribute to the MDA, but site characteristics were more prominent in relation to the SOC stock in Sweden. This might suggest that mapping these variables that were more decisive for SOC stock prediction and including them as covariates for mapping may improve accuracy. Since the primary focus of the present study was mainly to evaluate the GoC data versus the SSC data at different scale of modelling and not primarily for making a map, the observed data of the latter were used. However, since only the observed data of the site characteristics were considered, this study fails to consider that the mapping of these site characteristics would involve modelling errors and the propagation of these errors into the final maps of these site variables might actually reduce their predictive power. For completeness, we carried out a preliminary mapping of the site characteristics (SI-11) using additional soil inventory data, random forest and the GoC as predictors. These mapped site characteristics (mSSC) were then used as covariate for predicting the SOC stocks.

Table 6 presents the error metrics after independent validation for both local and global models along with the percentage margin of the RMSE in relation of the models based on GoC. First, the local mSSC based models still recorded lower RMSE as compared to global mSSC models. Compared to the GoC models, the overall positive percentage margin of the RMSE for the independent validations indicated that the mSSC models recorded the lowest RMSE. However, when assessing the RMSE margin between the global models of GoC and SSC, negative percentages were mainly recorded for the Northern Sweden independently of the depth. This indicated that the mSSC based global models were less accurate at predicting SOC stock locally in the Northern Sweden while the mSSC based local model did present a better prediction for Northern Sweden (SI-12) as compared to the pattern observed with the GoC maps. However, the uncertainty distribution was of similar magnitude (SI-12) as those observed for the maps based on GoC probably due to error propagations as these covariates were used

SOILD
to make the site specific characteristics maps. This suggests that the SSC should still be supplemented for improvement at this stage with other covariates different from the GoC such as multi-temporal spectral (e.g. normalized difference vegetation index) data that are able to capture vegetation dynamic in forests. Notwithstanding the fact that was obviously error propagation, the study of which was beyond the scope of this study, the preceding results tend to confirm the potential of the high resolution maps of the site characteristics to contribute to the improvement of SOC stock prediction as compared to using only the GoC data. Given that the preliminary mappings of the SSC recorded low kappa (0.17 – 0.48) values (SI-11) at this stage further improvements are still necessary to improve their predictive ability and associated coefficient of variations.

Table 6: Cross-validation, independent validation of the global and local random forest models based on the mapped site specific covariates compared to models based on observed site specific covariates and grouped of covariates

Specific comments

Maybe consider using "covariates" instead of "variables" as in the rest of the text. It would be more specific. Author's Response: "covariates" was considered Manuscript: Predicting the spatial distribution of soil organic carbon stock in Swedish forests using remotely sensed and site-specific covariates

Please add the reference depth of SOC stocks. Mineral soil = 0-50 cm (only), otherwise one would think down to bedrock or profile depth of usually 1-1.5 m

Author's Response: We added the depth to the mineral soil. Manuscript (line 28 - 31): The most important covariates that influence the humus layer, mineral soil (0 - 50 cm) and total SOC stock were related to the site characteristic covariates and include the soil moisture class, vegetation type, soil type and soil texture.

Not all of the mentioned geodata are remote sensing data, e.g. the geological map. Better use a broader formulation. Author's Response: We replace "remote sensing by SOILD
covariates" Manuscript (line 73 - 74): These studies and many others rely mostly on covariates existing as maps while survey data which present site specific information are left out during modelling.

Should "c." be "ca."? Author's Response: yes it is ca. It has been replaced. Manuscript (line 99 - 100): Soil samples are collected in a subset of the plots with humus sampling on ca. 10 000 plots and mineral soil sampling on ca. 4500 plots (Stendahl et al., 2017).

This does not include the O horizon, right? Maybe reformulate "below the O horizon down to 30 cm". Author's Response: Thanks for the suggestion. We reformulated to "below the O horizon down to 30 cm Manuscript (line 103 - 104): Therefore the content of inorganic carbon in mineral soil is considered negligible in the study area. Humus layer volumetric samples are taken using a soil core (core diameter 10 cm) below the O horizon down to 30 cm depth.

if available, better cite a data documentation that appears in the references list. Author's Response: Citation has been added. Manuscript (line 120 - 122): Topographic covariates were computed from high-resolution digital elevation models (DEM) derived from Light Detection And Ranging (LiDAR) produced by the Swedish National Mapping Agency. It was originally created with 2-m spatial resolution (Dowling et al., 2013).

Please give the algorithm of the aggregation procedure. Bilinear interpolation? The software is usually secondary, if the algorithm becomes clear. Author's Response: The bilinear interpolation algorithm was provided. Manuscript (line 122 - 124): However, the initial DEM was resampled in ArcGIS 10 software package using the aggregation procedure with bilinear interpolation to a final resolution of 10 m  $\times$  10 m which is reasonable for the data considered in the present study.

Please cite, or give used algorithms and versions. (maybe add in covariate table). Author's Response: Citation has been added. Manuscript (line 124 - 125): The topographical covariates were computed using the SAGA GIS software (Conrad et al.,
2015).

Please cite in reference list as a web source. Author's Response: Citation added. Manuscript (line 127 - 128): The depth to groundwater was obtained from the Swedish Forest Agency (SGU, 2018) and computes the difference in elevation in relation to surrounding cells following the vertical in Compatible Comp

Please add citation here. Or was this done in this study? Author's Response: This was done in this study Manuscript (line 141 - 143): The resulting gamma-ray data as well as the geochemical data were interpolated in this study into maps either by ordinary kriging or inverse distance weighing when geostatistic assumption such as normal distribution were not met.

Please check again this function notation. The superscript 1 is likely an indicator function, usually used without superscript. Moreover, "j" is not defined in the text. Author's Response: Thanks for the observation. We have checked and reproduce it as mentioned in Meinshausen (2006). j is also defined. Manuscript (line 182 - 185): The weight vector (Meinshausen, 2006) w\_i is defined as follows: w\_i (x, $\theta$ )= 1\_{x\_i II} R\_(I(x, $\theta$ )) } /(#{j:x\_j II} R\_(I(x, $\theta$ )) } ) (2)

Please indicate which values where tested for mtry in the grid search. Was ist 1:p with p: number of covariates? Author's Response: We provided the values tested: 2: n, with n the number of covariates. Manuscript (line 196 - 198): To reduce computational load, the ntree was set at 500 while the mtry was tuned using the grid search (2: p, with p the number of covariates) method in the R "caret" package (Kuhn, 2015).

Please give a citation. e.g. Hastie et al., 2011. Author's Response: Ciatation added. Manuscript (line 199 - 198): The importance of each input predictor can be assessed by the RF based on the mean decrease accuracy (MDA) (Hastie et al., 2011).

Were 10 m overlap really enough to allow for a smooth transition? This is just one pixel... Author's Response: Thanks for notification. This was a mistake. We corrected

SOILD
to 4km which was actually used. Manuscript (line 208 - 210): A buffer of 4 km was considered for the shapefiles of each subarea to create overlapping zones which ensured smooth transition while merging by averaging the SOC stock values within these shared units.

Please specify what kind of algorithm and maybe also the function name you used for de-correlation. Author's Response: The function has been added. We use the Pearson's correlation. Manuscript (line 220 - 222): Moreover, to reduce computation time while keeping relatively the same level of accuracy, we (1) used feature preprocessing capabilities implemented in the caret package (Kuhn, 2017) of R to remove highly correlated (Pearson's correlation) expressions using a cutoff point of 0.80...

maybe use "group of covariates" Author's Response: Thanks for the suggestion. We adopted "group of covariates" throughout the document.

Please specify: How was the splitting done into 20 vs. 80%? Simple random sample? Author's Response: Yes a simple random split was carried out. This has now been specified. Manuscript (line 229 - 230): Considering the subareas as strata, the local models were built by randomly splitting the local datasets into calibration (80%) and validation (20%) subset.

Bias is reported in the table, but definition is missing. Author's Response: Thanks for notifying. Bias has now been defined. Manuscript (line 240 - 245): Bias=  $\mu$ \_pred- $\mu$ \_obs

squared correlation coefficient, but rather the formula of the type: R2 = 1 - sum(P\_i - O\_i) / sum(O\_i - mean(O)). Caret has an implementation, see also help site for the formula: R2(p,o, form = "traditional") Author's Response: Thanks for notifying. This has now been corrected. Manuscript (line 240 - 245): R^2 = 1- ( $\sum_{i=1}^{\infty} \frac{1}{2} \frac$

SOILD
Response: In our study, the lines follow the 1:1 line quite well and we also thought that other upend end make no difference in this case.

Just to be precise ;-). Please consider using the peer-reviewed paper of this work. Nussbaum et al. 2014 (https://gmd.copernicus.org/articles/7/1197/2014/). We used an external-drift kriging model (that includes a linear regression of course). Then, the data was not only form the Swiss National Forest Inventory. Considerdropping this information as it is of no importance to the discussion. Author's Response: We have removed the reference.

A bit hard to read. Consider reformulation. Author's Response: We have reformulated that section and hope that it is now better to read. Manuscript (line 419 - 430): Low explained variances in predictive modelling could be related to different factors (Nelson et al., 2011). For example, the omission of key covariates with greater explanatory power or conversely using non-essential covariates with very low explanatory power which only increase the prediction error variance. Omitting key covariates in relation to SOC stock for forest ecosystem in the present study is less likely since covariates considered in this study well represent the surrogates for soil forming factors considered in the SCORPAN equation defined by McBratney et al. (2003). In addition, the removal of redundant and non-informative covariates was carried out via dimension reduction with the exclusion of highly correlated covariates and elimination of some others via recursive feature elimination. However, the Pearson correlations (min = 0, max = 0.28) between covariates and the different SOC stock were found to be poor though significant for most of the predictors (Table 5, SI 2-4). This could be expected because the data cover a wide range of different site conditions, soil types and parent materials.

Figure SI5 Author's Response: Correction made. Manuscript (line 436): On the other hand, applying geostatistical approaches (SI-5) for the humus layer...

Computing variograms from the response directly might reveal spatial autocorrelation. However, I do not recommend to do that as putting as much of the correlation into
the regression part e.g. with random forest is to be preferred. Better reformulate the text. Author's Response: We have reformulated focusing on the response directly. Manuscript (line 435 - 438): On the other hand, applying geostatistical approaches (SI-5) for the humus layer, mineral soil and total SOC stock revealed very low spatial autocorrelation for the different SOC stock suggesting that the structure of these SOC data is having a shorter range than the sampling interval.

Is it not the other way around, overestimation of small values and underestimation of the large values? RF has the tendency to predict the mean if correlation of response and covariate is weak. Hence, the extremes are not well predicted. Author's Response: Yes generally there is overestimation of small values and underestimation of the large values. We were commenting on the trend which can be seen in Fig. 3. We mainly observed an underestimation of the low and high values in our context.

Conclusion difficult, without considering the error of such predicted maps. Author's Response: We now provided error metrics based from RF models based on preliminary mapping of the site specific covariates which showed that they performed generally better than the group of covariates. Manuscript (line 580 - 607): Section on 4.4 Implication and limitations of the study DSM relies on existing maps for building regression models and ultimately prediction for mapping. The quality and accuracy of predictions depend as discussed earlier on choosing the most relevant covariates in relation to the target to be predicted. The present study revealed that covariates which were available as maps did contribute to the MDA, but site characteristics were more prominent in relation to the SOC stock in Sweden. This might suggest that mapping these variables that were more decisive for SOC stock prediction and including them as covariates for mapping may improve accuracy. Since the primary focus of the present study was mainly to evaluate the GoC data versus the SSC data at different scale of modelling and not primarily for making a map, the observed data of the latter were used. However, since only the observed data of the site characteristics were considered, this study fails to consider that the mapping of these site characteristics would involve modelling
errors and the propagation of these errors into the final maps of these site variables might actually reduce their predictive power. For completeness, we carried out a preliminary mapping of the site characteristics (SI 11) using additional soil inventory data, random forest and the GoC as predictors. These mapped site characteristics (mSSC) were then used as covariate for predicting the SOC stocks. Table 6 presents the error metrics after independent validation for both local and global models along with the percentage margin of the RMSE in relation of the models based on GoC. First, the local mSSC based models still recorded lower RMSE as compared to global mSSC models. Compared to the GoC models, the overall positive percentage margin of the RMSE for the independent validations indicated that the mSSC models recorded the lowest RMSE. However, when assessing the RMSE margin between the global models of GoC and SSC, negative percentages were mainly recorded for the Northern Sweden independently of the depth. This indicated that the mSSC based global models were less accurate at predicting SOC stock locally in the Northern Sweden while the mSSC based local model did present a better prediction for Northern Sweden for the humus layer and mineral SOC stock. Notwithstanding the fact that there could be error propagation, the study of which was beyond the scope of this study, the preceding results tend to confirm the potential of the high resolution maps of the site characteristics to contribute to the improvement of SOC stock prediction as compared to using only the GoC data. Given that the preliminary mappings of the SSC recorded low kappa values (SI11) at this stage further improvements are still necessary to improve their predictive ability. maybe use "open source" Author's Response: Thanks for suggesting. "open source" adopted. Manuscript (line 579 - 583): DSM approach in the present study allows: flexibility in future improvement upon acquisition of new covariates or data point, repeatability in modelling with the application of the same modelling principles using open source software (e.g. R) and capitalizing on multi-source information (topography, site characteristics, forest data, gamma radiometry and geochemical data).

In the correlation matrix in the supplement (Figure SI2-4) the aspect is given. As it is 0-360 degrees it is not meaningful as a covariate. Consider dropping it everywhere.
Author's Response: We dropped aspect in mapping the site covariates. In most cases, it was also dropped by RFE when all the variables were considered.

Please check again. These skewness values seem not possible from the other information. Author's Response: We are sorry for this mistake. The n values were repeated twice. We have now corrected this in the table.

What kind of test was performed? To be complete it would be good practice to report the test statistic Author's Response: Thanks for pointing out this. We have now changed the title of the table which take into account the test. Manuscript: Table 5: Random Forest variable importance for the global and local models for the humus layer, mineral soil and total SOC stock with their associated Pearson's coefficient of correlation with the covariates (values in bracket, \*:  $p \le 0.05$ ,  $p \le 0.05$ ;\*\*:  $p \le 0.01$ ;\*\*\*:  $p \le 0.001$ )

Please specify how the R2 and the RMSE were computed. Cross-validation independent data set obtained by data splitting? Author's Response: We have now specified that the r2 and RMSE were based on the independent data. Manuscript (line 950): Figure 2: Local and global models ranked by decreasing RMSE per subareas and category of covariates along with corresponding R2 based on the independent validation dataset. A: litter layer, B: mineral soil layer, C: total soil layer, SSC: site specific covariates, GoC: Group of covariates, allC: all covariates

These maps are very difficult to read. Consider either adequate generalization of the 10 m pixels for the display scale or showing but a section. I would actually be interested if there are artefacts at the border of the subareas like sudden change in values that are hard to explain. Such artefacts might be a reason against using local models even if their model performance is better. Author's Response: Thanks for pointing this out. It is actually challenging to have a color palette that display distinctly the pattern over the whole Sweden. We have now used another color palette for the SOC stocks and hope that visualization has improved. Visualizing further the maps,
Please also note the supplement to this comment: https://soil.copernicus.org/preprints/soil-2020-75/soil-2020-75-AC6-supplement.pdf

we did not observe any sudden change in values that would raise particular attention as the pattern at the border showed similar pattern for some location inside Sweden.

```
SOILD
```

**Supplement:**

Table 6: Cross-validation, independent validation of the global and local random forest models based on the mapped site specific covariates compared to models based on observed site specific covariates and grouped of covariates

| | | | RMSE (t C/ha) | $R^2$ | ΔRMSE GoC-mSSC (%) |
|---|---|---|---|---|---|
| | | | Independant validation | | |
| Global models | Humus layer | All Sweden | 20.87 | 0.27 | 5.57 |
| | Mineral soil | All Sweden | 26.57 | 0.28 | 13.44 |
| | Total | All Sweden | 34.35 | 0.28 | 10.55 |
| Local models | Humus layer | North | 19.22 | 0.22 | 2.46 |
| | | center | 17.56 | 0.16 | 2.47 |
| | | South | 24.81 | 0.31 | 6.37 |
| | Mineral soil | North | 20.21 | 0.22 | 4.68 |
| | | center | 22.89 | 0.09 | 20.23 |
| | | South | 32.11 | 0.29 | 11.04 |
| | Total | North | 29.82 | 0.28 | -0.74 |
| | | center | 27.36 | 0.09 | 22.06 |
| | | South | 41.79 | 0.23 | 6.30 |
| | | | Error metrics by Global models for local validation set | | |
| | Humus layer | North | 20.28 | 0.14 | -1.91 |
| | | center | 18.02 | 0.13 | -1.24 |
| | | South | 25.27 | 0.27 | 4.64 |
| | Mineral soil | North | 22.72 | 0.10 | -4.70 |
| | | center | 23.14 | 0.05 | 20.75 |
| | | South | 34.96 | 0.15 | 2.35 |
| | Total | North | 40.70 | 0.01 | -43.56 |
| | | center | 30.00 | 0.07 | 13.49 |
| | | South | 43.18 | 0.18 | 0.74 |

ΔRMSE GoC-mSSC (%): percentage estimate of the difference between the root mean square error of models based on group of covariates and mapped site specific covariates, negative values: models with mapped site specific covariates present higher root mean square error as compared to models based on either observed site specific variable or group of covariates, positive values : models with mapped site specific covariates present lower root mean square error as compared to models based on either observed site

[Figure]

Figure 6: Mean SOC stock prediction and Prediction uncertainties of the spatial distribution of the humus layer, mineral soil and total SOC stock based on the group of covariates

**Predicting the spatial distribution of soil organic carbon stock in Swedish boreal forest using remotely sensed and site-specific covariates**

**Kpade O. L. Hounkpatin[a1], Johan Stendahl[a], Mattias Lundblad[a] , Erik Karltun[a],**

[a] Department of Soil and Environment, Swedish University of Agricultural Sciences, P.O. Box 7014, SE-75007, Uppsala, Sweden

**Supplementary Information (SI)**
* * *
[1] Correspondence : ozias.hounkpatin@slu.se

Figure SI-1A: Coverage of 90 %-prediction intervals computed for the local and global random forest models for the humus layer, mineral soil and total SOC stock based on the site specific covariate models

[Figure]

Figure SI-1B: Coverage of 90 %-prediction intervals computed for the local and global random forest models for the humus layer, mineral soil and total SOC stock based on the group of covariate models

[Figure]

Figure SI-1C: Coverage of 90 %-prediction intervals computed for the local and global random forest models for the humus layer, mineral soil and total SOC stock based on all covariate models

[Figure]

Figure SI-2: Correlation matrix of all variables with the humus layer SOC stock dataset over Sweden. Refer to table 1 for the definition of the abbreviation

[Figure]

Figure SI-3: Correlation matrix of all variables with the mineral soil SOC stock dataset over Sweden. Refer to table 1 for the definition of the abbreviation

[Figure]

Figure SI-4: Correlation matrix of all variables with the total soil SOC stock dataset over Sweden. Refer to table 1 for the definition of the abbreviation

[Figure]

Figure SI-5: Variogram for the SOC stock and regression residuals of the local and global Random Forest models for the humus layer, mineral soil and total SOC stock. Log: log transformation, sqrt: square root transformation, Sph: spherical model, Ste: Matern, M. Stein's parameterization.

[Figure]

Figure SI-6: Box-plot for humus layer SOC stock by soil moisture, soil type, vegetation type, parent material and soil texture. Lines within the boxes give the median, red point within the boxes is the mean, boxes the 25th and 75th percentile (Refer to Table 2 for definition of the classes of each variable)

[Figure]

Figure SI-7: Box-plot for the mineral soil SOC stock by soil moisture, soil type, vegetation type, soil texture and parent material. Lines within the boxes give the median, red point within the boxes is the mean, boxes the 25th and 75th percentile (Refer to Table 2 for definition of the classes of each variable)

[Figure]

Figure SI-8: Box-plot for total SOC stock by soil moisture, soil type, vegetation type, soil texture and parent material. Lines within the boxes give the median, red point within the boxes is the mean, boxes the 25th and 75th percentile (Refer to Table 2 for definition of the classes of each variable)

[Figure]

Figure SI-9: Partial dependence plots. Refer to Table 1 for the definition of the abbreviation and numbers for the classes of the variables.

[Figure]

A: Partial plot between the humus layer SOC stock,
soil moisture and vegetation type
(RF Global model with all variables)

B: Partial plot between the humus layer SOC stock,
soil moisture and Northing
(RF Global model with all variables)

C: Partial plot between the mineral SOC stock,
Northing and soil type
(RF Global model with site specific variables)

D: Partial plot between the mineral SOC stock,
Easting and soil type
(RF Local model with site specific variables)

E: Partial plot between the total SOC stock,
soil moisture and soil type
(RF Global model all variables)

F: Partial plot between the total SOC stock,
precipitation and soil moisture
(RF Global model all variables)

Figure SI-10: Partial dependence plots between SOC stock, northing, temperature and precipitation.

A

[Figure]

A: Partial plot between the humus layer
SOC stock, Northing and Temperature
(RF Global model with all variables)

B

B: Partial plot between the humus layer
SOC stock, Northing and precipitation
(RF Global model with all variables)

C

[Figure]

C: Partial plot between the mineral SOC stock,
Northing and Temperature
(RF Global model with all variables)

D

[Figure]

D: Partial plot between the mineral SOC stock,
Northing and precipitation
(RF Global model with all variables)

E

[Figure]

E: Partial plot between the total SOC stock,
Northing and Temperature
(RF Global model with all variables)

F

[Figure]

F: Partial plot between the total SOC stock,
Northing and precipitation
(RF Global model with all variables)

Figure SI-11: Maps of the site specific covariates

[Figure]

[Figure]

**Figure SI-12: SOC prediction and uncertainties based on mapped site-specific covariates**

[Figure]

**Asset SI-13: R code for SOC stock prediction**

```
**-----------------------------------Script R SOC stock modelling ----------------------------------**
###################################################################
**Author : Kpade Ozias L. Hounkpatin**
**Paper : Predicting the spatial distribution of soil organic carbon stock in Swedish forests using remotely**
**sensed and site-specific variables**
**Code: The following code applies for litter layer, mineral soil and total soil carbon stock**

**Load necessary Packages**
library(quantregForest)
library(rgdal)
library(raster)
library(spatial.tools)
library(caret)
library(randomForest)
library(imputeTS)
library(stringr)
library(sp)
library(xgboost)
library(plyr)
library(dplyr)
library(parallel)
library(tidyverse)
library(Metrics)
library(ithir)
library(gridExtra)
library(doParallel)
library(pastecs)
library(utils)
library(broom) # for tidy()
library(knitr) # for kable()
library(stargazer)
library(pdp)
**Set options**
options(digits = 2,scipen=999)
**Set my working directory**
setwd("yourpath")

**--------------------------------------------------------------------------------------------------------------**
**-----Load data, Deal with NA values & remove problematic column-----------------------------------------------**
**--------------------------------------------------------------------------------------------------------------**
**---Load the table**
litterdata<- read.table("./path/litterdata.txt", header = T, sep = "\t")
**-- Change data class for site specific variables**
**cor.train <- lapply(cor.train[:], as.factor), #insert where factor variables start and end in [:]**
**cor.test<- lapply(cor.test[:], as.factor), #insert where factor variables start and end in [:]**
**Transform into spatial soildata frame**
coordinates(litterdata) <- ~EastC+NorthC
**Project the soildata into EPSG : 3006**
crs<-CRS("+init=epsg:3006")
proj4string(litterdata) <- crs
**-Load the predictors**
rlist<-list.files("path", pattern="tif$", full.names=TRUE)#Load in the covariates
covStack <- stack(rlist)#Stack all the covariates
**Extract multi value from covariates---TRAIN**
litterdata_cov<- raster::extract(covStack, litterdata,sp = 1, method = "simple")
```

```
**-----------------------------------------------------------------------------------------------------------------------**
**-------------------------train set, Validation set---------------------------------------------------------------------**
**-----------------------------------------------------------------------------------------------------------------------**
**Specify the percentage that will be used for training. the remaining will be used for testing**
set.seed(1)
split <- 0.8
**Training samples  for soil moisture**
trainIndex <- createDataPartition(c(litterdata_cov["C_lit"], recursive=T), p=split, list = F)
pcDat<- liptData[trainIndex,] # Training set
ptDat<-liptData[-trainIndex,] # Validation set

**-----------------------------------------------------------------------------------------------------------------------**
**------------------CHECKING HIGHLY CORRELATED COVARIATES----------------------------------------------------------------**
**-----------------------------------------------------------------------------------------------------------------------**
**Only for numeric variables**
**corstarsl: https://github.com/kyuni22/ksmv/blob/master/functions/corstarsl.R**
corstarsl <- function(x){
  require(Hmisc)
  x <- as.matrix(x)
  R <- rcorr(x)$r
  p <- rcorr(x)$P
  ## define notions for significance levels; spacing is important.
  mystars <- ifelse(p < .001, "***", ifelse(p < .01, "** ", ifelse(p < .05, "* ", " ")))
  ## trunctuate the matrix that holds the correlations to two decimal
  R <- format(round(cbind(rep(-1.11, ncol(x)), R), 2))[,-1]
  ## build a new matrix that includes the correlations with their apropriate stars
  Rnew <- matrix(paste(R, mystars, sep=""), ncol=ncol(x))
  diag(Rnew) <- paste(diag(R), " ", sep="")
  rownames(Rnew) <- colnames(x)
  colnames(Rnew) <- paste(colnames(x), "", sep="")
  ## remove upper triangle
  Rnew <- as.matrix(Rnew)
  Rnew[upper.tri(Rnew, diag = TRUE)] <- ""
  Rnew <- as.data.frame(Rnew)
  ## remove last column and return the matrix (which is now a data frame)
  Rnew <- cbind(Rnew[1:length(Rnew)-1])
  return(Rnew)
}
corr.data<-corstarsl(pcDat[sapply(pcDat,is.numeric)])
**Save correlation data**
write.table(corr.data, file = str_c("./path/litterSOC/RFEdata_LitterSOC.txt"), sep = "\t",row.names=F)
descrCorr <- cor(pcDat[sapply(pcDat,is.numeric)])
highcorr <- caret::findCorrelation(descrCorr, cutoff = 0.80,names = TRUE)
cor.train<-pcDat %>% dplyr::select(-highcorr)
cor.test<-ptDat%>% dplyr::select(-highcorr)

**-----------------------------------------------------------------------------------------------------------------------**
**------------------Feature Engineering_RFE------------------------------------------------------------------------------**
**-----------------------------------------------------------------------------------------------------------------------**
xtrain <- cor.train[, 2:ncol(cor.train)]
xtest <-cor.test[, 2:ncol(cor.test)]
ytrain <- cor.train[, 1]
ytest <- cor.test[, 1]
rfe.ctrl = rfeControl(functions = rfFuncs,
              method = "cv",
              number = 50,
              allowParallel = TRUE,
              verbose = TRUE)
set.seed(1)
RF_rfe<- rfe(x=xtrain, y=ytrain,rfeControl = rfe.ctrl)
```

```
modelFile <-paste("./path",sep = "", "rfemodel_LitterSOC", ".rds")
saveRDS(object = RF_rfe, file = modelFile)
**Get names of the eliminated variables**
RFNotsel<-xtrain %>% dplyr::select(-one_of(predictors(RF_rfe)[1:length(predictors(RF_rfe))]))
**Get final rfe train dataset**
se_RF_train<-xtrain %>% dplyr::select(one_of(predictors(RF_rfe)[1:length(predictors(RF_rfe))]))
RF_train<-cbind(cor.train$C_lit,se_RF_train)
names(RF_train)[1] <- "C_lit" #Rename First column
write.table(RF_train, file = str_c("./path/train_rfe_LitterSOC.txt"), sep = "\t",row.names=F)
**Get final rfe test dataset**
se_RF_test<-xtest %>% dplyr::select(one_of(predictors(RF_rfe)[1:length(predictors(RF_rfe))]))
RF_test<-cbind(cor.test$C_lit,se_RF_test)
names(RF_test)[1] <- "C_lit" #Rename First column
write.table(RF_test, file = str_c("./path/test_rfe_LitterSOC.txt"), sep = "\t",row.names=F)

**-------------------------------------------------------------------------------------------------------------**
**-------------------TUNE RF model-----------------------------------------------------------------------------**
**-------------------------------------------------------------------------------------------------------------**
RFgrid <- expand.grid(.mtry = 2:ncol(RF_train))
control <- trainControl(method="repeatedcv", number=10,
                repeats = 3,
                verboseIter = TRUE,
                returnData = "FALSE",
                returnResamp = "all",
                allowParallel = TRUE)
set.seed(1)
Tune_RF<- train(x=xtrain,y=ytrain,method = "rf",
         trControl=control,tuneGrid = RFgrid)
**Save the model**
modelFile <-paste("./path",sep = "", "tunemodel_LitterSOC", ".rds")
saveRDS(object = Tune_RF, file = modelFile)
RF_tune_Table<-as.data.frame(Tune_RF$bestTune)

**-------------------------------------------------------------------------------------------------------------**
**------------------CROSS VALIDATION with RF-------------------------------------------------------------------**
**-------------------------------------------------------------------------------------------------------------**
control <- trainControl(method="repeatedcv", number=10,
                repeats=5,
                verboseIter = TRUE,
                returnData = "FALSE",
                returnResamp = "all",
                allowParallel = TRUE)
RFgrid<-expand.grid(mtry = RF_tune_Table$mtry)
set.seed(7)
fit_RF <- train(x=RF_train[,2:ncol(RF_train)],
         y=RF_train$C_lit,method = "rf",
         trControl=control,
         tuneGrid = RFgrid,
         importance = TRUE)
modelFile <-paste("./path", sep="","CVmodelRF_LitterSOC",".rds")
saveRDS(object = fit_RF, file = modelFile)
**Get and Save resampling statistics**
CV_RF_Stat.desc<-as.data.frame(stat.desc(fit_RF$resample,basic=FALSE))
write.table(CV_RF_Stat.desc, file = str_c("./path/Stat.desc_LitterSOC.txt"), sep = "\t",row.names=F)
**Predict for external validation**
C_lit.pred.RF<- as.data.frame(predict(fit_RF, xtest))
**Get external validation statistics and save to disk**
obs.C_lit.pred.RF<-as.data.frame(cbind(ytest, C_lit.pred.RF))
names(obs.C_lit.pred.RF)[1]<-"obs"
names(obs.C_lit.pred.RF)[2]<-"pred"
```

```
RF.goof<-as.data.frame(goof(obs.C_lit.pred.RF$obs,obs.C_lit.pred.RF$pred))
**Write table for Results EV for RF**
RF.goof$mae<-mae(obs.C_lit.pred.RF$obs,obs.C_lit.pred.RF$pred)
write.table(RF.goof, file = str_c("./path/goofRF_LitterSOC.txt"), sep = "\t",row.names=F)

**--------------------------------------------------------------------------------------------------------------------------**
**------------------Variable Importance-------------------------------------------------------------------------------------**
**--------------------------------------------------------------------------------------------------------------------------**
VI.RF<-varImp(fit_RF)
RF.VImp<-as.data.frame(VI.RF[1])
names(RF.VImp)[1]<-"VImp.RF"
RF.VImp<-as_tibble(setNames(cbind(rownames(RF.VImp), RF.VImp, row.names = NULL), c("Factors",
"Importance")))
RF.VImp<-as.data.frame(RF.VImp[order(-RF.VImp$Importance),])
write.table(RF.VImp, file = str_c("./path/VarImpCVRF_LitterSOC.txt"), sep = "\t")

**--------------------------------------------------------------------------------------------------------------------------**
**------------------Density Plots-------------------------------------------------------------------------------------------**
**--------------------------------------------------------------------------------------------------------------------------**
RF_SOC.test<-cbind(RF_test$C_lit,C_lit.pred.RF)
RF_SOC.test<-setNames(RF_SOC.test, c("C_lit","RF_pred"))
write.table(RF_SOC.test, file = str_c("path/Obs_PredRF.data_LitterSOC.txt"), sep = "\t",row.names=F)
**RF_density plots with actual and predicted SOC**
RF_density<-RF_SOC.test[,c(1,length(RF_SOC.test))]
RF_density$Type1<-"Actual"
RF_density$Type2<-"PredictedC"
RF_d.soil<-RF_density[,c(1,3)]
RF_d.pred<-RF_density[,c(2,4)]
RF_d.soil<-setNames(RF_d.soil, c("SOC",          "Type"))
RF_d.pred<-setNames(RF_d.pred, c("SOC",        "Type"))
RF_density.data<-(as.data.frame(rbind(RF_d.soil,RF_d.pred)))
RF_predType<-as.data.frame(ddply(RF_density.data, "Type", summarise, rating.pred=mean(SOC)))
write.table(RF_predType, file = str_c("./path/densityRF_predType_LitterSOC.txt"), sep = "\t",row.names=F)
write.table(RF_density.data, file = str_c("./path/densityRF_data_LitterSOC.txt"), sep = "\t",row.names=F)
**Density plots with semi-transparent fill**
N10RF_densityP_lit<-ggplot(RF_density.data, aes(x=SOC, fill=Type))+geom_density(alpha=0.4)+
  geom_vline(data=RF_predType, aes(xintercept=rating.pred,  colour=Type),linetype="dashed")+
        labs(y = "Density", x = "SOC stock",title = "litter SOC stock",
        linetype="dashed", size=1,show.legend=FALSE)+
        scale_color_manual(values=c("#999999", "#E69F00"))+
        scale_y_continuous(breaks=c(0,0.035, 0.065))+
  theme(axis.title.x = element_blank())+
  theme(legend.position = "none")+
  theme(axis.text=element_text(size=9, face="bold"),axis.title=element_text(size=10, face="bold"),#,face="bold"
      plot.title = element_text(size = 14, face = "bold",hjust = 0.5))+
  theme(panel.background = element_rect(fill = "white", colour = "#f7f9fb",size = 2, linetype = "solid"))

**--------------------------------------------------------------------------------------------------------------------------**
**------------------Compute Partial dependance------------------------------------------------------------------------------**
**--------------------------------------------------------------------------------------------------------------------------**
**Compute partial dependence data**
var1<-as.character(RF.VImp["1",1])
var2<-as.character(RF.VImp["2",1])
var3<-as.character(RF.VImp["3",1])
RF_train<-as.data.frame(RF_train)
pd <- pdp::partial(fit_RF, pred.var = c(var1, var2),train=RF_train, rug=TRUE)
**Default PDP**
pdp1 <- plotPartial(pd)
**Add contour lines and use a different color palette**
rwb <- colorRampPalette(c("red", "white", "blue"))
```

```r
pdp2 <- plotPartial(pd, contour = TRUE, col.regions = rwb,rug=TRUE,train=RF_train)
**3-D surface**
pdp3 <- plotPartial(pd, levelplot = FALSE, zlab = "C_lit", screen = list(z = -20, x = -80))
**plot(pdp3)**
tiff("./path/3Dpdp_Var1_Var2__LitterSOC.tiff", height = 3, width = 4, units = 'in', res=300)
dev.off()
tiff("./path/grid_pdp_Var1_Var2_LitterSOC.tiff", height = 3, width = 6, units = 'in', res=300)
grid.arrange(pdp1, pdp2, pdp3,ncol = 3)
dev.off()
**-------------------------------------------For other variables**
p1d <- pdp::partial(fit_RF, pred.var = c(var1, var3),train=RF_train, rug=TRUE)
**Default PDP**
pdp11<- plotPartial(p1d)
**Add contour lines and use a different color palette**
rwb <- colorRampPalette(c("red", "white", "blue"))
pdp12 <- plotPartial(p1d, contour = TRUE, col.regions = rwb,rug=TRUE,train=RF_train)
**3-D surface**
pdp13 <- plotPartial(p1d, levelplot = FALSE, zlab = "C_lit", colorkey = FALSE,
            screen = list(z = -40, x = -80))
tiff("./path/3Dpdp_Var1_Var3CVRF_LitterSOC.tiff", height = 3, width = 4, units = 'in', res=300)
plot(pdp13)
dev.off()
tiff("./path/grid_pdp_Var1_Var3CVRF_LitterSOC.tiff", height = 3, width = 6, units = 'in', res=300)
grid.arrange(pdp11, pdp12, pdp13,ncol = 3)
dev.off()

**--------------------------------------------------------------------------------------------------------------------**
**-----------------Prediction interval coverage probability --------------------------------------------------------**
**--------------------------------------------------------------------------------------------------------------------**
C_lit.quantRF <- quantregForest(x = RF_train[,2:(ncol(RF_train))],
                    y = RF_train$C_lit, mtry=RF_tune_Table$mtry,
                    ntree=500)
**predict 0.01, 0.02,..., 0.99 quantiles for validation data**
C_lit.pred.distribution <- predict(C_lit.quantRF,
                    newdata = RF_test,
                    what = seq(0.01, 0.99, by = 0.01))
**Coverage probabilities plot**
**create sequence of nominal probabilities**
ss <- seq(0,1,0.01)
**compute coverage for sequence**
t.prop.inside <- sapply(ss, function(ii){
  boot.quantile <-  t( apply(C_lit.pred.distribution, 1, quantile,
                    probs = c(0,ii) ) )[,2]
  return(sum(boot.quantile <= RF_test$C_lit)/nrow(RF_test))
})
plot.data<-as.data.frame(cbind(ss,t.prop.inside[length(ss):1]))
names(plot.data)[2]<-"Int"
plot<-ggplot(plot.data, aes(x = ss, y = Int))+
  geom_line() +
  geom_abline(color="red",linetype="dashed", size=0.3)+
  labs( x="Confidence level",y = "Coverage probabilities")+
  theme(
    axis.title.x = element_blank(), axis.title.y= element_blank(),
    panel.background = element_rect(fill = "white", colour = "#f7f9fb",size = 2, linetype = "solid"),
    axis.text=element_text(size=6, face="bold"),axis.title=element_text(size=12),#,face="bold"
    plot.background = element_rect(fill = "white")
  )
```

---

## Author Response (AR1)

Dear Editors,

Please find enclosed the revision of our manuscript "Predicting the spatial distribution of soil organic carbon stock in Swedish forests using remotely sensed and site-specific covariates" by Hounkpatin et al. We thank you for your comments towards the improvement of our manuscript and we provide below answers to the point you raised.

Reviewers 1 & 2 have concerns about the validation and the prediction steps which will need to be addressed.

- Author's Response: We have answered Reviewers 1 & 2 regarding this concern and changed the writing in the manuscript to be more explicit about our approach in data split for calibration and validation set.

I think also that the use of R2 could be confusing. The problem is that R2 can refer both to the coefficient of determination of a linear regression between predicted and observed (i.e., the square of the Pearson correlation coefficient) as well as to the amount of variance explained by the model. The first evaluates how close predicted and observed are to a fitted straight line, while the second evaluates how close they are to the 1:1 line. It is the latter that we are after in statistical validation, and this is properly assessed using the (Nash-Sutcliffe) Model Efficiency Coefficient (MEC). In the manuscript, the way you compute the $R^2$ seems to be wrong. You are missing 1 minus the ratio.

- Author's Response: We are sorry for bringing confusion here because of the wrong formula we reported in the methodology. We actually were supposed to report the formula below:
$R^2 = 1 - \frac{\sum_{i=1}^{n}(P_i - O_i)^2}{\sum_{i=1}^{n}(O_i - \mu_{obs})^2}$ where "P" is the predicted value, "O" is the observed/true value, " $\mu_{obs}$" the mean of the observed. This is now corrected in the manuscript. In the code that we provided (first submission), we actually used the *goof* function from the "ithir" package from Brandane P. Malone (https://rdrr.io/rforge/ithir/man/goof.html). The use of goof function is also demonstrated in the reference below. The R2 provided by *goof* function actually evaluates how close predicted and observed values are to the 1:1 line. For completeness, we went ahead and further tested the Nash-Sutcliffe efficiency (NSE) using the NSE function from the *"hydroGOF" package* (https://www.rdocumentation.org/packages/hydroGOF/versions/0.4-0/topics/NSE ). We found the same value for $R^2$ for both goof and NSE functions as shown below considering as an example the observed and predicted litter SOC stock by the local models using all covariates.

```
> North_allcov_litter<-read.xlsx("path/All_covariates_observed_predicted_Litter.xlsx",
sheetIndex = 1)
> NSE(North_allcov_litter$pred,North_allcov_litter$obs, na.rm=TRUE)
[1] 0.2109418
> goof(North_allcov_litter$obs, North_allcov_litter$pred)
      R2 concordance      MSE    RMSE      bias
1 0.2109418   0.3254933 355.4248 18.85271 -0.4370792
>
> Center_allcov_litter<-read.xlsx("path/All_covariates_observed_predicted_Litter.xlsx",
sheetIndex = 2)
> NSE(Center_allcov_litter$pred,Center_allcov_litter$obs, na.rm=TRUE)
[1] 0.2256804
> goof(Center_allcov_litter$obs, Center_allcov_litter$pred)
      R2 concordance      MSE    RMSE     bias
1 0.2256804   0.3360015 285.7372 16.90376 1.145549
```

```
>
> South_allcov_litter<-read.xlsx("path/All_covariates_observed_predicted_Litter.xlsx",
sheetIndex = 3)
> NSE(South_allcov_litter$pred,South_allcov_litter$obs, na.rm=TRUE)
[1] 0.3544111
> goof(South_allcov_litter$obs, South_allcov_litter$pred)
     R2 concordance     MSE     RMSE     bias
1 0.3544111   0.5172758 556.1596 23.58304 2.308194
```

Reference

Malone, Brendan P., Budiman Minasny, and Alex B. McBratney. *Using R for digital soil mapping*. Vol. 35. Basel, Switzerland: Springer International Publishing, 2017.

**Answers to reviewer 1**

The manuscript attempted to compare the model performance of global and local models in predicting humus layer, mineral soil and total SOC stock and to identify the controlling factors for SOC stock prediction. Besides, this study also investigated the effect different combinations of data from site characteristics and remotely sensed variables on model performance. The results from independent dataset indicated that the local models generally had better model performance than the global models. The only use of remotely sensed variables had limited predictive ability while site characteristics had better explanatory strength in estimating SOC stocks. The authors suggest that further work can focus on mapping these influential site covariates. The manuscript is overall well-written with clear objectives and reasonable methodology.

However, two major limitation of this manuscript are:

(1) in comparing global and local models, I can tell from Table 4 and Table 5 that some local models had higher R2 than that of global models while the remaining local ones had lower R2. Therefore, the conclusion that local models have a comparative advantage over global models is not convincing for me. Again, it is not fair to compare the *performance indicators as mentioned in this study for local and global models*. Instead, for *the global model, authors should also calculate indicators for three regions separately to make them comparable for these in local models;*

- Author´s Response: Thanks for your comments and time investment to read our manuscript. Some local models had indeed higher R2 than that of global models while the remaining local ones had lower R2. However, we considered their performance in relation to the specific category of predictors the models were built from using mainly the RMSE as the main decision criteria. Based on the RMSE the local models showed generally better performance than the global models in term of RMSE. In that regards, our conclusion was that in general local models with site specific covariates only or those built using all the covariates did in most cases perform marginally better than the global models. Ideally one could hypothesize that local models with all covariates should have the lowest RMSE, followed by the local models with site specific covariates while local models with the group of covariate would come last. This because we suppose that models would benefit from more information using all the covariates compared to when they are considered separately. All these local models would also consistently outperform all the global models. Since our results showed otherwise, it meant that (1) there is inference in the feature space because of some redundancy between the information covered by the different covariate (as discussed in the ms), (2) each location (North, Center, South) has some specificity which could be best described by either global or local models depending on whether the covariate split in building the trees and related subdataset translated into accurate sub-predictions which in turn will result in better accuracy when aggregated. Since global models are based on all the training set over Sweden they could outperform some local models because of the larger range of data they are built from. However, since we did not also record consistently the lowest RMSE for all the global models in spite of their "bigger data advantage", we conclude that a locally calibrated model have potential to outperform a global model provided the associated covariates best describe the landscape and the variability of the SOC stock in that landscape. For example, we observed that the local model associated with the group of covariate for the soil SOC stock ranked first with the lowest RMSE. This because topographical variables were more correlated with soil SOC stock and this is consistent with Northern Sweden being characterized by higher elevation compared to the central and Southern Sweden.

- The error metrics were actually computed based on the same validation set for both global and local models, this to avoid comparison bias. Table 3 actually shows the error metrics for the global models at the national scale when the independent validation set for the (different strata) North, Central and South Sweden were considered as one dataset. Table 4 shows the specific

error metrics for the local models against their respective validation set. Figure 2 presents the error metric results of each global model for the three subareas along with corresponding local models (unfortunately repeating error metrics of local models from Table 4 for the independent validation. We now removed repeated information related to the independent validation from table 4). To ensure clarity, we provided more details in the section (see lines 224 – 231) related to the data splitting procedure.

Manuscript lines (see lines 224 – 231): The RF models built on data covering the whole area of Sweden are hereafter called "global models". The RF models created for each of the subareas are hereafter reported as "local models". Considering the subareas as strata, the local models were built by randomly splitting the local datasets into calibration (80%) and validation (20%) subset. Each local model was validated against their respective local validation set. For comparison, the global models were validated using the same local validation set used for the local models. The data used for calibrating the global model was made up of the 80% random split of the three local training set (Northern, Central and Southern training set). The same approach is used for validation at a national scale by considering as one dataset the 20% split of the three local validation dataset (Northern, Central and Southern validation set). We trained both global and local models based on tenfold cross validation with 5 repetitions using the R "caret" package (Kuhn, 2015).

(2) As mentioned by authors, the site characteristics are only available at the visited plots, therefore the digital maps can not be produced by the models built with site characteristics, which certainly limits the usage of site characteristics. I am afraid that the importance of these site characteristics in this study have been overlooked as the observed site characteristics are directly used in independent validation which certainly ignores the inaccuracy of these site characteristics if they are mapped by certain algorithms. I mean, the site characteristics used in a fair independent validation should come from the predicted maps of these relevant site characteristics, not from observed data. Therefore, I suggest a major revision before it can be published.

- Author´s Response: We share the concern of the reviewer that all covariates should actually be maps for DSM of the whole area while the site specific data were only observations. Initially, the focus was mainly evaluating the geodata we now have versus site specific variables and not necessarily making a map. We made the map based on the geodata to give a visual trend of the distribution of the SOC stock. We also agree that the current study neglects to consider mapping uncertainties from potential maps of site specific variables which might actually translate into lower prediction accuracies of SOC stock compared to models based on their observation data. Our interpretation failed to put this clearly into perspective and we have now created a section discussing this issue. To elaborate more on this issue in that section, we also made some preliminary maps of the site specific variables and used these maps only as covariate to assess the quality of their predictions.

Manuscript lines (see lines 585 – 615): see section 4.4 Implication and limitations of the study DSM relies on existing maps for building regression models and ultimately prediction for mapping. The quality and accuracy of predictions depend as discussed earlier on choosing the most relevant covariates in relation to the target to be predicted. The present study revealed that covariates which were available as maps did contribute to the MDA, but site characteristics were more prominent in relation to the SOC stock in Sweden. This might suggest that mapping these variables that were more decisive for SOC stock prediction and including them as covariates for mapping may improve accuracy. Since the primary focus of the present study was mainly to evaluate the GoC data versus the SSC data at different scale of modelling and not primarily for making a map, the observed data of the latter were used. However, since only the observed data of the site characteristics were considered, this study fails to consider that the mapping of these site characteristics would involve modelling errors and the propagation of these errors into the final maps of these site variables might actually reduce their predictive power. For completeness, we carried out a preliminary mapping of the site characteristics (SI-

11) using additional soil inventory data, random forest and the GoC as predictors. These mapped site characteristics (mSSC) were then used as covariate for predicting the SOC stocks.

Table 6 presents the error metrics after independent validation for both local and global models along with the percentage margin of the RMSE in relation of the models based on GoC. First, the local mSSC based models still recorded lower RMSE as compared to global mSSC models. Compared to the GoC models, the overall positive percentage margin of the RMSE for the independent validations indicated that the mSSC models recorded the lowest RMSE. However, when assessing the RMSE margin between the global models of GoC and SSC, negative percentages were mainly recorded for the Northern Sweden independently of the depth. This indicated that the mSSC based global models were less accurate at predicting SOC stock locally in the Northern Sweden. On the other hand, the mSSC based local model did present a better prediction for Northern Sweden for the humus layer and mineral SOC stock. The mean SOC predictions based on the mSSC showed a stronger increasing gradient from Northern to Southern Sweden (SI-12) as compared to the pattern observed with the GoC maps. However, the uncertainty distribution was of similar magnitude (SI-12) as those observed for the maps based on GoC probably due to error propagations as these covariates were used to make the site specific characteristics maps. This suggests that the SSC should still be supplemented for improvement at this stage with other covariates different from the GoC such as multi-temporal spectral (e.g. normalized difference vegetation index) data that are able to capture vegetation dynamic in forests. Notwithstanding the possibility of error propagation, the study of which was beyond the scope of this study, the preceding results tend to confirm the potential of the high-resolution maps of the site characteristics to contribute to the improvement of SOC stock prediction as compared to using only the GoC data. Given that the preliminary mappings of the SSC recorded low kappa (0.17 – 0.48) values (SI-11) at this stage further improvements are still necessary to improve their predictive ability and associated coefficient of variations.

Specific comments are listed below:

Line 49: which are nonspatial environmental covariates? If they are not spatial, how they can be used in DSM?

- Author´s Response: Sentence corrected with removal of the word non-spatial
  Manuscript lines (see lines 47 – 50): DSM aims at estimating the spatial distribution of soil classes or soil properties by coupling field and laboratory observations with spatial environmental covariates via quantitative relationships.

Line 58: Authors miss at least two papers on comparing local and global models in DSM.

Piikki, K., & Söderström, M. (2019). Digital soil mapping of arable land in Sweden–Validation of performance at multiple scales. Geoderma, 352, 342-350.

Song, X. D., Wu, H. Y., Ju, B., Liu, F., Yang, F., Li, D. C., ... & Zhang, G. L. (2020). Pedoclimatic zone-based three-dimensional soil organic carbon mapping in China. Geoderma, 363, 114145.

- Author´s Response: Thanks for mentioning these papers. The suggested papers have been included.
  Manuscript lines (see lines 58 – 60): However, models could be calibrated separately for subareas and their predictions can then be combined to cover the whole area (Somarathna et al., 2016; Piikki and Söderström, 2019; Song et al., 2020).

Line 66: SCORPAN also includes soil information compared to Jenny's soil forming theory.

- Author´s Response: We agree with the comment and included soil information in addition to location position for additional elements from SCORPAN compared to Jenny´s soil forming theory.

Manuscript lines (see lines 65 – 67): Building on the soil state-factor (climate, organisms, relief, parent material, age) equation developed by Jenny (1941), McBratney et al. (2003) introduced the conceptual framework for DSM referred to as SCORPAN which complemented the former with the inclusion of soil information and location coordinates.

Lines 89-90: It seems to me that this manuscript does not take the SOC changes during three periods into account. I would add relevant statements with supporting references.

-   Author´s Response: The reviewer is right that the manuscript does not focus on SOC changes over the three inventory periods. We rather focus on SOC stock by considering the averaged SOC at plot level as a good representative of the plots which we indicated in lines 107 – 110. We therefore explained clearly our focus and added further references for the dataset.
    Manuscript lines (see lines 89 – 93): The NFSI runs concurrently every year with the NFI and consist in repeated survey of forest vegetation and soil chemical and physical properties (Stendahl et al., 2017;Ortiz et al., 2013). Data from the following inventory periods were considered in the present study: 1993 – 2002, 2003 – 2012 and 2013 – 2015. However, the present paper did not focus on SOC changes over these three inventory periods but on SOC stock using plot scale as a unit.
    Manuscript lines (see lines 110 – 112): Since potential SOC stock change is very small compared to the entire SOC stock the averaged SOC stock between the inventories was considered representative of the plots and was therefore considered for all computations and modelling in order to reduce variability between plots

Lines 103-104: Since mineral soil is sampled at 0-10, 10-20 and 55-65 cm depth, which kind of interpolation is used to harmonize them at 0-50 cm? Please provide more details.

-   Author´s Response: The carbon stocks for each layer with data is calculated. Thereafter the carbon pool is assumed to change linearly between measured layers. Even if the deepest sampled horizon (55-60 cm) is deeper than the top 50 cm it is utilized in the interpolation but the carbon stock below 50 cm is not counted in.
    Manuscript lines (see lines 107 - 110): The total mineral SOC stock down to 50 cm depth for each site is calculated using the SOC stock of measured layers with empirical model for bulk density (Nilsson and Lundin, 2006), corrections for stoniness (Stendahl et al., 2009) and linear interpolation between measured layers

Line 158: RF is commonly used instead of RFR.

-   Author´s Response: RFR has been replaced by RF throughout the document.
    Manuscript lines (see lines 65): The Random Forest (RF) algorithm was selected for SOC stock prediction.

Line 216: Which method is used in recursive feature elimination in caret package? Random forest?

Author´s Response: Random Forest was used as method in recursive feature elimination in caret package. We now added this method in the referred line.
Manuscript lines (see lines 220 – 221): …and (2) the recursive feature elimination (RFE) using RF as method to select the optimal set of covariates for each RF model.

Line 247: What does p mentioned here?

-   Author´s Response: The p stands for probability value. We corrected and put the p in bracket after probability.
-   Manuscript lines (see lines 250): The latter is the graphical representation of the proportion of time the actual values of SOC stock fall within a series of probability (p) of prediction intervals (PI) limited by (1-p)/2 and (1+p)/2 quantiles.

**Answers to reviewer 2**

The manuscript shows the spatial prediction of carbon stocks in forests of Sweden for tow compartments (organic layer and mineral soil 0-50 cm depth) and the sum of the two. Random forest with parameter tuning and feature selection is used and prediction intervals are computed by quantile regression forest. Different model configuration is shown along two axes: 1) The data is split into subregions and separate model calibrations are presented. The results are then compared to a model covering the complete area. 2) Different subsets of covariates as model input are formed according to their origin. Covariates only available at the observed sites and geodata available for whole Sweden were tested separately. In addition models combining all covariates were fitted and compared to the other two versions.

The methods are fully described and the results thoroughly discussed. The prediction intervals are validated with data not used to create the models which is so far rarely seen. Moreover, the distribution of the actual data is compared to the distribution of all predicted pixels. The manuscript is well written and the figures mostly are well prepared. Well done, Ozias!

However, I share the concerns of referee #1:

1. On page 6 lines 227–232 the data splitting approach is explained. I do not fully understand how it was actually done as I got puzzled by the last sentence. Were the validation data the same data points for the global and the local models (last sentence) or were they independently chosen (rest of this paragraph)? Nevertheless, the validation statistics should be computed on the exact same validation dataset. Especially the measure R2 is very sensitive to different values of observed data. This means for the global model three local sets of validation statistics should be computed on the same data points. Of course overall statistics can also be presented, but should not be compared to the local statistics, because the standard deviation of the total vector of observed values might be larger. To achieve such a validation the 20-80% splitting should be done as stratified random sampling from the data points with the subareas as strata (which maybe already has been done).

- Author´s Response: Thanks for your enquiry. The error metrics were actually computed based on the same validation set for both global and local models, this to avoid comparison bias. Table 3 actually shows the error metrics for the global models when the independent validation set for the (different strata) North, Central and South Sweden were aggregated. Table 4 shows the specific error metrics for the local models against their respective validation set. Figure 2 presents the error metrics results of each global model for the three subareas along with corresponding local models (unfortunately repeating error metrics of local models from Table 4 for the independent validation. We have now removed redundant information from table 4). To ensure clarity, we provided more details in the section (see lines 224 – 231) related to the data splitting procedure.
- Manuscript lines (see lines 224 – 231): The RF models built on data covering the whole area of Sweden are hereafter called "global models". The RF models created for each of the subareas are hereafter reported as "local models". Considering the subareas as strata, the local models were built by randomly splitting the local datasets into calibration (80%) and validation (20%) subset. Each local model was validated against their respective local validation set. For comparison, the global models were validated using the same local validation set used for the local models. The data used for calibrating the global model was made up of the 80% random split of the three local training set (Northern, Central and Southern training set). The same approach is used for validation at a national scale by considering as one dataset the 20% split of the three local validation dataset (Northern, Central and Southern validation set). We trained both global and local models based on tenfold cross validation with 5 repetitions using the R "caret" package (Kuhn, 2015).

2. The covariates only available at the observed locations cannot be used to create prediction at locations without any observation. Hence, they are not useful to create area-wide maps unless one creates maps of these site specific characteristics beforehand. The same geodata would be used to create these predictions. Such a two-step prediction approach (some sort of a random forest co-kriging) is only meaningful if 1) there are more observations of this intermediate response (site characteristic) than for the final target response (SOC stock) which I understand is not the case for the current data set; 2) the intermediate response results for some reason in much better model performance and predictive accuracy. The current study neglects to consider errors of the needed spatial prediction of the site characteristic covariates. The error of the spatial prediction might possibly be as high as to render these covariates useless. At least, the interpretation should put this clearly into perspective.

- Author´s Response: We share the concern of the reviewer that all covariates should actually be maps for DSM of the whole area while the site specific data were only observations. Initially, the focus was mainly evaluating the geodata we now have versus site specific variables and not necessarily making a map. We made the map based on the geodata to give a visual trend of the distribution of the SOC stock. We also agree that the current study neglects to consider mapping uncertainties from potential maps of site specific variables which might actually translate into lower prediction accuracies of SOC stock compared to models based on their observation data. Our interpretation failed to put this clearly into perspective and we have now created a section discussing this issue. To elaborate more on this issue in that section, we made some preliminary maps (which need further improvements because of low kappa values) of the site specific variables using Random Forest and additional dataset from the inventory data. These maps were then used as covariates to predict SOC stock.

Manuscript lines (see lines 585 – 615): see section 4.4 Implication and limitations of the study DSM relies on existing maps for building regression models and ultimately prediction for mapping. The quality and accuracy of predictions depend as discussed earlier on choosing the most relevant covariates in relation to the target to be predicted. The present study revealed that covariates which were available as maps did contribute to the MDA, but site characteristics were more prominent in relation to the SOC stock in Sweden. This might suggest that mapping these variables that were more decisive for SOC stock prediction and including them as covariates for mapping may improve accuracy. Since the primary focus of the present study was mainly to evaluate the GoC data versus the SSC data at different scale of modelling and not primarily for making a map, the observed data of the latter were used. However, since only the observed data of the site characteristics were considered, this study fails to consider that the mapping of these site characteristics would involve modelling errors and the propagation of these errors into the final maps of these site variables might actually reduce their predictive power. For completeness, we carried out a preliminary mapping of the site characteristics (SI-11) using additional soil inventory data, random forest and the GoC as predictors. These mapped site characteristics (mSSC) were then used as covariate for predicting the SOC stocks.

Table 6 presents the error metrics after independent validation for both local and global models along with the percentage margin of the RMSE in relation of the models based on GoC. First, the local mSSC based models still recorded lower RMSE as compared to global mSSC models. Compared to the GoC models, the overall positive percentage margin of the RMSE for the independent validations indicated that the mSSC models recorded the lowest RMSE. However, when assessing the RMSE margin between the global models of GoC and SSC, negative percentages were mainly recorded for the Northern Sweden independently of the depth. This indicated that the mSSC based global models were less accurate at predicting SOC stock locally in the Northern Sweden while the mSSC based local model did present a better prediction for Northern Sweden for the humus layer and mineral SOC stock. The mean SOC predictions based on the mSSC showed a stronger increasing gradient from Northern to Southern Sweden (SI-12) as compared to the pattern observed with the GoC maps. However, the uncertainty distribution was of similar magnitude (SI-12) as those observed for the maps based on GoC probably due to error propagations as these covariates were used to make the site specific characteristics maps.

This suggests that the SSC should still be supplemented for improvement at this stage with other covariates different from the GoC such as multi-temporal spectral (e.g. normalized difference vegetation index) data that are able to capture vegetation dynamic in forests. Notwithstanding the fact that was obviously error propagation, the study of which was beyond the scope of this study, the preceding results tend to confirm the potential of the high resolution maps of the site characteristics to contribute to the improvement of SOC stock prediction as compared to using only the GoC data. Given that the preliminary mappings of the SSC recorded low kappa (0.17 – 0.48) values (SI-11) at this stage further improvements are still necessary to improve their predictive ability and associated coefficient of variations.

**Specific comments**

Maybe consider using "covariates" instead of "variables" as in the rest of the text. It would be more specific.
- Author´s Response: Thanks. "covariates" was considered
  Manuscript: Predicting the spatial distribution of soil organic carbon stock in Swedish forests using remotely sensed and site-specific covariates

Please add the reference depth of SOC stocks. Mineral soil = 0-50 cm (only), otherwise one would think down to bedrock or profile depth of usually 1-1.5 m

- Author´s Response: We added the depth to the mineral soil.
  Manuscript (line 28 - 31): The most important covariates that influence the humus layer, mineral soil (0 – 50 cm) and total SOC stock were related to the site characteristic covariates and include the soil moisture class, vegetation type, soil type and soil texture.

Not all of the mentioned geodata are remote sensing data, e.g. the geological map. Better use a broader formulation.
- Author´s Response: We replace "remote sensing by covariates"
  Manuscript (line 73 - 74): These studies and many others rely mostly on covariates existing as maps while survey data which present site specific information are left out during modelling.

Should "c." be "ca."?
- Author´s Response: yes it is ca. It has been replaced. Thanks.
  Manuscript (line 99 - 100): Soil samples are collected in a subset of the plots with humus sampling on ca. 10 000 plots and mineral soil sampling on ca. 4500 plots (Stendahl et al., 2017).

This does not include the O horizon, right? Maybe reformulate "below the O horizon down to 30 cm".
- Author´s Response: Thanks for the suggestion. We reformulated to "below the O horizon down to 30 cm
- Manuscript (line 103 - 104): Therefore the content of inorganic carbon in mineral soil is considered negligible in the study area. Humus layer volumetric samples are taken using a soil core (core diameter 10 cm) below the O horizon down to 30 cm depth.

if available, better cite a data documentation that appears in the references list.
- Author´s Response: Citation has been added.
  Manuscript (line 120 - 122): Topographic covariates were computed from high-resolution digital elevation models (DEM) derived from Light Detection And Ranging (LiDAR) produced by the Swedish National Mapping Agency. It was originally created with 2-m spatial resolution (Dowling et al., 2013).

Please give the algorithm of the aggregation procedure. Bilinear interpolation? The software is usually secondary, if the algorithm becomes clear.

- Author´s Response: The bilinear interpolation algorithm was provided.
  Manuscript (line 122 - 124): However, the initial DEM was resampled in ArcGIS 10 software package using the aggregation procedure with bilinear interpolation to a final resolution of 10 m × 10 m which is reasonable for the data considered in the present study.

Please cite, or give used algorithms and versions. (maybe add in covariate table).

- Author´s Response: Citation has been added.
  Manuscript (line 124 - 125): The topographical covariates were computed using the SAGA GIS software (Conrad et al., 2015).

Please cite in reference list as a web source.

- Author´s Response: Citation added.
  Manuscript (line 127 - 128): The depth to groundwater was obtained from the Swedish Forest Agency (SGU, 2018) and computes the difference in elevation in relation to surrounding cells following the vertical flow path.

Please add citation here. Or was this done in this study?

- Author´s Response: This was done in this study
  Manuscript (line 141 - 143): The resulting gamma-ray data as well as the geochemical data were interpolated in this study into maps either by ordinary kriging or inverse distance weighing when geostatistic assumption such as normal distribution were not met.

Please check again this function notation. The superscript 1 is likely an indicator function, usually used without superscript. Moreover, "j" is not defined in the text.

- Author´s Response: Thanks for the observation. We have checked and reproduce it as mentioned in Meinshausen (2006). j is also defined.
  Manuscript (line 182 - 185): The weight vector (Meinshausen, 2006) w_i is defined as follows:

$$ - \quad w_i(x, \theta) = \frac{1_{\{x_i \in R_{l(x,\theta)}\}}}{\#\{j : x_j \in R_{l(x,\theta)}\}} \qquad (2) $$

Please indicate which values where tested for mtry in the grid search. Was ist 1:p with p: number of covariates?

- Author´s Response: We provided the values tested: 2: p, with n the number of covariates.
  Manuscript (line 196 - 198): To reduce computational load, the ntree was set at 500 while the mtry was tuned using the grid search (2: p, with p the number of covariates) method in the R "caret" package (Kuhn, 2015).

Please give a citation. e.g. Hastie et al., 2011.

- Author´s Response: Ciatation added.
  Manuscript (line 199 - 198): The importance of each input predictor can be assessed by the RF based on the mean decrease accuracy (MDA) (Hastie et al., 2011).

Were 10 m overlap really enough to allow for a smooth transition? This is just one pixel...

- Author´s Response: Thanks for notification. This was a mistake. We corrected to 4km which was actually used.
  Manuscript (line 208 - 210): A buffer of 4 km was considered for the shapefiles of each subarea to create overlapping zones which ensured smooth transition while merging by averaging the SOC stock values within these shared units.

Please specify what kind of algorithm and maybe also the function name you used for de-correlation.

- Author´s Response: The function has been added. We use the Pearson´s correlation.
- Manuscript (line 220 - 222): Moreover, to reduce computation time while keeping relatively the same level of accuracy, we (1) used feature pre-processing capabilities implemented in the caret package (Kuhn, 2017) of R to remove highly correlated (Pearson's correlation) expressions using a cutoff point of 0.80…

maybe use "group of covariates"
- Author´s Response: Thanks for the suggestion. We adopted "group of covariates" throughout the document.

Please specify: How was the splitting done into 20 vs. 80%? Simple random sample?
- Author´s Response: Yes a simple random split was carried out. This has now been specified. Manuscript (line 229 - 230): Considering the subareas as strata, the local models were built by randomly splitting the local datasets into calibration (80%) and validation (20%) subset.

Bias is reported in the table, but definition is missing.
- Author´s Response: Thanks for notifying. Bias has now been defined. Manuscript (line 240 - 245): Bias= $\mu\_pred - \mu\_obs$

squared correlation coefficient, but rather the formula of the type: R2 = 1 - sum(P i - O i) / sum(O i - mean(O)). Caret has an implementation, see also help site for the formula: R2(p,o, form = "traditional")
- Author´s Response: Thanks for notifying. This has now been corrected.
  Manuscript (line 240 - 245): $R^2 = 1 - \frac{\sum_{i=1}^{n}(P_i - O_i)^2}{\sum_{i=1}^{n}(O_i - \mu_{obs})^2}$

upper end, not at both ends simultaneously. But, the one-sided version of this plot is harder to read. However, the lines in the plots follow the 1:1-line quite well, so little difference is to be expected.
- Author´s Response: In our study, the lines follow the 1:1 line quite well and we also thought that other upend end make no difference in this case.

Just to be precise ;-). Please consider using the peer-reviewed paper of this work. Nussbaum et al. 2014 (https://gmd.copernicus.org/articles/7/1197/2014/). We used an external-drift kriging model (that includes a linear regression of course). Then, the data was not only form the Swiss National Forest Inventory. Considerdropping this information as it is of no importance to the discussion.
- Author´s Response: We have removed the reference.

A bit hard to read. Consider reformulation.
- Author´s Response: We have reformulated that section and hope that it is now better to read. Manuscript (line 419 - 430): Low explained variances in predictive modelling could be related to different factors (Nelson et al., 2011). For example, the omission of key covariates with greater explanatory power or conversely using non-essential covariates with very low explanatory power which only increase the prediction error variance. Omitting key covariates in relation to SOC stock for forest ecosystem in the present study is less likely since covariates considered in this study well represent the surrogates for soil forming factors considered in the SCORPAN equation defined by McBratney et al. (2003). In addition, the removal of redundant and non-informative covariates was carried out via dimension reduction with the exclusion of highly correlated covariates and elimination of some others via recursive feature elimination. However, the Pearson correlations (min = 0, max = 0.28) between covariates and the different SOC stock were found to be poor though significant for most of the predictors (Table 5, SI 2-4). This could be expected because the data cover a wide range of different site conditions, soil types and parent materials.

Figure SI5
- Author´s Response: Correction made.

Manuscript (line 436): On the other hand, applying geostatistical approaches (SI-5) for the humus layer…

Computing variograms from the response directly might reveal spatial autocorrelation. However, I do not recommend to do that as putting as much of the correlation into the regression part e.g. with random forest is to be preferred. Better reformulate the text.
- Author´s Response: We have reformulated focusing on the response directly.
  Manuscript (line 435 - 438): On the other hand, applying geostatistical approaches (SI-5) for the humus layer, mineral soil and total SOC stock revealed very low spatial autocorrelation for the different SOC stock suggesting that the structure of these SOC data is having a shorter range than the sampling interval.

Is it not the other way around, overestimation of small values and underestimation of the large values? RF has the tendency to predict the mean if correlation of response and covariate is weak. Hence, the extremes are not well predicted.
- Author´s Response: Yes generally there is overestimation of small values and underestimation of the large values. We were commenting on the trend which can be seen in Fig. 3. We mainly observed an underestimation of the low and high values in our context.

Conclusion difficult, without considering the error of such predicted maps.
- Author´s Response: We now provided error metrics based from RF models based on preliminary mapping of the site specific covariates which showed that they performed generally better than the group of covariates.
Manuscript (line 580 - 607): Section on 4.4 Implication and limitations of the study

DSM relies on existing maps for building regression models and ultimately prediction for mapping. The quality and accuracy of predictions depend as discussed earlier on choosing the most relevant covariates in relation to the target to be predicted. The present study revealed that covariates which were available as maps did contribute to the MDA, but site characteristics were more prominent in relation to the SOC stock in Sweden. This might suggest that mapping these variables that were more decisive for SOC stock prediction and including them as covariates for mapping may improve accuracy. Since the primary focus of the present study was mainly to evaluate the GoC data versus the SSC data at different scale of modelling and not primarily for making a map, the observed data of the latter were used. However, since only the observed data of the site characteristics were considered, this study fails to consider that the mapping of these site characteristics would involve modelling errors and the propagation of these errors into the final maps of these site variables might actually reduce their predictive power. For completeness, we carried out a preliminary mapping of the site characteristics (SI 11) using additional soil inventory data, random forest and the GoC as predictors. These mapped site characteristics (mSSC) were then used as covariate for predicting the SOC stocks.

Table 6 presents the error metrics after independent validation for both local and global models along with the percentage margin of the RMSE in relation of the models based on GoC. First, the local mSSC based models still recorded lower RMSE as compared to global mSSC models. Compared to the GoC models, the overall positive percentage margin of the RMSE for the independent validations indicated that the mSSC models recorded the lowest RMSE. However, when assessing the RMSE margin between the global models of GoC and SSC, negative percentages were mainly recorded for the Northern Sweden independently of the depth. This indicated that the mSSC based global models were less accurate at predicting SOC stock locally in the Northern Sweden while the mSSC based local model did present a better prediction for Northern Sweden for the humus layer and mineral SOC stock. Notwithstanding the fact that there could be error propagation, the study of which was beyond the scope of this study, the preceding results tend to confirm the potential of the high resolution maps of the site characteristics to contribute to the improvement of SOC stock prediction as compared to using only the GoC data. Given

that the preliminary mappings of the SSC recorded low kappa values (SI11) at this stage further improvements are still necessary to improve their predictive ability.

maybe use "open source"
- Author´s Response: Thanks for suggesting. "open source" adopted.
  Manuscript (line 579 - 583): DSM approach in the present study allows: flexibility in future improvement upon acquisition of new covariates or data point, repeatability in modelling with the application of the same modelling principles using open source software (e.g. R) and capitalizing on multi-source information (topography, site characteristics, forest data, gamma radiometry and geochemical data).

In the correlation matrix in the supplement (Figure SI2-4) the aspect is given. As it is 0-360 degrees it is not meaningful as a covariate. Consider dropping it everywhere.
- Author´s Response: We dropped aspect in mapping the site covariates. In most cases, it was also dropped by RFE when all the variables were considered.

Please check again. These skewness values seem not possible from the other information.
- Author´s Response: We are sorry for this mistake. The n values were repeated twice. We have now corrected this in the table.

What kind of test was performed? To be complete it would be good practice to report the test statistic
- Author´s Response: Thanks for pointing out this. We have now changed the title of the table which take into account the test.
  Manuscript: Table 5: Random Forest variable importance for the global and local models for the humus layer, mineral soil and total SOC stock with their associated Pearson´s coefficient of correlation with the covariates (values in bracket, *: $p \leq 0.05$, $p \leq 0.05$;**: $p \leq 0.01$;***: $p \leq 0.001$)

Please specify how the R2 and the RMSE were computed. Cross-validation independent data set obtained by data splitting?
Author´s Response: We have now specified that the r2 and RMSE were based on the independent data.
- Manuscript (line 950): Figure 2: Local and global models ranked by decreasing RMSE per subareas and category of covariates along with corresponding R2 based on the independent validation dataset. A: litter layer, B: mineral soil layer, C: total soil layer, SSC: site specific covariates, GoC: Group of covariates, allC: all covariates

These maps are very difficult to read. Consider either adequate generalization of the 10 m pixels for the display scale or showing but a section. I would actually be interested if there are artefacts at the border of the subareas like sudden change in values that are hard to explain. Such artefacts might be a reason against using local models even if their model performance is better.
- Author´s Response: Thanks for pointing this out. It is actually challenging to have a color palette that display distinctly the pattern over the whole Sweden. We have now used another color palette for the SOC stocks and hope that visualization has improved. Visualizing further the maps, we did not observe any sudden change in values that would raise particular attention as the pattern at the border showed similar pattern for some location inside Sweden.

**Answers to Reviewer 3**

**General comment**

The manuscript is well written and generally clear, with an appropriate structure and the information provided in tables and figures is useful and necessary, although I have some specific comments for the presentation of some figures. The length of the paper is appropriate, as the presentation of the results is synthetic, and the discussion gives a concise explanation of the observed results supported by other findings in the SOC literature.

This research paper investigated the key variables for predicting soil organic carbon (SOC) stocks in the litter layer, mineral soil, and total SOC of forest soils in Sweden, and maps its spatial distribution using random forest models. The study compares the accuracy of global models (calibrated for the whole study area, Sweden) and local models for north, central and southern Sweden. The calibration data originated from the Swedish National Forest and as predictor variables they compared three different sets: 1) only site variables observed at the sampling plots, 2) remote sensing variables, and 3) all variables.

My main comment may be more a suggestion for the follow-up study. Mapping some of the site variables that were more decisive for SOC prediction (soil moisture class, vegetation type, soil type and soil texture) and including them as covariates for mapping may improve the model accuracy for mapping. However, as these variables will be themselves estimated with statistical models, there may be an increase in the uncertainty due to error propagation. Hence, the uncertainty of the map for a model including all variables may be a conservative estimate. Also, in that future scenario, consider that if you calibrate the model with the data observed at the plots but map it with the gridded estimates of the site variables, the accuracy may also overestimated. If you calibrate the model with the gridded predictions for soil moisture class, soil texture, etc., perhaps they may not be as relevant covariates, and maybe the accuracy of the model will not be the highest.

I recommend acceptance after minor revisions, for the following sections.

**Specific comments**

L51-53: Maybe you can include geostatistics as part of the modelling methods.

- Author´s Response: We thank the reviewer for pointing out "geostatistics" which is also used for DSM. We now included additional citation (Mallik et al., 2020) related to "geostatistics" and mention it along linear models and machine learning.
Manuscript lines (see lines 49 – 54): Many studies used DSM approaches for predicting SOC stock at different scales and for various land use/land cover, climate and across a wide range of soil types (Söderström et al., 2016;Tranter et al., 2011;Beguin et al., 2017;Mansuy et al., 2014; Mallik et al., 2020). These studies use different modelling techniques ranging from geostatistics, multiple linear regression to machine learning models such as artificial neural network, support vector machine and boosted regression trees.

L55: consider changing "modelling over a large landscape" to "large extent".

- Author´s Response: Correction made
Manuscript lines (see lines 55 – 58): The accuracy and precision of predictions resulting from modelling over a large extent are often reported to be poor because of the spatial heterogeneity encompassing different soil types, topography and soil properties (Grimm et al., 2008;Schulp and Verburg, 2009;Schulp et al., 2013;Tang et al., 2017).

L66: include soil with "the inclusion of the location coordinates".

- Author´s Response: Correction made with also the inclusion of soil information
  Manuscript lines (see lines 65 - 66): Building on the soil state-factor (climate, organisms, relief, parent material, age) equation developed by Jenny (1941), McBratney et al. (2003) introduced the conceptual framework for DSM referred to as SCORPAN which complemented the former with the inclusion of soil information and location coordinates.

L86-94: This paragraph is somewhat confusing. Does the NFSI run every year or every 5 years? Maybe give a reference for the NFSI or the NFI datasets.

- Author´s Response: We are sorry for the lack of clarity of this paragraph. Actually, the permanent plots of NFI are re-inventoried after 5 years and after 10 years for those of NFSI. We now provided citations and remove the 5 years mistakenly attributed to NFSI.
  Manuscript lines (see lines 49 – 50): The NFSI runs concurrently every year with the NFI and consists in repeated survey of forest vegetation and soil chemical and physical properties (Stendahl et al., 2017;Ortiz et al., 2013). Lines 96 – 97: Each plot of the NFSI are inventoried once every 10 years.

L99-L100: Please, indicate if the content of inorganic carbon in mineral soil is negligible in the study area.

- Author´s Response: Due to sufficient leaching pedogenetic carbonates are not formed in Swedish soils. There are a few areas with CaCO3 in sedimentary bedrock where soils can contain some carbonates. However, they cover less than 1% of the Swedish forest area and hence we regard them as negligible in this case.
- Manuscript lines (see lines 99 – 103): Soil samples are collected in a subset of the plots with humus sampling on ca. 10 000 plots and mineral soil sampling on c. 4500 plots (Stendahl et al., 2017). Based on the NFSI dataset, pedogenetic carbonates are not formed in these soils due to sufficient leaching and also sedimentary bedrocks which could potentially contain CaCO3 cover less than 1% of Swedish forests. Therefore the content of inorganic carbon in mineral soil is considered negligible in the study area.

L104: what type of interpolation? Splines? Linear?

- Author´s Response: Thanks for the inquiry. The carbon stocks for each layer with data is calculated. Thereafter the carbon pool is assumed to change linearly between measured layers. Even if the deepest sampled horizon (55-60 cm) is deeper than the top 50 cm it is utilized in the interpolation but the carbon stock below 50 cm is not counted in.
  Manuscript lines (see lines 107 - 109): The total mineral SOC stock down to 50 cm depth for each site is calculated using the SOC stock of measured layers with empirical model for bulk density (Nilsson and Lundin, 2006), corrections for stoniness (Stendahl et al., 2009) and linear interpolation between measured layers

L150: the soil moisture class is later shown as a very important variable. Perhaps you could describe it a bit more. It refers to the frequency of the year it is dry/moist due to the proximity of the water table, or also influenced by soil texture (e.g., soil drainage class)?

- Author´s Response: The soil moisture class is determined in the field through field staff judgement of the location of the average ground water table over the vegetation season. The field staff uses indicators of which soil texture is one and others are topographic position and vegetation.
- Manuscript lines (see lines 154 - 157): The records of site characteristics (Table 1) are also carried out during the NFSI. Site description include soil types, soil moisture class, soil texture class, vegetation type and parent material class. The soil classification was based on the World

Reference Base (WRB) for soil resources. The location of the average ground water table over the vegetation season was the main criterion for defining classes of soil moisture.

L153: The field layer refers to the understory?

- Author´s Response: Thanks for the question. The field layer refers to the understory, mainly the herbaceous plants.
Manuscript lines (see lines 158 - 160): The vegetation type as reported in Table 1 was defined by combining the descriptions of the field layers which refer to the understory. Field layers consisted of four main types which are categorized from fertile to poor, namely herb types (tall or low), grounds without field layer, grass types and dwarf-shrub types.

L159: did you also used the QRF models to predict the 5th and 95th percentiles and create the 90% prediction interval?

- Author´s Response: Yes we took advantage of the capacity of the QRF to compute quantiles of the predictions directly from the bootstrap (bootstrap, aggregation, bagging). Then, the 5th and 95th percentiles were predicted to create the 90% prediction interval. We have added this specific information in our manuscript.
Manuscript lines (see lines 249 – 251): The QRF was used to predict all percentiles including the 5th and 95th percentile required to create the 90 %-prediction intervals. Finally, the coverage of the 90 %-prediction intervals by the observation from the validation set was also analyzed.

L201-203: I would have used a larger buffer, but I imagine that you assessed that this length was appropriate. Later in the results, I miss some information on how the local maps overlapped, whether or not there was some edge effect. In figure 6 it seems that there were not large differences in the boundaries.

- Author´s Response: We apologize for this reporting mistake. A 4 km buffer area was actually defined. This has been corrected in the manuscript.
Manuscript lines (see lines 208 - 210): A buffer of 4 km was considered for the shapefiles of each subarea to create overlapping zones which ensured smooth transition while merging by averaging the SOC stock values within these shared units.

L222-225: This sentence is not very clear. You split the dataset into a calibration (80%) and independent validation (20%). And then you apply the tenfold cross-validation, repeated 5 times, on the calibration subset, right?

- Author´s Response: Yes, that is correct. Data was split into a calibration (80%) and independent validation (20%). And then the tenfold cross-validation, repeated 5 times, was applied on the calibration subset. We have reformulated for better clarity.

Manuscript lines (see lines 224 – 231): The RF models built on data covering the whole area of Sweden are hereafter called "global models". The RF models created for each of the subareas are hereafter reported as "local models". Considering the subareas as strata, the local models were built by randomly splitting the local datasets into calibration (80%) and validation (20%) subset. Each local model was validated against their respective local validation set. For comparison, the global models were validated using the same local validation set used for the local models. The data used for calibrating the global model was made up of the 80% random split of the three local training set (Northern, Central and Southern training set). The same approach is used for validation at a national scale by considering as one dataset the 20% split of the three local validation dataset (Northern, Central and Southern validation set). We trained both global and local models based on tenfold cross validation with 5 repetitions using the R "caret" package (Kuhn, 2015).

L276-285: Here, the results for the independent validation are a bit short compared with the report for the cross-validation, or if you refer to the independent validation as "local models after cross validation" (L282), maybe that it not very clear. I was looking at Table 4 and the numbers don't seem to correspond completely. Please revise this (e.g., R2 for allV at independent validation 12-36 % and 17-32 % at cross-validation).

- Author´s Response: We have now modified Table 4 which presents only the cross-validation results of the local models. This because the error metrics of these local independent validations were somehow repeated in Figure 2. Thanks for notifying about the numbers which are now corrected for the cross-validation results in Table 4.
  Manuscript lines (see lines 283 - 284): The variances explained by the local models based on cross validation varied from 17% to 32% for allC models, 12% to 25% for the SSC models and from 5% to 20% for the GoC models.

L292-293: This sentence is not completely clear. Please specify that the local models had better performance than the global models in term of RMSE within each set of variables.

- Author´s Response: Sentence corrected.
  Manuscript lines (see lines 296 – 297): For the humus layer (Fig. 2A), the mineral (Fig. 2B) and total soil layers (Fig. 2C), the local models had in general better performance than the global models in term of RMSE within each set of variables.

L360-362: Could you provide the map of standard deviation?

- Author´s Response: We have computed the percentage of the ratio standard deviation/mean SOC stock prediction (coefficient of variation) to express the uncertainties for easy interpretation.
  Manuscript : see Figure 6

L491-495: The relevance of soil texture as predictor of SOC stock is also explained by the physicochemical SOC stabilization mechanisms. Clay minerals and clay and silt sized particles generally have a positive correlation with mineral SOC stocks, as the association of organic matter with mineral surfaces, and occlusion inside aggregates hinders microbial decomposition and enhances SOC accumulation. There are many references in the literature on this topic. For example:

Lützow, M.V., Kögel-Knabner, I., Ekschmitt, K., Matzner, E., Guggenberger, G., Marschner, B. and Flessa, H., 2006. Stabilization of organic matter in temperate soils: mechanisms and their relevance under different soil conditions–a review. European journal of soil science, 57(4), pp.426-445.

- Author´s Response: Thanks for pointing out the role of physicochemical mechanisms in the stabilization of SOC. We have agreed with this argument and this aspect has now been taken into account in our discussion.
  Manuscript lines (see lines 503 – 507): On the other hand, the relevance of soil texture as predictor of SOC could be related to the physicochemical SOC stabilization mechanisms. Clay minerals as well as clay and silt sized particles generally have a positive correlation with mineral SOC stocks, as the association of organic matter with mineral surfaces, and occlusion inside aggregates hinders microbial decomposition and enhances SOC accumulation (Lützow et al., 2006; Zhang et al., 2020).

L550-554 & 559: Consider my comment on how there would be error propagation and the final uncertainty of the SOC predictions may be increased when maps of these site attributes are included as predictors for mapping. However, I would also include these maps as predictor variables when they are available.

- Author´s Response: We indeed agree with the fact that predictions with the maps of the site specific variables will involve additional uncertainties in final maps and prediction accuracy will drop as compared to our current approach which focused only on the observation data. Our interpretation failed to put this clearly into perspective and we have now created a section discussing this issue. To elaborate more on this issue in that section, we made some preliminary maps (which need further improvements because of low kappa values) of the site specific variables using Random Forest and additional dataset from the inventory data. These maps (SI-11) were then used as covariates to predict SOC stock (SI-12).

[revised manuscript text omitted]

**Technical comments**

L20: "Random Forest models".

- Author´s Response: Correction made.
  Manuscript lines (see line 20): We use the Swedish National Forest Soil Inventory (NFSI) database and a digital soil mapping approach to evaluate the prediction performance using Random Forest models calibrated locally for the northern, central and southern Sweden (local models) and for the whole Sweden (global model).

L43: Place the abbreviation of carbon (C) before in the text, in line 35 when you use it for the first time.

- Author´s Response: Correction made.
  Manuscript lines (see lines 35): About 30 % of the global terrestrial carbon (C) stock is stored in forests with 60 % located below ground (Pan et al., 2011)….(see line 43) In that context, analysis of the C cycle in forests is central to understanding management and climate-induced changes in global C pool.

L228: "built".

- Author´s Response: This section of the manuscript was changed and rephrased

L294: "best local models".

- Author´s Response: Correction made.
  Manuscript lines (see lines 297 - 298): The best local models were mostly associated with all covariates or site specific covariates especially for central and southern Sweden

L326: 40K (capitalize the K).

- Author´s Response: Correction made.
  Manuscript lines (see lines 333): A similar trend is observed in northern Sweden while the remaining models recorded 40K as second key variable in addition to soil moisture and soil type for the central and southern Sweden respectively.

L372: "When predictions were carried out".

- Author´s Response: Correction made.
  Manuscript lines (see lines 378): When predictions were carried out on the same validation set, local models including those of southern Sweden generally outperformed the global models

L385: "outperform"

- Author´s Response: Correction made.
  Manuscript lines (see lines 393 – 394): For example, local models in central Sweden required all covariates to outperform global models for the humus layer, mineral soil and total SOC stock.

L386-387: Please indicate that you refer to the best local models.

- Author´s Response: Correction made.
  Manuscript lines (see lines 394 – 396): The same pattern was observed for Southern Sweden except for the mineral SOC stock for which the best local model was associated with the SSC. The local best model for the total SOC stock in Northern Sweden was also associated with SSC.

L397: "remained low"

- Author´s Response: Correction made.
  Manuscript lines (see lines 406 - 408): However, despite the combination of these two category of covariates, the accuracy of the SOC stock prediction remained low for both the global models…

L475: "organic matter"

- Author´s Response: Correction made.
  Manuscript lines (see lines 482 - 483): This makes them less relevant in contrast to SSC taken at plot level which describe more closely factors controlling the decomposition and stabilization of organic matter.

Figure 6: Please, indicate in the caption which in the figure (left/right) are the global and the local models. Also, maybe you could include a figure with the standard deviation or the 5th and 95th percentiles (predicted with the quantile random forests regression) so we can also visualize the uncertainty. The colour scale for the total SOC stock (in less extent) but mainly for the mineral SOC stock is not very clear, as there are different tonalities of green and brown for different value ranges. Could you use a different sequential palette, like for the predictions of the humus layer? (maybe multi-hue sequential palette). For example, the package colorspace has many options.

- Author´s Response: Thanks for the comments. We have used another color scale for the figure 6 of the mineral soil SOC stock. We have computed the percentage of the ratio standard deviation/mean SOC stock prediction to express the uncertainties for easy interpretation.

Supplementary material S1: Maybe you can expand the supplementary material one more page and make the plots larger on their y axis. They are not very clear like this.

- Author´s Response: We have taken this into account by adding two more page to make the plots bigger.

When you map over the whole study area, what GIS layer did you use to mask non-forested areas? you could include that in the methods.

- Author´s Response: Thanks for the inquiry. We use only the forest dataset and subsequent results only refer to forested areas. However, the maps in figure 6 was extrapolated for the whole country not excluding non-forested areas because any use of these maps will only be related to forested areas.